# TFLEX: Temporal Feature-Logic Embedding Framework for Complex Reasoning over Temporal Knowledge Graph

**Xueyuan Lin**[1]    **Haihong E**[1]*    **Chengjin Xu**[2]*    **Gengxian Zhou**[1]
**Haoran Luo**[1]    **Tianyi Hu**[1]    **Fenglong Su**[3]    **Ningyuan Li**[1]    **Mingzhi Sun**[1]

[1] Beijing University of Posts and Telecommunications
[2] University of Bonn
[3] National University of Defense Technology

linxy59@mail2.sysu.edu.cn, ehaihong@bupt.edu.cn, xuc@iai.uni-bonn.de
z.gengxian@se18.qmul.ac.uk, {luohaoran, hutianyi}@bupt.edu.cn
sufenglong18@nudt.edu.cn, {jason.ningyuan.li, sunmingzhi}@bupt.edu.cn

## Abstract

Multi-hop logical reasoning over knowledge graph plays a fundamental role in many artificial intelligence tasks. Recent complex query embedding methods for reasoning focus on static KGs, while temporal knowledge graphs have not been fully explored. Reasoning over TKGs has two challenges: 1. The query should answer entities or timestamps; 2. The operators should consider both set logic on entity set and temporal logic on timestamp set. To bridge this gap, we introduce the multi-hop logical reasoning problem on TKGs and then propose the first temporal complex query embedding named Temporal Feature-Logic Embedding framework (TFLEX) to answer the temporal complex queries. Specifically, we utilize fuzzy logic to compute the logic part of the Temporal Feature-Logic embedding, thus naturally modeling all first-order logic operations on the entity set. In addition, we further extend fuzzy logic on timestamp set to cope with three extra temporal operators (**After**, **Before** and **Between**). Experiments on numerous query patterns demonstrate the effectiveness of our method.

## 1    Introduction

Multi-hop logical reasoning over knowledge graphs (KGs) is a fundamental issue in artificial intelligence. It aims to find the answer entities for a first-order logic (FOL) query which involves logical operators (existential quantification $\exists$, conjunction $\wedge$, disjunction$\vee$ and negation$\neg$). Generally, the query is parsed into computation graph, according to which subgraph matching is executed on KG to find the answers. The computation graph is a directed acyclic graph (DAG) whose nodes represent entity sets, and edges represent logical operators acting on entity sets. However, results are inevitably incorrect as KGs are incomplete and noisy. Besides, the computation complexity will spiral for large-scale KGs or large queries. Therefore, people propose to embed query into low-dimensional space to solve the problem.

Query embedding (QE) learns the embeddings of queries and entities, so that answer entities are close to its queries in the embedding space. It has attracted arising attention, as low-dimension embeddings can model implicit dependency and reduce computation. There follows a series of QE methods, including GQE[1], Query2box[2], BetaE[3], ConE[4], etc. However, existing works only focus

---

*Corresponding Authors

**Temporal Complex Query**

$q[V_?] = V_?, \ \exists \, T_a, T_b, T_c :$
be_elected_as(François Hollande, President of France, $T_a$) $\wedge$ **After**($T_a, T_c$)$\wedge$
step_down_from(François Hollande, President of France, $T_b$) $\wedge$ **Before**($T_b, T_c$)$\wedge$
make_a_visit(Xi Jinping, $V_?, T_c$) $\wedge \neg$make_a_visit(Barack Obama, $V_?, T_c$)

**Computation Graph**

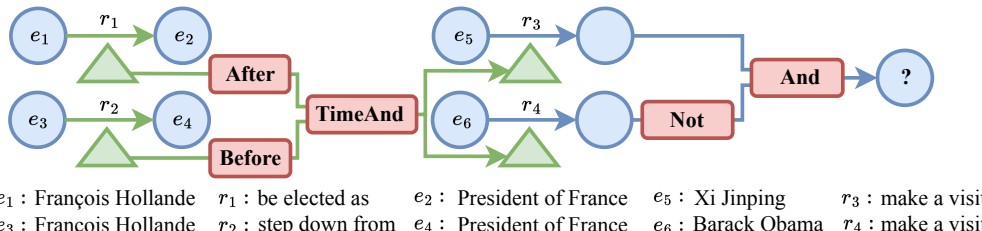

$e_1$ : François Hollande  $r_1$ : be elected as  $e_2$ : President of France  $e_5$ : Xi Jinping  $r_3$ : make a visit
$e_3$ : François Hollande  $r_2$ : step down from  $e_4$ : President of France  $e_6$ : Barack Obama  $r_4$ : make a visit

Figure 1: A typical multi-hop temporal complex query and its computation graph: "During François Hollande was the president of France, which countries did Xi Jinping visit but Barack Obama did not visit?". In the computation graph, there are entity set (blue circle), timestamp set (green triangle), time set projection (green arrow), entity set projection (blue arrow) and logical operators (red rectangle).

on queries over static KGs. These methods can neither handle an entity query involving temporal information and operators, nor answer the timestamp set for a temporal query.

Temporal knowledge graph (TKG) augments triples in static knowledge graphs with temporal information, represented as <**head, relation, tail, timestamp**> named fact. For example, the fact <**Angela Merkel, make a visit, China, 2010-07-16**> specifies the time when the event happens. Generally speaking, TKGs are more close to real world than static KGs, because most knowledge needs to be updated with time, and static KGs cannot express this change. In recommendation systems, TKGs are used to model user behaviors, which includes liking, buying, reading, commenting and so on. In financial applications, TKGs involve stock holding behaviors, trading behaviors, and financial events. These applications require reasoning over TKGs to answer complex queries. However, recent researches in TKGs focus on temporal link prediction, which is simply single-hop. A complex logical query involving multiple facts for multi-hop reasoning is not fully explored yet.

To fill the gap, we introduce a temporal multi-hop logical reasoning task over TKGs. The task aims to answer temporal complex queries, which have two main distinctions from existing queries over static KGs. Firstly, the answer sets for queries over TKGs are either entity sets or timestamp sets, while that for existing queries over static KGs can only be entity sets. Secondly, as temporal information is included in the query, temporal operators such as **After**, **Before** should be considered apart from FOL operators. To understand this new task, we provide the definition of temporal complex query in Section 3. In addition, an example of a temporal complex query is shown in Figure 1. This example query pertains to the financial event of China's visit and may be of interest to financial analysts.

According to the definition of the temporal complex query, we then generate datasets of temporal complex queries on three popular TKGs, and propose the first temporal complex query embedding framework, Temporal Feature-Logic Embedding framework (TFLEX) to answer these queries. In our framework, embeddings of objects (entity, query, timestamp) are divided into two parts, the entity part, and the timestamp part. The entity part handles the FOL operations over entities, while the timestamp part copes with the temporal operations over timestamps. Each part is further divided into feature and logic components. On the one hand, the computation of the logic components follows fuzzy logic, which enables our framework to handle all FOL operations. On the other hand, feature components are mingled and transformed under the guidance of logic components, thereby integrating logical information into the feature. Moreover, we extend fuzzy logic to support extra temporal operations (**After**, **Before** and **Between**) to handle temporal operations in the queries.

The contributions of our work are summarized as follows: (1) For the first time, the definition of the task of multi-hop logical reasoning over TKGs is given. (2) We propose the first multi-hop logical reasoning framework on TKGs, namely Temporal Feature-Logic Embedding framework (TFLEX),

which supports all FOL operations and extra temporal operations (**After**, **Before** and **Between**). (3) We generate three new TKG datasets for the task of multi-hop logical reasoning. Experiments on three generated datasets demonstrate the efficacy of the proposed framework. The source code of our framework and the datasets are available online [*].

## 2    Related Work

**Complex Query Embedding**. It learns embeddings of queries and entities, and the answer entities are close to queries in the embedding space. Existing methods utilize a lot of objects to create the embeddings, such as (1) probability distribution [3, 5] (2) geometric object [1, 2, 4, 6, 7] (3) fuzzy logic [8–10] and (4) others [11–13]. However, existing embedding-based methods are considered on static KGs. They cannot utilize temporal information in the TKGs, and therefore cannot handle temporal queries on a temporal KG. Firstly, static query embeddings (QEs) are built over (s, r, o) triples instead of (s, r, o, t) quartets, thus ignoring the timestamps for temporal complex reasoning. The second reason is the order property of timestamps, which is on the contrary that entities are unordered, leading to that static QEs are unable to handle **Before** and **After** temporal logic. In addition, we also notice the semantic conflict in experiments (see section 5.2) when concatenating the geometric embedding (static QE) with the fuzzy embedding (that can handle temporal logic) to promote the static QE to temporal one. Therefore, it is challenging for static QEs to utilize temporal information in the TKGs.

**Temporal Knowledge Graph Completion (TKGC)**. It aims at inferencing new facts in the TKGs. Existing TKGC methods could be categorized to (1) tensor decomposition [14–16], (2) timestamp-based transformation [17–21], (3) dynamic embedding [22–25], (4) Markov process models [26, 27], (5) autoregressive models [28–30], (6) others [31–33] and so on. Among these works, most of them only confined to the one-hop link prediction task, also known as one-hop reasoning. Some works [25, 28–30, 32] can perform multi-hop reasoning via a path consisting of connected quartets. But none of them could answer logical queries that involve multiple logical operations (conjunction, negation and disjunction). In this paper, we focus on the temporal complex query answering task, which is more challenging than TKGC task.

## 3    Definitions

**Temporal Knowledge Graph (TKG)** $G = \{\mathcal{V}, \mathcal{R}, \mathcal{T}, \mathcal{F}\}$ consists of entity set $\mathcal{V}$, relation set $\mathcal{R}$, timestamp set $\mathcal{T}$ and fact set $\mathcal{F} = \{(s, r, o, t)\} \subseteq \mathcal{V} \times \mathcal{R} \times \mathcal{V} \times \mathcal{T}$ containing subject-predicate-object-timestamp $(s, r, o, t)$ quartets. Without loss of generality, $G$ is a first-order logic knowledge base, where each quartet $(s, r, o, t)$ denotes an atomic formula $r(s, o, t)$, with $r$ a predicate and $s, o, t$ its arguments.

**Multi-hop Logical Reasoning over TKG** is the task to answer Temporal Complex Query $q$ when given a TKG $G$. We focus on Existential Positive First-Order (EPFO) query [34] over TKG, namely **Temporal Complex Query** $q$, which is categorized into entity query and timestamp query. Formally, the query $q$ consists of a target variable $A$, a non-variable anchor entity set $V_a \subseteq \mathcal{V}$, a non-variable anchor timestamp set $T_a \subseteq \mathcal{T}$, bound variables $V_1, \cdots, V_k$ and $T_1, \cdots, T_l$, logical operations (existential quantification $\exists$, conjunction $\wedge$, disjunction $\vee$, identity 1, negation $\neg$), and extra temporal operations (**After**, **Before**). **After**$(t_1, t_2)$ means $t_2$ is after $t_1$, while **Before**$(t_1, t_2)$ means $t_2$ is before $t_1$. Inspired by [2, 3], the disjunctive normal form (DNF) of temporal query $q$ is defined as:

$$q[A] = A \quad : \exists V_1, \cdots, V_k, T_1, \cdots, T_l : (e_1^1 \wedge \cdots \wedge e_{n_1}^1) \vee \cdots \vee (e_1^m \wedge \cdots \wedge e_{n_m}^m)$$
$$\text{where} \quad e = f \circ r(V_s, V_o \text{ or } A, T) \text{ or } f \circ r(V_s \text{ or } A, V_o, T) \text{ or } g(T_i, T_j) \text{ if } q \text{ is entity query,}$$
$$e = f \circ r(V_s, V_o, T \text{ or } A) \text{ or } g(T_i, T_j) \text{ or } g(T, A) \text{ or } g(A, T) \text{ if } q \text{ is timestamp query}$$
$$\text{with} \quad V_s, V_o \in V_a \cup \{V_1, \cdots, V_k\}, \quad T, T_i, T_j \in T_a \cup \{T_1, \cdots, T_l\},$$
$$r \in \mathcal{R}, f \in \{1, \neg\}, g \in \{\textbf{After}, \textbf{Before}\}$$

In the equation, the DNF is a disjunction of $m$ conjunctions, where $e_1^j \wedge \cdots \wedge e_{n_j}^j$ denotes a conjunction between $n_j$ logical atoms, and each $e_i^j$ denotes a logical atom. We ignore indices in the definition

---

[*] https://github.com/LinXueyuanStdio/TFLEX

of $e_i^j$ to keep the formula clean. The goal of answering the query $q$ is to find the set of entities (or timestamps) $[\![q]\!]$ that satisfy the query, such that $A \in [\![q]\!]$ iff $q[A]$ holds true, where $[\![q]\!]$ is the answer set of the query $q$.

Following existing static query embedding works, we introduce **Computation Graph**, which is a directed acyclic graph (DAG) representing the structure of temporal complex query. Its nodes represent entity/timestamp sets $S \subseteq V_a \cup V \cup T_a \cup T$, while directed edges represent logical or relational operations acting on these sets. A computation graph specifies how the reasoning of the query has proceeded on the TKG. Starting from anchor sets, we obtain the answer set after applying operations iteratively on non-answer sets according to the directed edges in the computation graph. The edge types on the computation graph are defined as follows:

1. Relational Projection $\mathcal{P}$. Given an entity set $S_1 \subseteq \mathcal{V}$, a timestamp set $S_2 \subseteq \mathcal{T}$ (or an entity set $S_2 \subseteq \mathcal{V}$ for entity projection) and a relation $r \in \mathcal{R}$, projection operation maps $S_1$ and $S_2$ to another set: $S' = \begin{cases} \cup_{(v \in S_1, t \in S_2)}\{v'|(v, r, v', t) \in \mathcal{F}\}, & \mathcal{P} \text{ is entity projection} \\ \cup_{(v \in S_1, v' \in S_2)}\{t|(v, r, v', t) \in \mathcal{F}\}, & \mathcal{P} \text{ is timestamp projection} \end{cases}$

2. Intersection $\mathcal{I}$ (Union $\mathcal{U}$, etc.). Given entity sets or timestamp sets $\{S_1, \cdots, S_n\}$, the intersection (union, etc.) operation computes logical intersection (union, etc.) of these sets $\cap_{i=1}^n S_i$ ($\cup_{i=1}^n S_i$, etc.).

3. Complement/Negation $\mathcal{C}$. The complement set of a given set $S$ is $\bar{S} = \begin{cases} \mathcal{V} - S, & S \subseteq \mathcal{V} \\ \mathcal{T} - S, & S \subseteq \mathcal{T} \end{cases}$

4. Extended temporal operators $f$. Given a timestamp set $S$, extended operators compute a certain set of timestamps $S'$: $S' = \begin{cases} \{t'|\text{for some } t' \in \mathcal{T}, t' > \max(S)\}, & f \text{ is After} \\ \{t'|\text{for some } t' \in \mathcal{T}, t' < \min(S)\}, & f \text{ is Before} \end{cases}$

In order to efficiently compute the answer set of a temporal complex query, we consider embedding the query set into a low-dimensional vector space, where the answer set is also represented by a continuous embedding vector. Formally, the **Temporal Query Embedding** $\mathbf{V}_q$ of a query $q$ is a continuous embedding vector, generated by executing operations according to the computation graph, starting from the temporal embeddings of anchor entity or timestamp sets. The **Temporal Query Answer** to the query $q$ is the entity $v$ (or timestamp $t$) whose embedding $\mathbf{v}$ (or $\mathbf{t}$) has the smallest distance $dist(\mathbf{v}, \mathbf{V}_q)$ (or $dist(\mathbf{t}, \mathbf{V}_q)$) to the embedding of query $q$.

## 4 Method

In this section, we replace the variables in the query formulation with temporal feature-logic embeddings, and perform logical operations via neural networks. We first introduce the temporal feature-logic embedding for entities, timestamps, and queries in Section 4.1. Afterwards, we introduce logical operators in Section 4.2 and how to train the model in Section 4.3.

### 4.1 Temporal Feature-Logic Embeddings for Queries and Entities

In this section, we design temporal embeddings for queries, entities and timestamps. In general, the answers to queries may be entities or timestamps. Therefore, we consider a part of the embedding as an entity part to answer entities, while the other is the timestamp part to answer timestamps. Formally, the embedding of query set $[\![q]\!]$ is $\mathbf{V}_q = (\boldsymbol{q}_f^e, \boldsymbol{q}_l^e, \boldsymbol{q}_f^t, \boldsymbol{q}_l^t)$ where $\boldsymbol{q}_f^e \in \mathbb{R}^d$ is entity feature, $\boldsymbol{q}_l^e \in [0, 1]^d$ is entity logic, $\boldsymbol{q}_f^t \in \mathbb{R}^d$ is time feature, $\boldsymbol{q}_l^t \in [0, 1]^d$ is time logic respectively, $d$ is the embedding dimension. The parameter $\boldsymbol{q}_l$ is the uncertainty $\boldsymbol{q}_l \vec{s} + (1 - \boldsymbol{q}_l)\vec{n}$ of the feature, according to fuzzy logic. An entity $v \in \mathcal{V}$ is a special query without uncertainty. We propose to represent an entity as the query with logic part $\mathbf{0}$, which indicates that the entity's uncertainty is $0$. Formally, the embedding of entity $v$ is $\mathbf{v} = (\boldsymbol{v}_f^e, \mathbf{0}, \mathbf{0}, \mathbf{0})$, where $\boldsymbol{v}_f^e \in \mathbb{R}^d$ is the entity feature part and $\mathbf{0}$ is a $d$-dimensional vector with all elements being $0$. Similarly, the embedding of timestamp $t$ is $\mathbf{t} = (\mathbf{0}, \mathbf{0}, \boldsymbol{t}_f^t, \mathbf{0})$ with entity part and time logic being $\mathbf{0}$.

Attention that we introduce vector logic, which is a type of fuzzy logic over vector space, to cope with logical transformation in the vector space. Fuzzy logic is a generalization of Boolean logic, such that the truth value of a logical atom is a real number in $[0, 1]$. In comparison, as a generalization

of a real number, the truth value in vector logic is a vector $[0, 1]^d$ in the semantic space. We denote the logical operations in vector logic as $\mathbf{AND}(\wedge), \mathbf{OR}(\vee), \mathbf{NOT}(\neg)$, and so on, which receive one or multiple vectors and output one vector as answer. For more details about fuzzy logic, please refer to Appendix A.1.

## 4.2 Logical Operators for Temporal Feature-Logic Embeddings

In this section, we introduce the designed neural logical operators, including projection, intersection, complement, and all other dyadic operators.

**Projection Operator** $\mathcal{P}_e$ **and** $\mathcal{P}_t$. The goal of operator $\mathcal{P}_e$ is to map an entity set to another entity set under a given relation and a given timestamp, while operator $\mathcal{P}_t$ outputting timestamp set given relation and two entity queries. We define a function $\mathcal{P}_e : \mathbf{V}_q, \mathbf{r}, \mathbf{V}_t \mapsto \mathbf{V}_q'$ in the embedding space to represent $\mathcal{P}_e$, and $\mathcal{P}_t : \mathbf{V}_{q_1}, \mathbf{r}, \mathbf{V}_{q_2} \mapsto \mathbf{V}_q'$ for $\mathcal{P}_t$ respectively. To implement $P_e$ and $P_t$, we first represent relations as translations on query embeddings and assign each relation with relational embedding $\mathbf{r} = (\boldsymbol{r}_f^e, \boldsymbol{r}_l^e, \boldsymbol{r}_f^t, \boldsymbol{r}_l^t)$. We follow the assumption of translation-based methods: $q_o \approx q_s + r + t$. As a comparison, static KGE TransE [35] has $o \approx s + r$, and temporal KGE TTransE [36] has $o_t \approx s_t + r$. The addition represents a semantic translation starting from the source entity set, following the relation and timestamp conditioning, ending at the target entity set. Therefore, we define $\mathcal{P}_e$ and $\mathcal{P}_t$ as:

$$
\begin{aligned}
\mathcal{P}_e(\mathbf{V}_q, \mathbf{r}, \mathbf{V}_t) &= g(\mathbf{MLP}_0^e(\mathbf{V}_q + \mathbf{r} + \mathbf{V}_t)) \\
\mathcal{P}_t(\mathbf{V}_{q_1}, \mathbf{r}, \mathbf{V}_{q_2}) &= g(\mathbf{MLP}_0^t(\mathbf{V}_{q_1} + \mathbf{r} + \mathbf{V}_{q_2}))
\end{aligned}
\tag{1}
$$

where $\mathbf{MLP} : \mathbb{R}^{4d} \to \mathbb{R}^{4d}$ is a multi-layer perception network (MLP), $+$ is element-wise addition and $g$ is an activate function to generate $\boldsymbol{q}_l^e \in [0, 1]^d$, $\boldsymbol{q}_l^t \in [0, 1]^d$. We use MLP and activation function $g(.)$ to make projection operator output a valid query embedding, which shows the closure property of the operator. $\mathcal{P}_e$ and $\mathcal{P}_t$ do not share parameters so the MLPs are different. We define $g$ as:

$$
g(\mathbf{x}) = [\mathbf{x}[0:d]; \sigma(\mathbf{x}[d:2d]); \mathbf{x}[2d:3d]; \sigma(\mathbf{x}[3d:4d])]
\tag{2}
$$

where $\mathbf{x}[0:d]$ is the slice containing element $x_i$ of vector $\mathbf{x}$ with index $0 \le i < d$, $\sigma(\cdot)$ is Sigmoid function and $[\cdot; \cdot]$ is the concatenation operator.

**Dyadic Operators**. There are two types of dyadic operators for our framework. One for entity set and the other for timestamp set. Each type includes intersection (AND), union (OR), the symmetric difference (XOR), etc. With the help of fuzzy logic, our framework can model all dyadic operators directly. Below we take a unified way to build these operators.

We start from intersection operators $\mathcal{I}_e$ (on entity set) and $\mathcal{I}_t$ (on timestamp set). The goal of intersection operator $\mathcal{I}_e$ ($\mathcal{I}_t$) is to represent $[\![q]\!] = \cap_{i=1}^n [\![q_i]\!]$ based on their entity parts (timestamp parts). Suppose that $\mathbf{V}_{q_i} = (\boldsymbol{q}_{i,f}^e, \boldsymbol{q}_{i,l}^e, \boldsymbol{q}_{i,f}^t, \boldsymbol{q}_{i,l}^t)$ is temporal feature-logic embedding for $[\![q_i]\!]$. We notice that there exists **Alignment Rule** in the process of reasoning. When performing entity set intersection $\mathcal{I}_e$, we should also perform intersection on timestamp parts in order to align the entities into the same time set. The same also holds for timestamp set intersection $\mathcal{I}_t$ and all other dyadic operators. Therefore, we firstly define the intersection operators as follows:

Fuzzy Logic  Alignment Rule

$$
\mathcal{I}_e(\mathbf{V}_{q_1}, \cdots, \mathbf{V}_{q_n}) = \left( \sum_{i=1}^n \boldsymbol{\alpha}_i^e \boldsymbol{q}_{i,f}^e, \; \underset{i=1}{\overset{n}{\mathbf{AND}}}\{\boldsymbol{q}_{i,l}^e\}, \; \sum_{i=1}^n \boldsymbol{\beta}_i^e \boldsymbol{q}_{i,f}^t, \; \underset{i=1}{\overset{n}{\mathbf{AND}}}\{\boldsymbol{q}_{i,l}^t\} \right)
$$

$$
\mathcal{I}_t(\mathbf{V}_{q_1}, \cdots, \mathbf{V}_{q_n}) = \left( \sum_{i=1}^n \boldsymbol{\alpha}_i^t \boldsymbol{q}_{i,f}^e, \; \underset{i=1}{\overset{n}{\mathbf{AND}}}\{\boldsymbol{q}_{i,l}^e\}, \; \sum_{i=1}^n \boldsymbol{\beta}_i^t \boldsymbol{q}_{i,f}^t, \; \underset{i=1}{\overset{n}{\mathbf{AND}}}\{\boldsymbol{q}_{i,l}^t\} \right)
$$

Alignment Rule  Fuzzy Logic

where **AND** is the conjunction in fuzzy logic, $\boldsymbol{\alpha}_i$ and $\boldsymbol{\beta}_i$ are attention weights. To notice the changes of logic, we compute $\boldsymbol{\alpha}_i^{e,t}$ and $\boldsymbol{\beta}_i^{e,t}$ via the following attention mechanism:

$$
\boldsymbol{\alpha}_i^{e,t} = \frac{\exp(\mathbf{MLP}_1^{e,t}([\boldsymbol{q}_{i,f}^e; \boldsymbol{q}_{i,l}^e]))}{\sum_{j=1}^n \exp(\mathbf{MLP}_1^{e,t}([\boldsymbol{q}_{j,f}^e; \boldsymbol{q}_{j,l}^e]))}, \qquad \boldsymbol{\beta}_i^{e,t} = \frac{\exp(\mathbf{MLP}_2^{e,t}([\boldsymbol{q}_{i,f}^t; \boldsymbol{q}_{i,l}^t]))}{\sum_{j=1}^n \exp(\mathbf{MLP}_2^{e,t}([\boldsymbol{q}_{j,f}^t; \boldsymbol{q}_{j,l}^t]))}
\tag{3}
$$

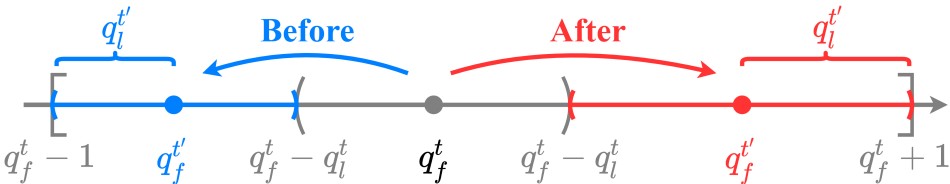

Figure 2: The computation of time part in temporal operators Before and After.

where $\mathbf{MLP}_{1,2}^{e,t} : \mathbb{R}^{2d} \to \mathbb{R}^d$ are MLP networks, $[\cdot; \cdot]$ is concatenation. The first self-attention neural network will learn the hidden information from entity logic and leverage to entity feature, while the second one gathers logical information from time logic to time feature. Note that the computation of entity logic, and time logic obeys the law of fuzzy logic, without any extra learnable parameters. In this way, all dyadic operators can be generated from fuzzy logic in our framework. Due to space limitation, we present the union, exclusive or, implication operators and so on in Appendix A.3.

**Complement Operators:** $\mathcal{C}_e$ and $\mathcal{C}_t$. The aim of $\mathcal{C}_e$ is to identify the complement of query set $\llbracket q \rrbracket$ such that $\llbracket \neg q \rrbracket = \mathcal{V} \backslash \llbracket q \rrbracket$, while $\mathcal{C}_t$ aims to calculate the complement $\llbracket \neg q \rrbracket = \mathcal{T} \backslash \llbracket q \rrbracket$ by the time parts. Suppose that $\mathbf{V}_q = (\boldsymbol{q}_f^e, \boldsymbol{q}_l^e, \boldsymbol{q}_f^t, \boldsymbol{q}_l^t)$, we define the complement operator $\mathcal{C}_e$ and $\mathcal{C}_t$ as:

$$\mathcal{C}_e(\mathbf{V}_q) = (f_{\text{not}}^e(\boldsymbol{q}_f^e), \mathbf{NOT}(\boldsymbol{q}_l^e), \boldsymbol{q}_f^t, \boldsymbol{q}_l^t), \qquad \mathcal{C}_t(\mathbf{V}_q) = (\boldsymbol{q}_f^e, \boldsymbol{q}_l^e, f_{\text{not}}^t(\boldsymbol{q}_f^t), \mathbf{NOT}(\boldsymbol{q}_l^t)) \qquad (4)$$

where $f_{\text{not}}^e(\boldsymbol{q}_f) = \tanh(\mathbf{MLP}_3([\boldsymbol{q}_f^e; \boldsymbol{q}_l^e])), f_{\text{not}}^t(\boldsymbol{q}_f^t) = \tanh(\mathbf{MLP}_4([\boldsymbol{q}_f^t; \boldsymbol{q}_l^t]))$ are feature negation functions, two $\mathbf{MLP}_{3,4} : \mathbb{R}^{2d} \to \mathbb{R}^d$ are MLP networks, $\mathbf{NOT}$ is negation in fuzzy logic.

**Temporal Operators: After** $\mathcal{A}_t$, **Before** $\mathcal{B}_t$ **and Between** $\mathcal{D}_t$. The operator After $\mathcal{A}_t : \mathbf{V}_q \mapsto \mathbf{V}_q'$ (Before $\mathcal{B}_t : \mathbf{V}_q \mapsto \mathbf{V}_q'$) aims to deduce the timestamps after(before) a given fuzzy time set $\llbracket q \rrbracket$. Let $\mathbf{V}_q = (\boldsymbol{q}_f^e, \boldsymbol{q}_l^e, \boldsymbol{q}_f^t, \boldsymbol{q}_l^t)$, we define $\mathcal{A}_t$ and $\mathcal{B}_t$ as:

$$\mathcal{A}_t(\mathbf{V}_q) = (\boldsymbol{q}_f^e, \boldsymbol{q}_l^e, \boldsymbol{q}_f^t + \frac{1 + \boldsymbol{q}_l^t}{2}, \frac{1 - \boldsymbol{q}_l^t}{2}), \qquad \mathcal{B}_t(\mathbf{V}_q) = (\boldsymbol{q}_f^e, \boldsymbol{q}_l^e, \boldsymbol{q}_f^t - \frac{1 + \boldsymbol{q}_l^t}{2}, \frac{1 - \boldsymbol{q}_l^t}{2}) \qquad (5)$$

The entity part does not change after computation because temporal operator only affects the time part (time feature $\boldsymbol{q}_f^t$ and time logic $\boldsymbol{q}_l^t$). The motivation of computation is illustrated in Figure 2. Since $\boldsymbol{q}_l^t$ is the uncertainty of time feature $\boldsymbol{q}_f^t$, the time part can be viewed as an interval $[\boldsymbol{q}_f^t - \boldsymbol{q}_l^t, \boldsymbol{q}_f^t + \boldsymbol{q}_l^t]$ whose center is $\boldsymbol{q}_f^t$ and half-length is $\boldsymbol{q}_l^t$. The interval is covered by $[\boldsymbol{q}_f^t - 1, \boldsymbol{q}_f^t + 1]$ because the probability $\boldsymbol{q}_l^t < 1$. Then, after interval $[\boldsymbol{q}_f^t - \boldsymbol{q}_l^t, \boldsymbol{q}_f^t + \boldsymbol{q}_l^t]$ is the interval $[\boldsymbol{q}_f^t + \boldsymbol{q}_l^t, \boldsymbol{q}_f^t + 1]$ whose center is $\boldsymbol{q}_f^t + \frac{1 + \boldsymbol{q}_l^t}{2}$ and half-length is $\frac{1 - \boldsymbol{q}_l^t}{2}$, which gives the time part of embedding $\mathcal{A}_t(\mathbf{V}_q)$. Similarly, the time part of embedding $\mathcal{B}_t(\mathbf{V}_q)$ is $\boldsymbol{q}_f^t - \frac{1 + \boldsymbol{q}_l^t}{2}$ (time feature) and $\frac{1 - \boldsymbol{q}_l^t}{2}$ (time logic), which are generated from $[\boldsymbol{q}_f^t - 1, \boldsymbol{q}_f^t - \boldsymbol{q}_l^t]$ before $[\boldsymbol{q}_f^t - \boldsymbol{q}_l^t, \boldsymbol{q}_f^t + \boldsymbol{q}_l^t]$. The operator Between $\mathcal{D}_t : \mathbf{V}_{q_1}, \mathbf{V}_{q_2} \mapsto \mathbf{V}_q'$ inferences the time set after $\llbracket q_1 \rrbracket$ and before $\llbracket q_2 \rrbracket$. Therefore, we define Between $\mathcal{D}_t$ as $\mathcal{D}_t(\mathbf{V}_{q_1}, \mathbf{V}_{q_2}) = \mathcal{I}_t(\mathcal{A}_t(\mathbf{V}_{q_1}), \mathcal{B}_t(\mathbf{V}_{q_2}))$ to output the time between $\llbracket q_1 \rrbracket$ and $\llbracket q_2 \rrbracket$.

### 4.3 Learning Temporal Feature-Logic Embeddings

We expect the model to achieve high scores for the answers to the given query $q$, and low scores for $v' \notin \llbracket q \rrbracket$. Therefore, we firstly define a distance function to measure the distance between a given answer embedding and a query embedding, and then we train the model with negative sampling loss.

**Distance Function** Given an entity embedding $\mathbf{v} = (\boldsymbol{v}_f^e, \mathbf{0}, \mathbf{0}, \mathbf{0})$, a timestamp embedding $\mathbf{t} = (\mathbf{0}, \mathbf{0}, \boldsymbol{t}_f^t, \mathbf{0})$ and a query embedding $\mathbf{V}_q = (\boldsymbol{q}_f^e, \boldsymbol{q}_l^e, \boldsymbol{q}_f^t, \boldsymbol{q}_l^t)$, we define the distance $d$ between the answer and the query $q$ as the sum of the feature distance (between the feature parts) and the logic part (to expect uncertainty to be 0). If the query answers entities, the distance is $d^e(\mathbf{v}; \mathbf{V}_q) = \|\boldsymbol{v}_f^e - \boldsymbol{q}_f^e\|_1 + \boldsymbol{q}_l^e$. Otherwise, the distance is $d^t(\mathbf{t}; \mathbf{V}_q) = \|\boldsymbol{t}_f^t - \boldsymbol{q}_f^t\|_1 + \boldsymbol{q}_l^t$ for queries answering timestamp set, where $\| \cdot \|_1$ is the $L_1$ norm and $+$ is element-wise addition. The distance function aims to optimize two losses. One is to push the answers to the neighbor of query in the embedding

space. It corresponds to the term L1 distance between answer and query. The other is to reduce the uncertainty of the query (the probability interpretation of the logic part), to make the answers more accurate.

**Loss Function** Given a training set of queries, we optimize a negative sampling loss

$$L = -\log \sigma(\gamma - d(\mathbf{v}; \mathbf{V}_q)) - \frac{1}{k} \sum_{i=1}^{k} \log \sigma(d(\mathbf{v}_i'; \mathbf{V}_q) - \gamma) \tag{6}$$

where $\gamma > 0$ is a fixed margin, $k$ is the number of negative entities, and $\sigma(\cdot)$ is the sigmoid function. When query $q$ is answering entities (timestamps), $v \in [\![q]\!]$ is a positive entity (timestamp), $v_i' \notin [\![q]\!]$ is the $i$-th negative entity (timestamp).

## 5  Experiments

In this section, we evaluate the ability of TFLEX to reason over TKGs. We first introduce experimental settings in Section 5.1, and then present the experimental results in Section 5.2.

### 5.1  Experimental Settings

**Datasets and Query Generation**   Experiments are performed on three new datasets generated from standard benchmarks for TKGC: ICEWS14 [37], ICEWS05-15 [37], and GDELT-500 [38] (with statistics in Appendix B.1). We predefined 40 kinds of complex queries for each dataset. The definition of the 40 kinds of complex queries and the query generation process details are described in Appendix B.2. We consider 27 kinds of queries for training and all 40 kinds for evaluation and testing. Please refer to Appendix B.3 for summaries of the final datasets.

To briefly summarize the results, we aggregate groups of queries that can be answered: entities ($\text{avg}_e$), timestamps ($\text{avg}_t$), entities with negation ($\text{avg}_{e,\mathcal{C}_e}$), timestamps with negation ($\text{avg}_{t,\mathcal{C}_t}$), entities with unseen union ($\text{avg}_{\{\mathcal{U}_e\}}$), timestamps with unseen union ($\text{avg}_{\{\mathcal{U}_t\}}$), and other hybrid unseen structures ($\text{avg}_x$). These groups are inspired by the experiment settings of existing static QEs [1–4]. The detail that which query belongs to which group will be shown in Appendix B.7. Note that the training set only involves 4 groups of queries: $\text{avg}_e$, $\text{avg}_t$, $\text{avg}_{e,\mathcal{C}_e}$, and $\text{avg}_{t,\mathcal{C}_t}$.

**Evaluation**   Given a test query $q$, for each non-trivial answer $v \in [\![q]\!]_{\text{test}} - [\![q]\!]_{\text{valid}}$ of the query $q$, we rank it against non-answer entities $\mathcal{V} - [\![q]\!]_{\text{test}}$ (or non-answer timestamps $\mathcal{T} - [\![q]\!]_{\text{test}}$ if the query is answering timestamps). Then we calculate Mean Reciprocal Rank (MRR) based on the rank. The higher, the better. Please refer to Appendix B.5 for the definition of MRR.

### 5.2  Main Results

For each group, we report the average MRR in Table 1. The raw MRR results on all query structures for each dataset in detail are presented in Appendix B.7.

**How well can TFLEX answer temporal complex queries?**. We compare TFLEX with state-of-the-art query embeddings Query2box [2], BetaE [3] and ConE [4] on answering entities. Existing query embeddings only handle FOL on entity set, but unable to cope with temporal logic over timestamp set. Therefore, the results of these three methods have to be obtained by ignoring the timestamps, so that the $\text{avg}_t$, $\text{avg}_{t,\mathcal{C}_t}$, $\text{avg}_{\{\mathcal{U}_t\}}$ and $\text{avg}_x$ of three methods are zeros. Comparing the results of these methods in Table 1, we can see that TFLEX outperforms all the baselines on all the metrics. These results demonstrate that TFLEX can perform reasoning over TKGs well, at least better than the existing query embeddings.

**Ablation on entity part**. The variant **X(ConE)** replaces the entity part of TFLEX with ConE [4] to handle the logic over entity sets. In the entity part, ConE is geometric while TFLEX is fuzzy. The variant performs even worse than TFLEX and ConE. The dropping MRR indicates that the entity part plays an important role in the framework. Considering the performance of ConE, we think there is semantic conflict between the time part of TFLEX and the entity part of ConE. Simply combining static QEs with dynamic QEs is not a clever way to achieve the best performance.

Table 1: Average MRR results for different groups of temporal complex queries. **X** denotes the variant of TFLEX. **X(ConE)** replaces the entity part with ConE [4]. **FLEX** ablates the time part. **X-1F** merges entity feature and timestamp feature into one feature. **X-logic** removes the logic part.

| Dataset | Metrics | Query2box | BetaE | ConE | TFLEX | X(ConE) | FLEX | X-1F | X-logic |
|---|---|---|---|---|---|---|---|---|---|
| ICEWS14 | $\text{avg}_e$ | 25.06 | 37.19 | 41.94 | 56.79 | 40.93 | 43.67 | 56.89 | 56.64 |
| | $\text{avg}_{e,\mathcal{C}_e}$ | | 36.69 | 44.88 | 50.82 | 42.15 | 44.41 | 49.78 | 51.17 |
| | $\text{avg}_t$ | | | | 17.56 | 16.41 | | 18.77 | 18.03 |
| | $\text{avg}_{t,\mathcal{C}_t}$ | | | | 36.37 | 35.24 | | 37.73 | 36.39 |
| | $\text{avg}_{\{\mathcal{U}_e\}}$ | | 19.95 | 26.47 | 35.74 | 25.46 | 29.25 | 34.48 | 34.68 |
| | $\text{avg}_{\{\mathcal{U}_t\}}$ | | | | 26.24 | 24.07 | | 28.04 | 26.36 |
| | $\text{avg}_x$ | | | | 28.03 | 26.65 | | 29.31 | 28.61 |
| | **AVG** | | | | 35.93 | 30.13 | | 36.43 | 35.98 |
| ICEWS05-15 | $\text{avg}_e$ | 24.00 | 31.33 | 40.93 | 48.99 | 36.29 | 38.96 | 49.90 | 44.80 |
| | $\text{avg}_{e,\mathcal{C}_e}$ | | 29.70 | 43.52 | 46.17 | 38.12 | 42.10 | 46.11 | 41.92 |
| | $\text{avg}_t$ | | | | 4.39 | 4.41 | | 4.43 | 3.29 |
| | $\text{avg}_{t,\mathcal{C}_t}$ | | | | 30.16 | 29.49 | | 30.26 | 28.34 |
| | $\text{avg}_{\{\mathcal{U}_e\}}$ | | 21.54 | 43.02 | 54.37 | 36.37 | 44.38 | 54.05 | 45.36 |
| | $\text{avg}_{\{\mathcal{U}_t\}}$ | | | | 28.69 | 26.40 | | 27.70 | 23.39 |
| | $\text{avg}_x$ | | | | 24.26 | 21.69 | | 24.41 | 21.95 |
| | **AVG** | | | | 33.72 | 27.54 | | 33.98 | 29.86 |
| GDELT-500 | $\text{avg}_e$ | 9.67 | 14.75 | 18.44 | 19.60 | 17.83 | 19.07 | 17.92 | 17.36 |
| | $\text{avg}_{e,\mathcal{C}_e}$ | | 11.15 | 12.67 | 13.52 | 12.34 | 13.35 | 12.13 | 12.11 |
| | $\text{avg}_t$ | | | | 5.38 | 3.16 | | 5.49 | 5.75 |
| | $\text{avg}_{t,\mathcal{C}_t}$ | | | | 6.31 | 3.93 | | 6.50 | 6.86 |
| | $\text{avg}_{\{\mathcal{U}_e\}}$ | | 6.20 | 6.96 | 7.58 | 7.41 | 7.44 | 6.92 | 6.91 |
| | $\text{avg}_{\{\mathcal{U}_t\}}$ | | | | 6.71 | 6.35 | | 6.59 | 6.80 |
| | $\text{avg}_x$ | | | | 6.17 | 6.17 | | 6.47 | 6.64 |
| | **AVG** | | | | 9.32 | 8.17 | | 8.86 | 8.92 |

**Ablation on time part**. The variant **FLEX** removes the time part of the temporal feature-logic embedding. Then, **FLEX** can only answer entities. The results show that **FLEX** slightly outperforms ConE. However, the performance of **FLEX** is worse than TFLEX with a large margin on all the datasets. Therefore, we conclude that the time part also plays an important role in the framework.

**Ablation on feature part**. If we remove the entity and timestamp feature, the embeddings of entities and timestamps will crash to zeros. Instead, we consider another way to explore the impact of the feature part. Noticing that some TKGC approaches [36, 39] embed entities and timestamps into the same semantic space, we propose **X-1F** to merge the entity and timestamp features into one feature. Compared with TFLEX, **X-1F** achieves higher scores on ICEWS14 and ICEWS05-15, but lower on GDELT. The results imply that unifying the feature of entity and timestamp is potentially beneficial in some datasets.

**Ablation on logic part**. The variant **X-logic** removes both entity logic and time logic. It achieves lower scores than TFLEX on all the datasets. This is because the logic part is responsible for reasoning over TKGs. Removing the logic part results in that neural logical operators completely rely on neural network to learn the logic, which is not enough to handle various temporal complex queries.

**Out-of-data reasoning**. The results on $\text{avg}_{\{\mathcal{U}_e\}}, \text{avg}_{\{\mathcal{U}_t\}}, \text{avg}_x$ support that the framework can reason over unseen entity logic, unseen time logic as well as their hybrid. The entity union and time union operators are not included in the training set, but the framework can still handle them well.

**Sensitive analysis**. (1) The selection of hyperparameters (embedding dimension $d$, the margin $\gamma$) has a substantial influence on the effectiveness of TFLEX. We train the model with embedding dimension $d \in \{300, 400, 500, 600, 700, 800, 900, 1000\}$, the margin $\gamma \in \{5, 10, 15, 20, 25, 30, 35, 40\}$. The best performance is achieved when $d = 800$ and $\gamma = 15$. Too small and too large $d, \gamma$ both lead to worse results, reported in Appendix B.6. (2) Besides, we also investigate the stability of TFLEX. We train and test for five times with different random seeds and report the error bars in Appendix B.6. The small standard variances demonstrate that the performance of TFLEX is stable.

**Comparison between TFLEX and TKGC methods**. It's natural to compare TFLEX with SOTA TKGC methods, since all of them can answer one-hop temporal queries. We present the comparison results in Table 2. We can see that TFLEX is competitive with translation-based methods

Table 2: TKGC Results (%) on ICEWS14, ICEWS05-15, and GDELT. The results from top to bottom are organized as static KGEs, timestamp-based transformation TKGEs, tensor decomposition, autoregressive models and ours. Best results are in bold. †, ⋆ indicate the results taken from [40, 17]. Other results are the best numbers reported in their respective paper.

| Model | ICEWS14 | | | | ICEWS05-15 | | | | GDELT | | | |
|---|---|---|---|---|---|---|---|---|---|---|---|---|
| | MRR | Hit@1 | Hit@3 | Hit@10 | MRR | Hit@1 | Hit@3 | Hit@10 | MRR | Hit@1 | Hit@3 | Hit@10 |
| TransE⋆ [35] | 28.0 | 9.4 | - | 63.7 | 29.4 | 9.0 | - | 66.3 | 11.3 | 0.0 | 15.8 | 31.2 |
| DistMult⋆ [41] | 43.9 | 32.3 | - | 67.2 | 45.6 | 33.7 | - | 69.1 | 19.6 | 11.7 | 20.8 | 34.8 |
| SimplE⋆ [42] | 45.8 | 34.1 | 51.6 | 68.7 | 47.8 | 35.9 | 53.9 | 70.8 | 20.6 | 12.4 | 22.0 | 36.6 |
| ConT [43] | 18.5 | 11.7 | 20.5 | 31.5 | 16.3 | 10.5 | 18.9 | 27.2 | 14.4 | 8.0 | 15.6 | 26.5 |
| TTransE [36] | 25.5 | 7.4 | - | 60.1 | 27.1 | 8.4 | - | 61.6 | 11.5 | 0.0 | 16.0 | 31.8 |
| HyTE [44] | 29.7 | 10.8 | 41.6 | 65.5 | 31.6 | 11.6 | 44.5 | 68.1 | 11.8 | 0.0 | 16.5 | 32.6 |
| TA-DistMult [45] | 47.7 | 36.3 | - | 68.6 | 47.4 | 34.6 | - | 72.8 | 20.6 | 12.4 | 21.9 | 36.5 |
| DE-TransE [46] | 32.6 | 12.4 | 46.7 | 68.6 | 31.4 | 10.8 | 45.3 | 68.5 | 12.6 | 0.0 | 18.1 | 35.0 |
| DE-DistMult [46] | 50.1 | 39.2 | 56.9 | 70.8 | 48.4 | 36.6 | 54.6 | 71.8 | 21.3 | 13.0 | 22.8 | 37.6 |
| DE-SimplE [46] | 52.6 | 41.8 | 59.2 | 72.5 | 51.3 | 39.2 | 57.8 | 74.8 | 23.0 | 14.1 | 24.8 | 40.3 |
| ChronoR [17] | **62.5** | **54.7** | **66.9** | **77.3** | **67.5** | **59.6** | **72.3** | **82.0** | - | - | - | - |
| TuckERT [39] | 59.4 | 51.8 | 64.0 | 73.1 | 62.7 | 55.0 | 67.4 | 76.9 | **41.1** | **31.0** | **45.3** | **61.4** |
| TuckERTNT [39] | 60.4 | 52.1 | 65.5 | 75.3 | 63.8 | 55.9 | 68.6 | 78.3 | 38.1 | 28.3 | 41.8 | 57.6 |
| RGCRN† [47, 40] | 33.3 | 24.0 | 36.5 | 51.5 | 35.9 | 26.2 | 40.0 | 54.6 | 18.6 | 11.5 | 19.8 | 32.4 |
| CyGNet† [48] | 34.6 | 25.3 | 38.8 | 53.1 | 35.4 | 25.4 | 40.2 | 54.4 | 18.0 | 11.1 | 19.1 | 31.5 |
| RE-NET† [28] | 35.7 | 25.9 | 40.1 | 54.8 | 36.8 | 26.2 | 41.8 | 57.6 | 19.6 | 12.0 | 20.5 | 33.8 |
| RE-GCN† [40] | 37.7 | 27.1 | 42.5 | 58.8 | 38.2 | 27.4 | 43.0 | 59.9 | 19.1 | 11.9 | 20.4 | 33.1 |
| TFLEX-1p | 43.9 | 31.4 | 49.6 | 64.4 | 40.6 | 29.1 | 47.5 | 66.1 | 16.5 | 8.6 | 17.3 | 33.1 |
| TFLEX | 48.2 | 35.7 | 56.5 | 72.3 | 43.0 | 30.0 | 49.8 | 69.5 | 18.5 | 10.1 | 19.6 | 34.9 |

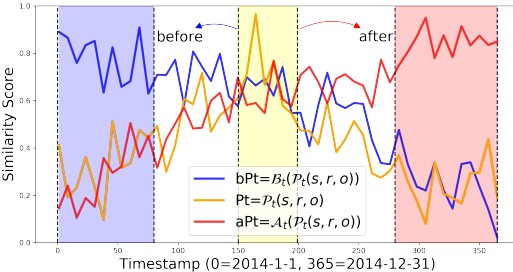

Figure 3: Score distributions of **Pt**, **bPt** and **aPt**.

Table 3: MRR of **Pe** on ICEWS14, ICEWS05-15, and GDELT-500. The best results are in bold. Results of TTransE and HyTE are taken from Goel et al. [46].

| Datasets | TTransE* | HyTE* | TFLEX-1p | TFLEX |
|---|---|---|---|---|
| **ICEWS14** | 25.5 | 29.7 | 42.9 | **48.2** |
| **ICEWS05-15** | 27.1 | 31.6 | 40.6 | **43.0** |
| **GDELT-500** | 11.5 | 11.8 | 16.5 | **18.5** |

(ConT [43], TTransE [36], HyTE [44], etc.), but it doesn't outperform the SOTA TKGC methods like ChronoR [17] and TuckERT [39]. However, the result doesn't affect the novelty and contribution of this paper. Please note that the projection operator of TFLEX is as simple as TTransE, not further optimized for TKGC tasks only. Upgrading the projection operator to outperform SOTA TKGC methods remains a future work.

**Necessity of training on temporal complex queries**. Our experiments demonstrate that training on complex queries is necessary to achieve the best performance. We compare with translation-based $\mathcal{P}_e$ operators (TTransE [36], HyTE [44] and TFLEX's variant **TFLEX-1p**) using only one-hop **Pe** queries for training. We choose TTransE and HyTE because our projection operator is also translation-based ($\mathcal{P}_e(\mathbf{V}_q, \mathbf{r}, \mathbf{V}_t) \propto \mathbf{V}_q + \mathbf{r} + \mathbf{V}_t$). From the result table 3, we observe that TFLEX achieves the best performance when comparing with these translation-based baselines on all datasets. Besides, compared with **TFLEX-1p**, TFLEX achieves 7.9% relative improvement on average on MRR, which demonstrates that training on complex queries could improve the one-hop query-answering ability.

**Effectiveness of neural temporal operators**. We found that the temporal operators $\mathcal{A}_t$ and $\mathcal{B}_t$ change the semantic of predicted timestamp embedding logically. We randomly select an event $(s, r, o)$ from ICEWS14 and consider three temporal queries **Pt**, **bPt** and **aPt**. Then, **Pt**$(= \mathcal{P}_t(s, r, o))$ predicts the date $t$ when this event happened. And **bPt**$(=\mathcal{B}_t(\mathcal{P}_t(s, r, o)))$ should predict the date before $t$, while **aPt**$(=\mathcal{A}_t(\mathcal{P}_t(s, r, o)))$ is supposed to predict the time after $t$. The output of three queries are time embeddings. Because the time embedding is fuzzy, we score it to all possible Timestamps, and visualize the normalized similarity score distribution over all days in a year, from 0 to 365 in ICEWS14. The higher the score, the more possible the predicted date is the day. We plot the three score distributions in Figure 3. For each distribution, we highlight the periods of the highest scores

with a colored background. These colored blocks represent the most likely happening time interval of the event. We observe that the order of colored blocks (corresponding to the predictions of **bPt**, **Pt**, and **aPt**) matches the logical meanings of these operators (Before the event, On the event, After the event). The visualization shows that neural temporal operators perform the time transformation correctly. More examples are attached in Appendix D.

**Explaining answers with temporal feature-logic framework**. We take query **Pe2** for example. Given temporal query **Pe2**: $q[V_?] = V_?, \exists V_a, r_1(e_1, V_a, t_1) \wedge r_2(V_a, V_?, t_2)$, let's try an example which can be written as: On 2014-04-04, who consulted the man who was appealed to or requested by the Head of Government (Latvia) on 2014-08-01? In this example, $e_1$ is "Head of Government (Latvia)", $r_1$ is "Make an appeal or request", $t_1$ is "2014-08-01", $r_2$ is "consult$^{-1}$", and $t_2$ is "2014-04-04". Then we use TFLEX to execute the query and get answers. We classify the answers into easy, hard and wrong ones. The easy answer is the correct answer that appears in the training set, while the hard answer is the correct answer that exists in the testing set instead of training set.

We present five most likely answers for interpretation in Figure 4. From the table we observe that TFLEX ranks easy answers "François Hollande", "Taavi Rõivas" and hard answer "Andris Berzins" higher than wrong answers "Angela Merkel" and "Head of Government (Latvia)". This shows that TFLEX successfully finds the hard answer by performing complex reasoning, and distinguishes the right answers from the wrong ones.

We provide 36 examples in Appendix E, including the visualization and intermediate explanation of answers for each query structure.

| | Query Sentence | On 2014-04-04, who consulted the man who was appealed to or requested by the Head of Government (Latvia) on 2014-08-01? | | |
|---|---|---|---|---|
| | Temporal Query | $q[V_?] = V_?, \exists V_a, r_1(e_1, V_a, t_1) \wedge r_2(V_a, V_?, t_2)$ | | |
| | Rank | Query Answers | Correctness | Answer Type |
| | 1 | François Hollande | ✔ | Easy |
| | 2 | Taavi Rõivas | ✔ | Easy |
| | 3 | Jyrki Katainen | ✔ | Hard |
| | 4 | Angela Merkel | ✘ | - |
| | 5 | Head of Government (Latvia) | ✘ | - |

Figure 4: Intermediate variable assignments and ranks for example Pe2 query. Correctness indicates whether the answer belongs to the ground-truth set of answers.

# 6  Conclusion

In this paper, we firstly define a temporal multi-hop logical reasoning task on temporal knowledge graphs. Then we generate three new TKG datasets and propose the Temporal Feature-Logic embedding framework, TFLEX, to handle temporal complex queries in datasets. Fuzzy logic is used to guide all FOL transformations on the logic part of embeddings. We also further extended fuzzy logic to implement extra temporal operators (**Before**, **After** and **Between**). To the best of our knowledge, TFLEX is the first framework to support multi-hop complex reasoning on temporal knowledge graphs. Experiments on benchmark datasets demonstrate the efficacy of the proposed framework in handling different operations in complex queries.

## Acknowledgments and Disclosure of Funding

This work is supported by the National Science Foundation of China (Grant No. 62176026) and Beijing Natural Science Foundation (M22009); Engineering Research Center of Information Networks, Ministry of Education. We would like to express our sincere gratitude to our corresponding authors, Prof. Haihong E and Dr. Chengjin Xu, for their guidance, expertise, and unwavering support throughout the project. Furthermore, we would like to thank Fenglong Su, Gengxian Zhou, Tianyi Hu, Ningyuan Li, Mingzhi Sun, and Haoran Luo for their individual contributions and collaboration in various aspects of this project. Their expertise and dedication have significantly enriched the final outcome. Additionally, we extend our thanks to the anonymous reviewers for their time and efforts in reviewing our work. Their constructive feedback and comments have been instrumental in improving the overall clarity and rigor of this research.

## Broader Impact

Multi-hop reasoning makes the information stored in TKGs more valuable. With the help of multi-hop reasoning, we can digest more hidden and implicit information in TKGs. It will broaden and deepen KG applications in various fields, such as question answering, recommend systems, and information retrieval. It may also bring about risks exposing unexpected personal information on public data.

Finance temporal knowledge graph is a good example to illustrate the broader impact of multi-hop reasoning. In the financial field, the information stored in TKGs is very sensitive. The one-hop reasoning can complete the hidden financial transaction, while the multi-hop reasoning can help to detect the fraud. The After operator could also be used to predict the future financial transaction. People may take the advantages of logical reasoning to digest financial factor to obtain excess returns.

Military temporal knowledge graph is another example. With the help of multi-hop logical reasoning, we may predict the future military strategy of a country. Besides, with the fuzzy completion of the hidden military information, we can also detect the hidden militarily moves, which may save thousands of soldiers' lives.

The last example is social temporal knowledge graph. The behaviors of people are left and stored in TKGs. With the help of logical reasoning, we may predict a short future of a person. For example, the query may answer the evolution of user profiles: how long may a person transfer to another role, from student to worker, becoming a parent, or being a grandparent. Tracking the user's identity change can provide super benefits for merchants' advertising. Detecting the role of a person is also helpful to provide more personalized services.

However, we should agree that the multi-hop reasoning is still at an extremely early stage, though it may bring about risks. Therefore, we should pay more attention to the security of TKGs and the privacy of users at the mean time when we explore the technology of multi-hop reasoning over TKGs.

## Limitation & Future Work

In this section, we list 4 limitations and talk about the possible future work.

**Limitation 1: The temporal operators are not enough.**. We define three extra temporal operators (**Before**, **After** and **Between**) in query generation. However, there exists more temporal operators in the real world. For example, Allen [49] defined 13 types of temporal relations represented by two intervals, including before/after, during/contains, overlaps/overlapped-by, meets/met-by, starts/started-by, finishes/finished-by and equal. In the future we would like to promote these temporal operators to TKGs.

**Limitation 2: The temporal embedding could improve.**. In this paper, the embedding of the timestamp is randomly initialized and finally learned by the model via logical advantages. Such embedding ignores the order of different timestamps. The order property is learned by the After and Before operators, which may be not enough. We recall that in the field of NLP, positional embeddings also have order features, which may be used for the construction of timestamp embeddings.

**Limitation 3: The query generation is time-consuming.**. There are 40 predefined query structures in our query generation module. Each query structure has 10k+ queries for training. With the scale of TKGs increasing, the number of queries will also increase, even 40x faster. Therefore, we need to find a more efficient way to generate queries for large scale TKGs.

**Limitation 4: The MRR and Hits@k are weak.**. The MRR and Hits@k evaluation metric may not inflect the performance of multi-hop reasoning. We observe that some queries have lots of answers. When the number of answers is larger than k, the MRR and Hits@k will be low, even if all answers are correctly ranked at top. Because the right answers that ranked after k are labeled as false. The Hits@k decreases with the increase number of right answers, which is not reasonable. This disadvantage prevents us from comparing the performance across different datasets. In this paper, GDELT is much denser, and the count of right answers is larger than the other two datasets. Therefore, the MRR of GDELT is lower than the other two datasets. It can also be verified that on the average MRR, group $avg_t$ is lower than group $avg_e$. The reason is that the number of right answers of group $avg_t$ is larger than that of group $avg_e$, which can be seen from the statistic of average answer count of queries in Table 10. In the future, there should be a more reasonable evaluation metric for multi-hop reasoning.

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

# Appendix

## A  More Details about Neural Operators

### A.1  Fuzzy Logic

**Fuzzy logic** is a generalization of Boolean logic, which is based on the idea that the truth of a statement can be expressed as a number between 0 and 1. It is a mathematical framework for reasoning with imprecise information. In fuzzy logic, there are also logical operators, such as conjunction, disjunction, negation, implication, equivalence, and exclusive or. Most popular fuzzy logic operators are defined as follows ($a, b \in [0, 1]$):

(1) **conjunction**: $a \wedge b = \min(a, b)$. **disjunction**: $a \vee b = \max(a, b)$.

(2) **negation**: $\neg a = 1 - a$. **exclusive or**: $a \oplus b = \min(a, 1 - b) + \min(1 - a, b)$.

(3) **implication**: $a \rightarrow b = \max(1 - a, b)$. **equivalence**: $a \leftrightarrow b = \min(a, b) + \max(1 - a, 1 - b)$.

However, the fuzzy logic operators are defined over real space, not suitable for reasoning over vector space. We expect that the fuzzy logic operators can be executed by matrix computation. Because the embedding space of knowledge graph is a vector space. In order to cope with tensor computation in neural networks, we introduce the fuzzy logic operators over vector space in the following, which is called vector logic.

### A.2  Vector Logic

Vector logic is an elementary logical model based on matrix algebra. In vector logic, true values are mapped to the vector, and logical operators are executed by matrix computation.

**Truth Value Vector Space** $V_2$. A two-valued vector logic uses two $d$-dimensional ($d \geq 2$) column vectors $\vec{s}$ and $\vec{n}$ to represent true and false in the classic binary logic. The two vectors $\vec{s}$ and $\vec{n}$ are real-valued, normally orthogonal to each other, and normalized vectors, *i.e.*, $\|\vec{s}\| = 1, \|\vec{n}\| = 1$. Truth value vector space are generated by $V_2 = \{\vec{s}, \vec{n}\}$, and operations on vectors in truth value space is based on scalar product.

**Operators**. The basic logical operators are associated with their own matrices by vectors in truth-value vector space. Two common types of operators are monadic and dyadic.

(1) **Monadic Operators** are functions: $V_2 \rightarrow V_2$. Two examples are Identity $I = \vec{s}\vec{s}^T + \vec{n}\vec{n}^T$ and Negation $N = \vec{n}\vec{s}^T + \vec{s}\vec{n}^T$ such that $I\vec{s} = \vec{s}, I\vec{n} = \vec{n}, N\vec{n} = \vec{s}, N\vec{s} = \vec{n}$. Note that the truth tables of $I$ and $N$ fulfill the real logical rules of identity and negation in classic boolean logic.

(2) **Dyadic operators** are functions: $V_2 \otimes V_2 \rightarrow V_2$, where $\otimes$ denotes Kronecker product. Dyadic operators include conjunction $C$, disjunction $D$, implication IMPL, equivalence $E$, exclusive or XOR, etc. For example, the conjunction between two logical propositions ($p \wedge q$) is performed by $C(\vec{u} \otimes \vec{v})$, where $C = \vec{s}(\vec{s} \otimes \vec{s})^T + \vec{n}(\vec{s} \otimes \vec{n})^T + \vec{n}(\vec{n} \otimes \vec{s})^T + \vec{n}(\vec{n} \otimes \vec{n})^T$. It can be verified that $C(\vec{s} \otimes \vec{s}) = \vec{s}, C(\vec{s} \otimes \vec{n}) = C(\vec{n} \otimes \vec{s}) = C(\vec{n} \otimes \vec{n}) = \vec{n}$. Dyadic operators which correspond to logical operations in classic binary logic are defined by its formulation to perform logical operations on truth value vectors. Their associated matrices has $d^2$ rows and $d$ columns.

**Many-valued Two-dimensional Logic**. Many-valued logic is introduced to include uncertainties in the logic vector. Weighting $\vec{s}$ and $\vec{n}$ by probabilities, uncertainties are introduced: $\vec{f} = \epsilon\vec{s} + \delta\vec{n}$, where $\epsilon, \delta \in [0, 1]$, $\epsilon + \delta = 1$. Besides, operations on vectors can be simplified to computation on the scalar of these vectors. For example, given two vectors $\vec{u} = \alpha\vec{s} + \beta\vec{n}, \vec{v} = \alpha'\vec{s} + \beta'\vec{n}$, we have:

$$
\begin{aligned}
\text{NOT}(\vec{u}) &= N\vec{u} = (1 - \alpha)\vec{s} + \alpha\vec{n} & \text{NOT}(\alpha) &= \vec{s}^T N\vec{u} = 1 - \alpha \\
\text{OR}(\vec{u}, \vec{v}) &= D(\vec{u} \otimes \vec{v}) & \text{OR}(\alpha, \alpha') &= \vec{s}^T D(\vec{u} \otimes \vec{v}) = \alpha + \alpha' - \alpha\alpha' \\
\text{AND}(\vec{u}, \vec{v}) &= C(\vec{u} \otimes \vec{v}) & \text{AND}(\alpha, \alpha') &= \vec{s}^T C(\vec{u} \otimes \vec{v}) = \alpha\alpha' \qquad (7) \\
\text{IMPL}(\vec{u}, \vec{v}) &= L(\vec{u} \otimes \vec{v}) & \text{IMPL}(\alpha, \alpha') &= \vec{s}^T L(\vec{u} \otimes \vec{v}) = 1 - \alpha(1 - \alpha') \\
\text{XOR}(\vec{u}, \vec{v}) &= X(\vec{u} \otimes \vec{v}) & \text{XOR}(\alpha, \alpha') &= \vec{s}^T X(\vec{u} \otimes \vec{v}) = \alpha + \alpha' - 2\alpha\alpha'
\end{aligned}
$$

## A.3 Neural Dyadic Operators

In this section, we show that all dyadic operators are generated from fuzzy logic in our framework. Below we take union operator for example. We define entity union operator $\mathcal{U}_e$ and time union operator $\mathcal{U}_t$ according to **Alignment Rule** 4.2 as follows:

Fuzzy Logic     Alignment Rule

$$\mathcal{U}_e(\mathbf{V}_{q_1}, \cdots, \mathbf{V}_{q_n}) = (\sum_{i=1}^{n} \boldsymbol{\alpha}_i \boldsymbol{q}_{i,f}^e, \; \mathbf{OR}_{i=1}^{n}\{\boldsymbol{q}_{i,l}^e\}, \; \sum_{i=1}^{n} \boldsymbol{\beta}_i \boldsymbol{q}_{i,f}^t, \; \mathbf{AND}_{i=1}^{n}\{\boldsymbol{q}_{i,l}^t\} \;)$$

$$\mathcal{U}_t(\mathbf{V}_{q_1}, \cdots, \mathbf{V}_{q_n}) = (\sum_{i=1}^{n} \boldsymbol{\alpha}_i \boldsymbol{q}_{i,f}^e, \; \mathbf{AND}_{i=1}^{n}\{\boldsymbol{q}_{i,l}^e\}, \; \sum_{i=1}^{n} \boldsymbol{\beta}_i \boldsymbol{q}_{i,f}^t, \; \mathbf{OR}_{i=1}^{n}\{\boldsymbol{q}_{i,l}^t\} \;)$$

Alignment Rule     Fuzzy Logic

where **AND**, **OR** are the conjunction, disjunction operators in fuzzy logic respectively, $\alpha_i$ and $\beta_i$ are attention weights, as designed in intersection operators. In the time part of entity union operator $\mathcal{U}_e$ and the entity part of time union operator $\mathcal{U}_t$, we follows **Alignment Rule** 4.2 to perform intersection. Besides, please be aware that each operator owns its MLPs and parameters. These operators do not share parameters with each other.

For $n$ queries to intersection/union, the **AND** and **OR** can be written as:

$$\mathbf{AND}_{i=1}^{n}\{\boldsymbol{q}_{i,l}\} = \prod_{i=1}^{n} q_{i,l}$$

$$\mathbf{OR}_{i=1}^{n}\{\boldsymbol{q}_{i,l}\} = \sum_{i=1}^{n} q_{i,l} - \sum_{1 \leqslant i < j \leqslant n} q_{i,l}q_{j,l} + \sum_{1 \leqslant i < j < k \leqslant n} q_{i,l}q_{j,l}q_{k,l} + \ldots + (-1)^{n-1}\prod_{i=1}^{n} q_{i,l}$$

The equations come from the definition of AND and OR in vector logic. The cost of **OR** operator's implementation is not as expensive as it seems to be. In fact, it's $O(n*d)$ complexity, where $d$ is the embedding dimension. We provide the proof in next section.

Other dyadic operators can be generated in the same way. Since listing all operators is boring, we provide the last example. Then, the entity implication operator $\mathcal{L}_e$ and time implication operator $\mathcal{L}_t$ are defined as follows:

Fuzzy Logic     Alignment Rule

$$\mathcal{L}_e(\mathbf{V}_{q_1}, \cdots, \mathbf{V}_{q_n}) = (\sum_{i=1}^{n} \boldsymbol{\alpha}_i \boldsymbol{q}_{i,f}^e, \; \mathbf{IMPL}_{i=1}^{n}\{\boldsymbol{q}_{i,l}^e\}, \; \sum_{i=1}^{n} \boldsymbol{\beta}_i \boldsymbol{q}_{i,f}^t, \; \mathbf{AND}_{i=1}^{n}\{\boldsymbol{q}_{i,l}^t\} \;)$$

$$\mathcal{L}_t(\mathbf{V}_{q_1}, \cdots, \mathbf{V}_{q_n}) = (\sum_{i=1}^{n} \boldsymbol{\alpha}_i \boldsymbol{q}_{i,f}^e, \; \mathbf{AND}_{i=1}^{n}\{\boldsymbol{q}_{i,l}^e\}, \; \sum_{i=1}^{n} \boldsymbol{\beta}_i \boldsymbol{q}_{i,f}^t, \; \mathbf{IMPL}_{i=1}^{n}\{\boldsymbol{q}_{i,l}^t\} \;)$$

Alignment Rule     Fuzzy Logic

## A.4 Theoretical Analysis

To understand why the feature-logic framework works, we have the following proposition, which shows that the designed intersection and union operators obey commutative law of real logical operations. The proofs are presented in next section in Appendix A.5:

**Proposition 1.** *Commutativity: Given Temporal Feature-Logic embedding $\mathcal{V}_{q_a}, \mathcal{V}_{q_b}$, we have $\mathcal{I}_e(\mathcal{V}_{q_a}, \mathcal{V}_{q_b}) = \mathcal{I}_e(\mathcal{V}_{q_b}, \mathcal{V}_{q_a})$ and $\mathcal{U}_e(\mathcal{V}_{q_a}, \mathcal{V}_{q_b}) = \mathcal{U}_e(\mathcal{V}_{q_b}, \mathcal{V}_{q_a})$, $\mathcal{I}_t(\mathcal{V}_{q_a}, \mathcal{V}_{q_b}) = \mathcal{I}_t(\mathcal{V}_{q_b}, \mathcal{V}_{q_a})$ and $\mathcal{U}_t(\mathcal{V}_{q_a}, \mathcal{V}_{q_b}) = \mathcal{U}_t(\mathcal{V}_{q_b}, \mathcal{V}_{q_a})$.*

## A.5 Proofs

In this part, we prove that TFLEX has the property of commutativity. Besides, we also show that the computation complexity of **OR** operation is linear to the number of queries.

*Proof.* (Proposition 1) **Commutativity**:

For the intersection operations, as the calculations of $\mathcal{I}_e$ and $\mathcal{I}_t$ are identical, here, we only prove that $\mathcal{I}_e$ complies with commutative law. The entity feature part and the time feature part of the result are computed as a weighted summation of each query's corresponding parts. Since addition is commutative and the attention weights do not concern the order of calculations, both feature parts' calculations are commutative.

Then, we discuss the logic parts. The logic parts only include the AND in fuzzy logic which, essentially, is just the multiplication of each element by the definition provided above. Because multiplication is surely commutative, the calculation of either entity logic part or time logic part is commutative. Thus the intersection operation $\mathcal{I}_e(\mathcal{I}_t)$ is commutative.

As for $\mathcal{U}_e$ and $\mathcal{U}_t$, their feature parts have the same form of weighted summation as the intersection operations do. Thus, the feature parts of both $\mathcal{U}_e$ and $\mathcal{U}_t$ comply to commutative law. Also, the *time logic part* of $\mathcal{U}_e$ and the *entity logic part* of $\mathcal{U}_t$ solely concern AND operator which has been proved commutative before. The OR operator, by definition, gives the aggregation of different accumulative parts each of which is commutative itself. Also, the multiplications in each of the summations are commutative. Hence, OR operation is invariant to the order of calculations, which finally gives the calculations of *entity logic part* of $\mathcal{U}_e$ and the *time logic part* of $\mathcal{U}_t$ are commutative. Then we can naturally affirm that $\mathcal{U}_e$ and $\mathcal{U}_t$ are commutative as well.

$\square$

*Proof.* Additionally, we prove that the cost of OR operator's implementation is not as expensive as it seems to be. The process of computation can be described as follows:

---

**Algorithm 1:** OR

---

**Input**: queries $\{q_1...q_n\}$
**Output**: the union of queries
  1: Let $u = 0$.
  2: For $q$ In $\{q_1...q_n\}$
  3:   $u = u + q - u * q$
  4: EndFor
  5: **return** $u$

---

As we implement OR step by step on a series of *n* queries, the loop goes n-1 times in total. Assuming the embedding dimension of a query is *d*, we can have the cost of OR is O(*n*d*). $\square$

# B  More Details about Experiments

In this section, we show more details about our experiments. Firstly, we introduce the origin datasets in Section B.1. Then, we describe the details on how we define the query structure and how to sample queries in Section B.2. With the generated queries, we construct three datasets and report their statistics in Section B.3. We also show the details of our model implementation in Section B.4 and evaluation protocol in Section B.5. Besides, we show more experimental data for sensitive analysis in Section B.6. Finally, we present the detail metrics of main results in Section B.7.

## B.1  Details of Origin Datasets

To construct the reasoning dataset, we use three datasets of temporal KGs that have become standard benchmarks for TKGC: ICEWS14 [37], ICEWS05-15 [37], and GDELT-500 [38]. The two subsets of ICEWS are generated by Boschee et al. [37]: ICEWS14 corresponding to the facts in 2014 and ICEWS05-15 corresponding to the facts between 2005 and 2015. GDELT-500 generated by Leetaru and Schrodt [38] corresponds to the facts from April 1, 2015, to March 31, 2016. The statistics of the dataset are shown in Table 8.

Table 4: Definition of query structures in static query embeddings [2–4]. Operators are defined on entity sets only, including Pe, And, Or, and Not. 'e', 'r', 'n', 'u' denote entity, relation, negation and union, respectively. The braket '()' denotes a relational projection or intersection operation.

| Query Name | Query Structure Definition |
|---|---|
| **1p** | ('e', ('r',)) |
| **2p** | ('e', ('r', 'r')) |
| **3p** | ('e', ('r', 'r', 'r')) |
| **2i** | (('e', ('r',)), ('e', ('r',))) |
| **3i** | (('e', ('r',)), ('e', ('r',)), ('e', ('r',))) |
| **ip** | ((('e', ('r',)), ('e', ('r',))), ('r',)) |
| **pi** | (('e', ('r', 'r')), ('e', ('r',))) |
| **2in** | (('e', ('r',)), ('e', ('r', 'n'))) |
| **3in** | (('e', ('r',)), ('e', ('r',)), ('e', ('r', 'n'))) |
| **inp** | ((('e', ('r',)), ('e', ('r', 'n'))), ('r',)) |
| **pin** | (('e', ('r', 'r')), ('e', ('r', 'n'))) |
| **pni** | (('e', ('r', 'r', 'n')), ('e', ('r',))) |
| **2u-DNF** | (('e', ('r',)), ('e', ('r',)), ('u',)) |
| **up-DNF** | ((('e', ('r',)), ('e', ('r',)), ('u',)), ('r',)) |
| **2u-DM** | ((('e', ('r', 'n')), ('e', ('r', 'n'))), ('n',)) |
| **up-DM** | ((('e', ('r', 'n')), ('e', ('r', 'n'))), ('n', 'r')) |

Table 5: Basic functions. $Qe$ is entity set and $Qt$ is timestamp set. $\mathcal{E}$ is the set of all entities, $\mathcal{T}$ is the set of all timestamps and $\mathcal{F}$ is the set of triples.

| Name | Symbol | Input | Output |
|---|---|---|---|
| And | And | $Qe_1, Qe_2$ | $Qe_1 \cap Qe_2$ |
| Or | Or | $Qe_1, Qe_2$ | $Qe_1 \cup Qe_2$ |
| Not | Not | $Qe$ | $\mathcal{E} - Qe$ |
| EntityProjection | Pe | $Qe$, r, $Qt$ | $\{o | s \in Qe, t \in Qt, (s, r, o, t) \in \mathcal{F}\}$ |
| TimeProjection | Pt | $Qe_1$, r, $Qe_2$ | $\{t | s \in Qe_1, o \in Qe_2, (s, r, o, t) \in \mathcal{F}\}$ |
| TimeAnd | TimeAnd | $Qt_1, Qt_2$ | $Qt_1 \cap Qt_2$ |
| TimeOr | TimeOr | $Qt_1, Qt_2$ | $Qt_1 \cup Qt_2$ |
| TimeNot | TimeNot | $Qt$ | $\mathcal{T} - Qt$ |
| before | before | $Qt$ | $\{t | t < \min(Qt)\}$ |
| after | after | $Qt$ | $\{t | t > \max(Qt)\}$ |

## B.2 Details of Query Generation

**The Design of Query Structure Schema**. Previous query embedding methods share the same query structure schema (Table 4) to perform complex logical reasoning over static knowledge graph. When it comes to dynamic knowledge graph (TKG), the static schema is not sufficient to capture the temporal reasoning. In this paper, we propose a novel query structure schema for reasoning. The schema is composed of basic functions and query structures. The basic functions are defined in Table 5. The query structures are shown in Table 6 with visualization in Figure 5 and Figure 6. The query structures are based on the basic functions: **And**, **Or**, **Not**, **EntityProjection**, **TimeProjection**, **TimeAnd**, **TimeOr**, **TimeNot**, **Before**, and **After**. Of course, users can define more basic functions or query structures to extend the schema to multi-modal KGs, hyper-relation KGs and so on. It is a flexible schema overall.

In order to keep a similar experiment settings with previous static query embeddings, the query structures are further aggregated into groups, shown in Table 7. The groups $\text{avg}_e, \text{avg}_t, \text{avg}_{e,\mathcal{C}_e}, \text{avg}_{t,\mathcal{C}_t}$ are for training and testing. Besides, extra groups $\text{avg}_{\{\mathcal{U}_e\}}, \text{avg}_{\{\mathcal{U}_t\}}, \text{avg}_x$ are for evaluation and testing only.

In implementation, we define the basic functions and query structures as functions in python, which are named **Complex Query Function (CQF)**. There are lots of advantages of using python functions to define query structures. Firstly, the functions are easy to understand and debug. Secondly, the functions are easy to be reused in the dataset-sampling process and model-training process. Thirdly,

Table 6: Definition of query structures in temporal query embeddings. Operators on entity sets include Pe, And, Or, and Not. Operators on timestamp sets include Pt, TimeAnd, TimeOr, TimeNot, after, and before. The prefix "a" and "b" represent "after" and "before" respectively. The prefix "s" or "o" mean that the sub-query is a subject query or an object query respectively.

| Type | Query Name | Query Structure Definition |
|---|---|---|
| entity multi-hop | **Pe** | Pe(s, r, t) |
| | **Pe2** | Pe(Pe(s1, r1, t1), r2, t2) |
| | **Pe3** | Pe(Pe(Pe(s1, r1, t1), r2, t2), r3, t3) |
| | **e2i** | And(Pe(s1, r1, t1), Pe(s2, r2, t2)) |
| | **e3i** | And(Pe(s1, r1, t1), Pe(s2, r2, t2), Pe(e3, r3, t3)) |
| entity not | **e2i_N** | And(Pe(s1, r1, t1), Not(Pe(s2, r2, t2))) |
| | **e3i_N** | And(Pe(s1, r1, t1), Pe(s2, r2, t2), Not(Pe(s3, r3, t3))) |
| | **Pe_e2i_N** | Pe(And(Pe(s1, r1, t1), Not(Pe(s2, r2, t2))), r3, t3) |
| | **e2i_PeN** | And(Pe(Pe(s1, r1, t1), r2, t2), Not(Pe(s2, r3, t3))) |
| | **e2i_NPe** | And(Not(Pe(Pe(s1, r1, t1), r2, t2)), Pe(s2, r3, t3)) |
| entity union | **e2u** | Or(Pe(s1, r1, t1), Pe(s2, r2, t2)) |
| | **Pe_e2u** | Pe(Or(Pe(s1, r1, t1), Pe(s2, r2, t2)), r3, t3) |
| time multi-hop | **Pt** | Pt(s, r, o) |
| | **aPt** | after(Pt(s, r, o)) |
| | **bPt** | before(Pt(s, r, o)) |
| | **Pe_Pt** | Pe(s1, r1, Pt(s2, r2, o1)) |
| | **Pt_sPe_Pt** | Pt(Pe(s1, r1, Pt(s2, r2, o1)), r3, o2) |
| | **Pt_oPe_Pt** | Pt(s1, r1, Pe(s2, r2, Pt(s3, r3, o1))) |
| | **t2i** | TimeAnd(Pt(s1, r1, o1), Pt(s2, r2, o2)) |
| | **t3i** | TimeAnd(Pt(s1, r1, o1), Pt(s2, r2, o2), Pt(s3, r3, o3)) |
| time not | **t2i_N** | TimeAnd(Pt(s1, r1, o1), TimeNot(Pt(s2, r2, o2))) |
| | **t3i_N** | TimeAnd(Pt(s1, r1, o1), Pt(s2, r2, o2), TimeNot(Pt(s3, r3, o3))) |
| | **Pe_t2i_N** | Pe(s1, r1, TimeAnd(Pt(Pe(s2, r2, t1), r3, o1), TimeNot(Pt(s3, r4, o2)))) |
| | **t2i_NPt** | TimeAnd(TimeNot(Pt(Pe(s1, r1, t1), r2, o1)), Pt(s2, r3, o2)) |
| | **t2i_PtN** | TimeAnd(Pt(Pe(s1, r1, t1), r2, o1), TimeNot(Pt(s2, r3, o2))) |
| time union | **t2u** | TimeOr(Pt(s1, r1, o1), Pt(s2, r2, o2)) |
| | **Pe_t2u** | Pe(s1, r1, TimeOr(Pt(s2, r2, o1), Pt(s3, r3, o2))) |
| hybrid multi-hop | **between** | TimeAnd(after(Pt(s1, r1, o1)), before(Pt(s2, r2, o2))) |
| | **e2i_Pe** | And(Pe(Pe(s1, r1, t1), r2, t2), Pe(s2, r3, t3)) |
| | **Pe_e2i** | Pe(e2i(s1, r1, t1, s2, r2, t2), r3, t3) |
| | **t2i_Pe** | TimeAnd(Pt(Pe(s1, r1, t1), r2, o1), Pt(s2, r3, o2)) |
| | **Pe_t2i** | Pe(s1, r1, t2i(s2, r2, o1, s3, r3, o2)) |
| | **Pe_aPt** | Pe(s1, r1, after(Pt(s2, r2, o1))) |
| | **Pe_bPt** | Pe(s1, r1, before(Pt(s2, r2, o1))) |
| | **Pe_at2i** | Pe(s1, r1, after(t2i(s2, r2, o1, s3, r3, o2))) |
| | **Pe_bt2i** | Pe(s1, r1, before(t2i(s2, r2, o1, s3, r3, o2))) |
| | **Pt_sPe** | Pt(Pe(s1, r1, t1), r2, o1) |
| | **Pt_oPe** | Pt(s1, r1, Pe(s2, r2, t1)) |
| | **Pt_se2i** | Pt(e2i(s1, r1, t1, s2, r2, t2), r3, o1) |
| | **Pt_oe2i** | Pt(s1, r1, e2i(s2, r2, t1, s3, r3, t2)) |

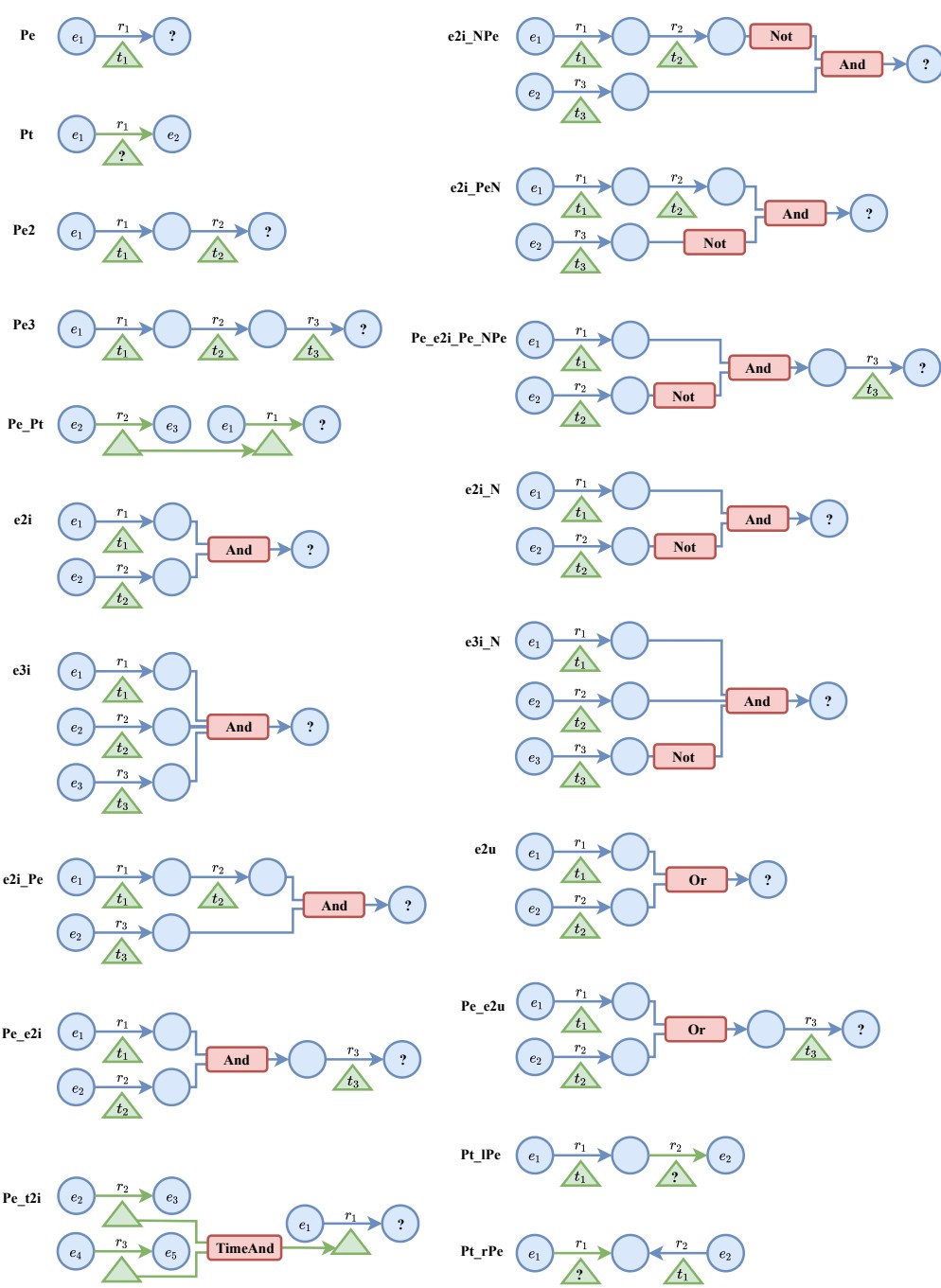

Figure 5: Visualization of temporal query structures answering entity sets. These structures are basic functions that can be used to construct more complex temporal query structures.

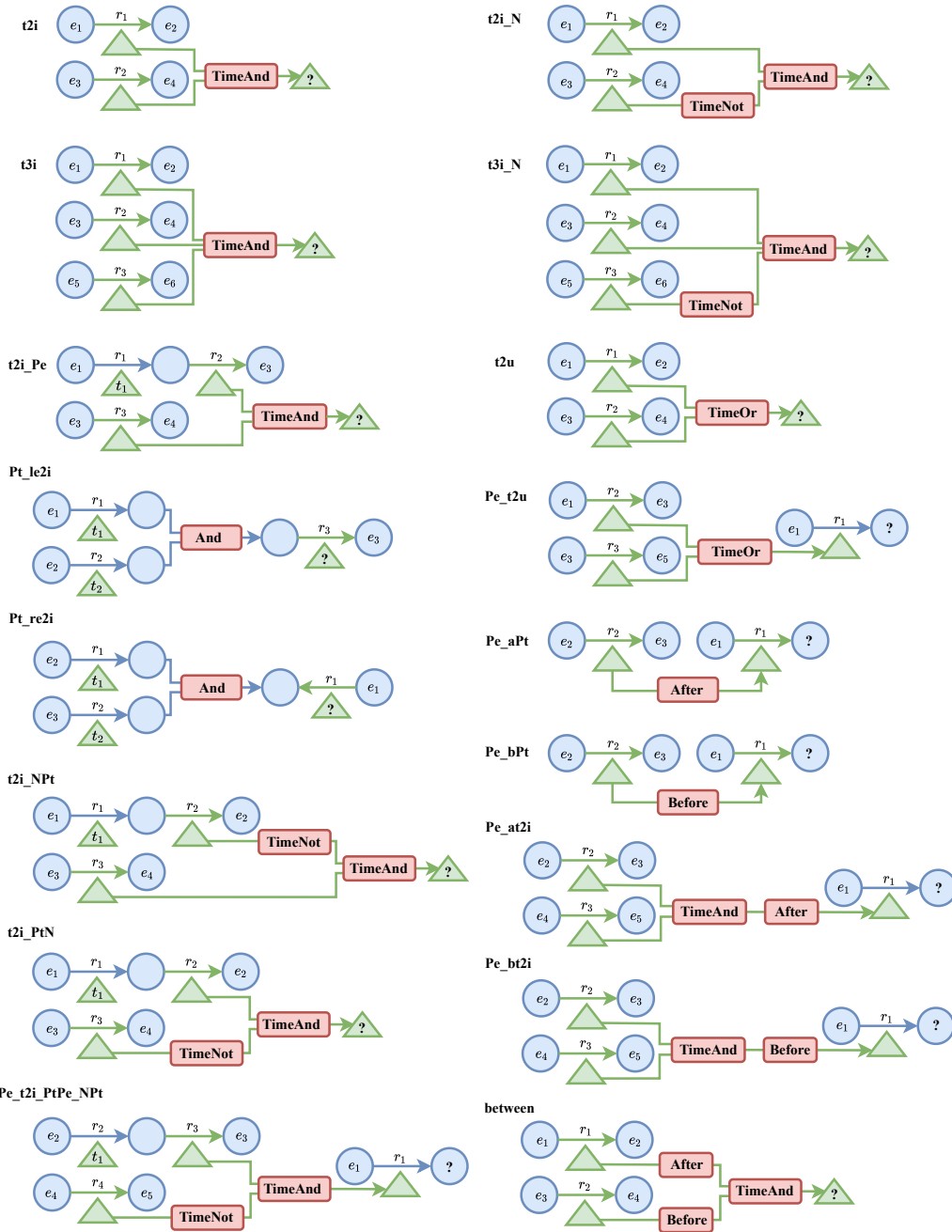

Figure 6: Visualization of temporal query structures answering timestamp sets. These structures are basic functions that can be used to construct more complex temporal query structures.

Table 7: The query structures are aggregated into groups. The groups are inspired by the experiment settings in static query embeddings [1–4]. The groups $\text{avg}_e$, $\text{avg}_t$, $\text{avg}_{e,\mathcal{C}_e}$, $\text{avg}_{t,\mathcal{C}_t}$ are for training and testing. Besides, extra groups $\text{avg}_{\{\mathcal{U}_e\}}$, $\text{avg}_{\{\mathcal{U}_t\}}$, $\text{avg}_x$ are for evaluation and testing only.

| | 1p | 2p | 3p | 2i | 3i |
|---|---|---|---|---|---|
| $\text{avg}_e$ | Pe | Pe2 | Pe3 | e2i | e3i |
| $\text{avg}_t$ | Pt, aPt, bPt | Pe_Pt | Pt_sPe_Pt, Pt_oPe_Pt | t2i | t3i |
| | 2in | 3in | inp | pin | pni |
| $\text{avg}_{e,\mathcal{C}_e}$ | e2i_N | e3i_N | Pe_e2i_N | e2i_PeN | e2i_NPe |
| $\text{avg}_{t,\mathcal{C}_t}$ | t2i_N | t3i_N | Pe_t2i_N | t2i_PtN | t2i_NPt |
| | 2u | up | | | |
| $\text{avg}_{\{\mathcal{U}_e\}}$ | e2u | Pe_e2u | | | |
| $\text{avg}_{\{\mathcal{U}_t\}}$ | t2u | Pe_t2u | | | |
| | pi | ip | | | |
| $\text{avg}_x$ | e2i_Pe | Pe_e2i | | Pe_aPt | Pe_bPt |
| | t2i_Pe | Pe_t2i | | Pe_at2i | Pe_bt2i |
| | | | | Pt_sPe | Pt_oPe |
| | | | between | Pt_se2i | Pt_oe2i |

the functions are easy to be extended to support more complex query structures and reimplement the existing query embedding methods.

As is introduced in Section 4, we use the computation graph to represent the reasoning process. To get the computation graph of a CQF, we use the python interpreter to parse the function to Abstract Syntax Tree (AST), which is a more friendly readable subset of computation graph. In this way, executing the computation graph is equivalent to executing the CQF in python interpreter. Since we have the god privileged access to the AST, we can modify the interpreter to support various dynamic reasoning processes. Below we show how to unify (1) the reasoning of ground-truth, (2) the sampling of anchors, and (3) the reasoning of embedding-based methods by modifying the python interpreter.

**Ground-Truth Reasoning Interpreter**.

In the reasoning of ground-truth, we need to get all real answers of a query under a given TKG. To achieve this, the first we need to do is to implement the basic functions. We wrap the python built-in set to QuerySet class to store entities and timestamps. The basic functions are implemented as lambda function in python, with input and output as QuerySet. Then, the basic functions are registered as symbols to the python interpreter. When executing the CQF, the python interpreter will call the corresponding basic functions to get the final answer. The answers are finally generated from subgraph matching, which depends on the ground truth in TKG. We show the pseudocode of the ground-truth reasoning interpreter in Figure 7.

**Anchors Sampling Interpreter**.

The anchor, which may be entity, relation or timestamp, is the input argument of the CQF. The aim of the anchors sampling interpreter is to get the possible anchors and answers of a query under a given TKG. For the sampled anchors, we expect the answer set not empty. Since the answers could be obtained from the ground-truth reasoning interpreter, we only focus on the sampling of anchors in the anchors sampling interpreter.

Given the standard split of edges into training ($\mathcal{F}_{\text{train}}$), validation ($\mathcal{F}_{\text{valid}}$) and test ($\mathcal{F}_{\text{test}}$) sets, we append inverse relations and double the number of edges in the graph. Then we create three graph: $\mathcal{G}_{\text{train}} = \{\mathcal{V}, \mathcal{R}, \mathcal{T}, \mathcal{F}_{\text{train}}\}$, $\mathcal{G}_{\text{valid}} = \{\mathcal{V}, \mathcal{R}, \mathcal{T}, \mathcal{F}_{\text{train}} + \mathcal{F}_{\text{valid}}\}$, $\mathcal{G}_{\text{test}} = \{\mathcal{V}, \mathcal{R}, \mathcal{T}, \mathcal{F}_{\text{train}} + \mathcal{F}_{\text{valid}} + \mathcal{F}_{\text{test}}\}$. Given a query $q$, let $[\![q]\!]_{\text{train}}$, $[\![q]\!]_{\text{valid}}$, and $[\![q]\!]_{\text{test}}$ denote a set of answers (entities or timestamps) obtained by subgraph matching of $q$ on $\mathcal{G}_{\text{train}}$, $\mathcal{G}_{\text{valid}}$ and $\mathcal{G}_{\text{test}}$. For each query $q$, the reasoning process starts from anchor nodes and computes the final answer set via subgraph matching.

For the basic function **Pe** and **Pt**, we simply use all the triples and extract the entities and timestamps as the anchors. These anchors are supposed to have no empty answers under **Pe** and **Pt**. However, when it comes to some hard queries, such as **e2i** and **e3i**, the random sampled anchors may have empty answers. Because the TKG is sparse and incomplete, the intersection of two query sets has a high probability to be empty. The empty answers are meaningless for model-training and damage to

```python
# 1. predefined query structures and the Interpreter
ComplexQueryFunctions = {
    "Pe2": "def Pe2(e1, r1, t1, r2, t2): return Pe(Pe(e1, r1, t1), r2, t2)",  # 2p
    "Pe3": "def Pe3(e1, r1, t1, r2, t2, r3, t3): return Pe(Pe(Pe(e1, r1, t1), r2, t2), r3, t3)",  # 3p
    "aPt": "def aPt(s, r, o): return TimeAfter(Pt(s, r, o))",  # a for after
    "bPt": "def bPt(s, r, o): return TimeBefore(Pt(s, r, o))",  # b for before
    ...
}
class Interpreter:
    def __init__(self, symbols: Dict[str, Any]):
        self.symbols = symbols
        for k, CQF in ComplexQueryFunctions.items():
            self.eval(CQF)
    def eval(self, line: str) -> Any: return ast.parse(line).eval(self)

# 2. GroundTruth Reasoning Interpreter
QuerySet  = set
EntitySet = set
TimeSet   = set
class GroundTruthReasoningInterpreter(Interpreter):
    def __init__(self, TKG):
        Pe = defaultdict(lambda:defaultdict(lambda:defaultdict(set)))
        Pt = defaultdict(lambda:defaultdict(lambda:defaultdict(set)))
        for s, r, o, t in TKG.F:  # F is the set of all facts
            Pe[s][r][t].add(o)
            Pt[s][r][o].add(t)
        symbols = {
            "Pe": lambda (qs, r, qt): EntitySet(reduce(|, [Pe[s][r][t] for s, t in product(qs, qt)])),
            "Pt": lambda (qs, r, qo): TimeSet(reduce(|,   [Pt[s][r][o] for s, o in product(qs, qo)])),
            "And": lambda (q1, q2): EntitySet(q1 & q2)
            "Or":  lambda (q1, q2): EntitySet(q1 | q2)
            "Not": lambda (qe): EntitySet(TKG.V - qe)  # V is the set of all entities
            "TimeAnd": lambda (q1, q2): TimeSet(q1 & q2)
            "TimeOr":  lambda (q1, q2): TimeSet(q1 | q2)
            "TimeNot": lambda (qt): TimeSet(TKG.T - qt)  # T is the set of all timestamps
            "TimeBefore": lambda (qt): TimeSet([t for t in TKG.T if t < min(qt)])
            "TimeAfter":  lambda (qt): TimeSet([t for t in TKG.T if t > max(qt)])
        }
        super().__init__(symbols)

# TKG.F = {(Angela Merkel, make a visit, China, 2010-07-16)}
p = GroundTruthReasoningInterpreter(TKG)
answer = p.eval("Pe({'Angela Merkel'}, 'make a visit', {'2010-07-16'})")
assert answer == EntitySet('China')
```

Figure 7: Python-style pseudocode of ground-truth reasoning interpreter.

the performance of data-sampling process. To solve this problem, we use the following two strategies to accelerate the sampling.

(1) **Inverse Sampling**: We randomly sample an entity as objective, denoted $e_o$. Then we sample subjective entity $e_{s_1}, e_{s_2}$, relation $r_1, r_2$ and timestamp $t_1, t_2$, according to the fact $(e_{s_1}, r_1, e_o, t_1)$ and $(e_{s_2}, r_2, e_o, t_2)$. The sampled anchors are $(e_{s_1}, r_1, t_1; e_{s_2}, r_2, t_2)$. These anchors under **e2i** have the answer $e_o$ at least, which asserts that the answer set is not empty. Such sampling method that samples the answer first and then samples the anchors is called **Inverse Sampling**.

(2) **Bi-directional Sampling**: For long multi-hop queries, such as **Pe2**, the complexity of random sampling is $O(N^{2L})$, where $N$ is the entity count and $L$ is the length of the query. **Pe2** has $L = 2$. It contains an anchor-sampling complexity of $O(N^L)$ and a ground-truth reasoning (to get answers) complexity of $O(N^L)$. The origin sampling is to sample one subjective entity, get the objective as answers as next subjective anchors, again get the next objective answers. The final answers recursively depend on the initial subjective entity, leading to low performance. To accelerate, we sample an entity at the middle of AST, denoted $e_m$. Then we sample in two directions. One is forward, $e_m$ as the subjective entity, to sample relation $r_2$ and timestamp $t_2$. The other is backward, $e_m$ as the objective entity, to sample subjective anchor $e_1$, relation $r_1$ and timestamp $t_1$. The sampled anchors are $(e_1, r_1, t_1; r_2, t_2)$. The final answer set is asserted not empty, because it at least contains all the answers of $Pe(e_m, r_2, t_2)$. Be aware that the two directions are independent, which can be sampled in parallel. Such sampling method that samples the answer in two directions is called **Bi-directional Sampling**, which can reduce the computation complexity to $O(N^{L+\frac{L}{2}})$, composed of a sampling complexity of $O(N^{\frac{L}{2}})$ and a ground-truth reasoning complexity of $O(N^L)$.

```
$ python run_reasoning_interpreter.py
you: use_dataset(data_home="/data/TFLEX/data"); use_embedding_reasoning_interpreter("TFLEX_dim800_gamma15", device="cuda:1");
data already prepared, using cache
load best model at step 190000 with score 0.07013575653002799
bot: using embedding reasoning interpreter
you: sample(task_name="e2i", k=1);
def e2i(e1, r1, t1, e2, r2, t2): return And(Pe(e1, r1, t1), Pe(e2, r2, t2))
sample 1 e2i(e1, r1, t1, e2, r2, t2) data:
queries: [29, 155, 47, 29, 372, 47], easy_answer: {2669, 1751}, hard_answer: {2083}
bot: [[[29, 155, 47, 29, 372, 47], {2669, 1751}, {2083, 2669, 1751}]]
you: emb_e1=entity_token(29); emb_r1=relation_token(155); emb_t1=timestamp_token(47);
you: emb_e2=entity_token(29); emb_r2=relation_token(372); emb_t2=timestamp_token(47);
you: emb_q1 = Pe(emb_e1, emb_r1, emb_t1)
you: emb_q2 = Pe(emb_e2, emb_r2, emb_t2)
you: emb_q = And(emb_q1, emb_q2)
you: embedding_answer_entities(emb_q, topk=3)
bot: [[('Djibouti', 7.566648960113525), ('Head of Government (Djibouti)', 6.4008941650390625), ('Ethiopia', 4.457242012023926)]]
you: use_groundtruth_reasoning_interpreter()
bot: using groundtruth interpreter
you: groundtruth_answer(e2669) + groundtruth_answer(e1751) + groundtruth_answer(e2083)
bot: ['Head of Government (Djibouti)', 'Djibouti', 'Ethiopia']
you: OK. The bot correctly predicts the hard answer which only exists in the test set!█
```

Figure 8: The screenshot of a real running interpreter.

**Embedding Reasoning Interpreter**.

On the contrary of the ground-truth reasoning interpreter which has '**set**' as the input and output of the CQFs, in the embedding reasoning interpreter, the input and output of CQFs are '**embedding vectors**'. The embedding-based method for reasoning over TKG is called **Temporal Query Embedding**. The computation complexity of the embedding reasoning interpreter is $O(L + N)$, where $L$ is the length of the query and $N$ is the entity count. The complexity consists of a query-embedding complexity of $O(L)$ and a scoring-to-answer complexity of $O(N)$. The embedding reasoning is much faster than the ground-truth reasoning, which has a computation complexity of $O(N^L)$.

To implement the embedding reasoning interpreter, we just need to replace the basic functions with neural networks. The symbols dict to pass to the interpreter is

$$
\begin{aligned}
\{ \\
&\text{"Pe": } \mathcal{P}_e, \text{ "Pt": } \mathcal{P}_t, \\
&\text{"And": } \mathcal{I}_e, \text{ "Or": } \mathcal{U}_e, \text{ "Not": } \mathcal{N}_e, \\
&\text{"TimeAnd": } \mathcal{I}_t, \text{ "TimeOr": } \mathcal{U}_t, \text{ "TimeNot": } \mathcal{N}_t, \\
&\text{"TimeAfter": } \mathcal{A}_t, \text{ "TimeBefore": } \mathcal{B}_t \\
\}
\end{aligned}
\tag{8}
$$

The embedding vectors are learned as is introduced in Section 4. Unlike the ground-truth reasoning, the embedding reasoning is fuzzy. The output of the embedding reasoning interpreter is an embedding vector, which is a fuzzy set representation. To get the final answer set, we need to use the distance function (in Section 4.3) to score to candidate answers. In this way, the embedding vector is converted to the final answer set. The final answers are given in the ranking list, where each answer is followed by its distance to the query. Usually the top-k answers are accepted as the final answers.

Note that the interpreter is flexible, and easy to implement and extend. In order to reproduce static query embeddings [2–4] over dynamic knowledge graphs, the only thing to do is to implement the symbols dict and the distance function.

We hope that the design of interpreter is helpful for the future research of temporal query embedding.

**Screenshot**.

Additionally, we show the screenshot of a real running interpreter in Figure 8. In the screenshot, we randomly select an example of **e2i** query. Then, we use the embedding reasoning interpreter to get the final answer set step by step. The final answer set is shown in the ranking list, where each answer is followed by its similarity score to the query. Finally, the bot correctly predicts the hard answer which only exists in the test set!

## B.3 Details of Generated Datasets

Finally, we generate three datasets for the task of temporal complex reasoning using the process in Section B.2. The statistics of the generated datasets are listed below. Table 9 show the count of query

Table 8: Statistics on ICEWS14, ICEWS05-15, and GDELT-500.

| Dataset | Entities | Relations | Timestamps | \|Training\| | \|Validation\| | \|Test\| | Total Edges |
|---|---|---|---|---|---|---|---|
| ICEWS14 | 7,128 | 230 | 365 | 72,826 | 8,941 | 8,963 | 90,730 |
| ICEWS05-15 | 10,488 | 251 | 4,017 | 368,962 | 46,275 | 46,092 | 479,329 |
| GDELT-500 | 500 | 20 | 366 | 2,735,685 | 341,961 | 341,961 | 3,419,607 |

Table 9: Queries count for each dataset.

| Query Name | ICEWS14 | | | ICEWS05-15 | | | GDELT-500 | | |
|---|---|---|---|---|---|---|---|---|---|
| | Train | Validate | Test | Train | Validate | Test | Train | Validate | Test |
| Pe | 66783 | 8837 | 8848 | 344042 | 45829 | 45644 | 1115102 | 273842 | 273432 |
| Pe2 | 72826 | 3482 | 4037 | 368962 | 10000 | 10000 | 2215309 | 10000 | 10000 |
| Pe3 | 72826 | 3492 | 4083 | 368962 | 10000 | 10000 | 2215309 | 10000 | 10000 |
| e2i | 72826 | 3305 | 3655 | 368962 | 10000 | 10000 | 2215309 | 10000 | 10000 |
| e3i | 72826 | 2966 | 3023 | 368962 | 10000 | 10000 | 2215309 | 10000 | 10000 |
| Pt | 42690 | 7331 | 7419 | 142771 | 28795 | 28752 | 687326 | 199780 | 199419 |
| aPt | 13234 | 4411 | 4411 | 68262 | 10000 | 10000 | 221530 | 10000 | 10000 |
| bPt | 13234 | 4411 | 4411 | 68262 | 10000 | 10000 | 221530 | 10000 | 10000 |
| Pe_Pt | 7282 | 3385 | 3638 | 36896 | 10000 | 10000 | 221530 | 10000 | 10000 |
| Pt_sPe_Pt | 13234 | 5541 | 6293 | 68262 | 10000 | 10000 | 221530 | 10000 | 10000 |
| Pt_oPe_Pt | 13234 | 5480 | 6242 | 68262 | 10000 | 10000 | 221530 | 10000 | 10000 |
| t2i | 72826 | 5112 | 6631 | 368962 | 10000 | 10000 | 2215309 | 10000 | 10000 |
| t3i | 72826 | 3094 | 3296 | 368962 | 10000 | 10000 | 2215309 | 10000 | 10000 |
| e2i_N | 7282 | 2949 | 2975 | 36896 | 10000 | 10000 | 221530 | 10000 | 10000 |
| e3i_N | 7282 | 2913 | 2914 | 36896 | 10000 | 10000 | 221530 | 10000 | 10000 |
| Pe_e2i_N | 7282 | 2968 | 3012 | 36896 | 10000 | 10000 | 221530 | 10000 | 10000 |
| e2i_PeN | 7282 | 2971 | 3031 | 36896 | 10000 | 10000 | 221530 | 10000 | 10000 |
| e2i_NPe | 7282 | 3061 | 3192 | 36896 | 10000 | 10000 | 221530 | 10000 | 10000 |
| t2i_N | 7282 | 3135 | 3328 | 36896 | 10000 | 10000 | 221530 | 10000 | 10000 |
| t3i_N | 7282 | 2924 | 2944 | 36896 | 10000 | 10000 | 221530 | 10000 | 10000 |
| Pe_t2i_N | 7282 | 3031 | 3127 | 36896 | 10000 | 10000 | 221530 | 10000 | 10000 |
| t2i_PtN | 7282 | 3300 | 3609 | 36896 | 10000 | 10000 | 221530 | 10000 | 10000 |
| t2i_NPt | 7282 | 4873 | 5464 | 36896 | 10000 | 10000 | 221530 | 10000 | 10000 |
| e2u | - | 2913 | 2913 | - | 10000 | 10000 | - | 10000 | 10000 |
| Pe_e2u | - | 2913 | 2913 | - | 10000 | 10000 | - | 10000 | 10000 |
| t2u | - | 2913 | 2913 | - | 10000 | 10000 | - | 10000 | 10000 |
| Pe_t2u | - | 2913 | 2913 | - | 10000 | 10000 | - | 10000 | 10000 |
| between | 7282 | 2913 | 2913 | 36896 | 10000 | 10000 | 221530 | 10000 | 10000 |
| e2i_Pe | - | 2913 | 2913 | - | 10000 | 10000 | - | 10000 | 10000 |
| Pe_e2i | - | 2913 | 2913 | - | 10000 | 10000 | - | 10000 | 10000 |
| t2i_Pe | - | 2913 | 2913 | - | 10000 | 10000 | - | 10000 | 10000 |
| Pe_t2i | - | 2913 | 2913 | - | 10000 | 10000 | - | 10000 | 10000 |
| Pe_aPt | 7282 | 4134 | 4733 | 68262 | 10000 | 10000 | 221530 | 10000 | 10000 |
| Pe_at2i | 7282 | 4607 | 5338 | 36896 | 10000 | 10000 | 221530 | 10000 | 10000 |
| Pt_sPe | 7282 | 4976 | 5608 | 36896 | 10000 | 10000 | 221530 | 10000 | 10000 |
| Pt_se2i | 7282 | 3226 | 3466 | 36896 | 10000 | 10000 | 221530 | 10000 | 10000 |
| Pe_bPt | 7282 | 3970 | 4565 | 36896 | 10000 | 10000 | 221530 | 10000 | 10000 |
| Pe_bt2i | 7282 | 4583 | 5386 | 36896 | 10000 | 10000 | 221530 | 10000 | 10000 |
| Pt_oPe | 7282 | 3321 | 3621 | 36896 | 10000 | 10000 | 221530 | 10000 | 10000 |
| Pt_oe2i | 7282 | 3236 | 3485 | 36896 | 10000 | 10000 | 221530 | 10000 | 10000 |

structures in the split of training, validation and testing. The average answers count of each query structure are reported in Table 10. The number of queries for each dataset is shown in Table 11.

## B.4 Implementation Details

We use the Embedding Reasoning Interpreter as is introduced in Appendix B.2. We implement our model with PyTorch and use Adam [50] as a gradient optimizer. For each experiment, we use one GTX1080 graphic card. The hyperparameters for each dataset are shown in Table 12. Note that we do not perform hyperparameter tuning for each dataset. Instead, we use the same hyperparameters

Table 10: Average answers count for each dataset. All numbers are rounded to two decimal places.

| Query Name | ICEWS14 | | | ICEWS05-15 | | | GDELT-500 | | |
|---|---|---|---|---|---|---|---|---|---|
| | **Train** | **Validate** | **Test** | **Train** | **Validate** | **Test** | **Train** | **Validate** | **Test** |
| **Pe** | 1.09 | 1.01 | 1.01 | 1.07 | 1.01 | 1.01 | 2.07 | 1.21 | 1.21 |
| **Pe2** | 1.03 | 2.19 | 2.23 | 1.02 | 2.15 | 2.19 | 2.61 | 6.51 | 6.13 |
| **Pe3** | 1.04 | 2.25 | 2.29 | 1.02 | 2.18 | 2.21 | 5.11 | 10.86 | 10.70 |
| **e2i** | 1.02 | 2.76 | 2.84 | 1.01 | 2.36 | 2.52 | 1.05 | 2.30 | 2.32 |
| **e3i** | 1.00 | 1.57 | 1.59 | 1.00 | 1.26 | 1.26 | 1.00 | 1.20 | 1.35 |
| **Pt** | 1.71 | 1.22 | 1.21 | 2.58 | 1.61 | 1.60 | 3.36 | 1.66 | 1.66 |
| **aPt** | 177.99 | 176.09 | 175.89 | 2022.16 | 2003.85 | 1998.71 | 156.48 | 155.38 | 153.41 |
| **bPt** | 181.20 | 179.88 | 179.26 | 1929.98 | 1923.75 | 1919.83 | 160.38 | 159.29 | 157.42 |
| **Pe_Pt** | 1.58 | 7.90 | 8.62 | 2.84 | 18.11 | 20.63 | 26.56 | 42.54 | 41.33 |
| **Pt_sPe_Pt** | 1.79 | 7.26 | 7.47 | 2.49 | 13.51 | 10.86 | 4.92 | 14.13 | 12.80 |
| **Pt_oPe_Pt** | 1.75 | 7.27 | 7.48 | 2.55 | 13.01 | 14.34 | 4.62 | 14.47 | 12.90 |
| **t2i** | 1.19 | 6.29 | 6.38 | 3.07 | 29.45 | 25.61 | 1.97 | 8.98 | 7.76 |
| **t3i** | 1.01 | 2.88 | 3.14 | 1.08 | 10.03 | 10.22 | 1.06 | 3.79 | 3.52 |
| **e2i_N** | 1.02 | 2.10 | 2.14 | 1.01 | 2.05 | 2.08 | 2.04 | 4.66 | 4.58 |
| **e3i_N** | 1.00 | 1.00 | 1.00 | 1.00 | 1.00 | 1.00 | 1.02 | 1.19 | 1.37 |
| **Pe_e2i_N** | 1.04 | 2.21 | 2.25 | 1.02 | 2.16 | 2.19 | 3.67 | 8.54 | 8.12 |
| **e2i_PeN** | 1.04 | 2.22 | 2.26 | 1.02 | 2.17 | 2.21 | 3.67 | 8.66 | 8.36 |
| **e2i_NPe** | 1.18 | 3.03 | 3.11 | 1.12 | 2.87 | 2.99 | 4.00 | 8.15 | 7.81 |
| **t2i_N** | 1.15 | 3.31 | 3.44 | 1.21 | 4.06 | 4.20 | 2.91 | 8.78 | 7.56 |
| **t3i_N** | 1.00 | 1.02 | 1.03 | 1.01 | 1.02 | 1.02 | 1.15 | 3.19 | 3.20 |
| **Pe_t2i_N** | 1.08 | 2.59 | 2.70 | 1.08 | 2.47 | 2.62 | 4.10 | 12.02 | 11.37 |
| **t2i_PtN** | 1.41 | 5.22 | 5.47 | 1.70 | 8.10 | 8.11 | 4.56 | 12.56 | 11.32 |
| **t2i_NPt** | 8.14 | 25.96 | 26.23 | 66.99 | 154.01 | 147.34 | 17.58 | 35.60 | 32.22 |
| **e2u** | - | 3.12 | 3.17 | - | 2.38 | 2.40 | - | 5.04 | 5.41 |
| **Pe_e2u** | - | 2.38 | 2.44 | - | 1.24 | 1.25 | - | 9.39 | 10.78 |
| **t2u** | - | 4.35 | 4.53 | - | 5.57 | 5.92 | - | 9.70 | 10.51 |
| **Pe_t2u** | - | 2.72 | 2.83 | - | 1.24 | 1.28 | - | 9.90 | 11.27 |
| **between** | 122.61 | 120.94 | 120.27 | 1407.87 | 1410.39 | 1404.76 | 214.16 | 210.99 | 207.85 |
| **e2i_Pe** | - | 1.00 | 1.00 | - | 1.00 | 1.00 | - | 1.07 | 1.10 |
| **Pe_e2i** | - | 2.18 | 2.24 | - | 1.32 | 1.33 | - | 5.08 | 5.49 |
| **t2i_Pe** | - | 1.03 | 1.03 | - | 1.01 | 1.02 | - | 1.34 | 1.44 |
| **Pe_t2i** | - | 1.14 | 1.16 | - | 1.07 | 1.08 | - | 2.01 | 2.20 |
| **Pe_aPt** | 4.67 | 16.73 | 16.50 | 18.68 | 43.80 | 46.23 | 49.31 | 66.21 | 68.88 |
| **Pe_at2i** | 7.26 | 22.63 | 21.98 | 30.40 | 60.03 | 53.18 | 88.77 | 101.60 | 101.88 |
| **Pt_sPe** | 8.65 | 28.86 | 29.22 | 71.51 | 162.36 | 155.46 | 27.55 | 45.83 | 43.73 |
| **Pt_se2i** | 1.31 | 5.72 | 6.19 | 1.37 | 9.00 | 9.30 | 2.76 | 8.72 | 7.66 |
| **Pe_bPt** | 4.53 | 17.07 | 16.80 | 18.70 | 45.81 | 48.23 | 67.67 | 84.79 | 83.00 |
| **Pe_bt2i** | 7.27 | 21.92 | 21.23 | 30.31 | 61.59 | 64.98 | 88.80 | 100.64 | 100.67 |
| **Pt_oPe** | 1.41 | 5.23 | 5.46 | 1.68 | 8.36 | 8.21 | 3.84 | 11.31 | 10.06 |
| **Pt_oe2i** | 1.32 | 6.51 | 7.00 | 1.44 | 10.49 | 10.89 | 2.55 | 8.17 | 7.27 |

Table 11: Number of queries. Pe represents query answering (s,r,?,t). Pt represents query answering (s,r,o,?). QoE represents the query of entities (except Pe). QoT represents query of timestamps (except Pt). n1p represents a query that is not Pe or Pt.

| Dataset | Training | | | | Validation | | | Test | | |
|---|---|---|---|---|---|---|---|---|---|---|
| | **Pe** | **Pt** | **QoE** | **QoT** | **Pe** | **Pt** | **n1p** | **Pe** | **Pt** | **n1p** |
| ICEWS14 | 273,710 | 27,371 | 59,078 | 8,000 | 66,990 | 66,990 | 10,000 | 66,990 | 66,990 | 10,000 |
| ICEWS05-15 | 149,689 | 14,968 | 20,094 | 5,000 | 66,990 | 22,804 | 10,000 | 66,990 | 66,990 | 10,000 |
| GDELT-500 | 107,982 | 10,798 | 16,910 | 4,000 | 66,990 | 17,021 | 10,000 | 66,990 | 66,990 | 10,000 |

Table 12: Hyperparameters on each dataset. $d$ is the embedding dimension, $b$ is the batch size, $n$ is the negative sampling size, $\gamma$ is the parameter in the loss function, $m$ is the maximum training step, and $l$ is the learning rate.

| Dataset | $d$ | $b$ | $n$ | $\gamma$ | $m$ | $l$ |
|---|---|---|---|---|---|---|
| ICEWS14 | 800 | 512 | 128 | 15 | 300k | $1 \times 10^{-4}$ |
| ICEWS05-15 | 800 | 512 | 128 | 30 | 300k | $1 \times 10^{-4}$ |
| GDELT-500 | 800 | 512 | 128 | 30 | 300k | $1 \times 10^{-4}$ |

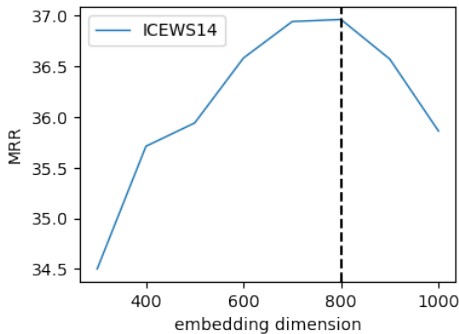

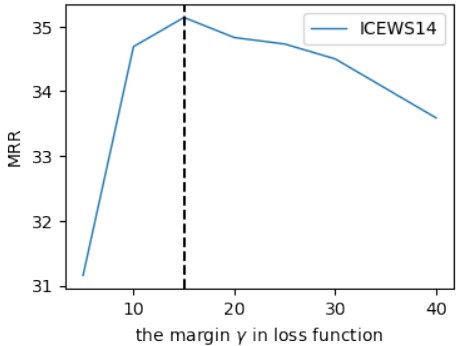

(a) The impact of embedding dimension on MRR.    (b) The impact of the margin $\gamma$ on MRR.

Figure 9: The impact of embedding dimensionality $d$ and the margin $\gamma$ on MRR.

found in sensitive analysis on ICEWS14 (Section B.6) for all datasets. Besides, the source code is available at Gitbub[†] . We cite these projects [‡] [§], and thank them for their great contributions!

## B.5 Evaluation

Given the standard split of edges into training ($\mathcal{F}_{\text{train}}$), validation ($\mathcal{F}_{\text{valid}}$) and test ($\mathcal{F}_{\text{test}}$) sets, we append inverse relations and double the number of edges in the graph. Then we create three graph: $\mathcal{G}_{\text{train}} = \{\mathcal{V}, \mathcal{R}, \mathcal{T}, \mathcal{F}_{\text{train}}\}$, $\mathcal{G}_{\text{valid}} = \{\mathcal{V}, \mathcal{R}, \mathcal{T}, \mathcal{F}_{\text{train}} + \mathcal{F}_{\text{valid}}\}$, $\mathcal{G}_{\text{test}} = \{\mathcal{V}, \mathcal{R}, \mathcal{T}, \mathcal{F}_{\text{train}} + \mathcal{F}_{\text{valid}} + \mathcal{F}_{\text{test}}\}$. Given a query $q$, let $[\![q]\!]_{\text{train}}$, $[\![q]\!]_{\text{valid}}$, and $[\![q]\!]_{\text{test}}$ denote a set of answers (entities or timestamps) obtained by subgraph matching of $q$ on $\mathcal{G}_{\text{train}}$, $\mathcal{G}_{\text{valid}}$ and $\mathcal{G}_{\text{test}}$. For a test query $q$ and its non-trivial answer set $v \in [\![q]\!]_{test} - [\![q]\!]_{valid}$, we denote the rank of each answer $v_i$ as $rank(v_i)$. The mean reciprocal rank (MRR) is $\text{MRR}(q) = \frac{1}{||v||} \sum_{v_i \in v} \frac{1}{rank(v_i)}$ and Hits at K (Hits@K) is $\text{Hits@K}(q) = \frac{1}{||v||} \sum_{v_i \in v} f(rank(v_i))$, where $f(n) = \begin{cases} 1, & n \leq K \\ 0, & n > K \end{cases}$.

## B.6 Experiment Data for Sensitive Analysis

We conduct experiments on ICEWS14 to analyze the impact of the embedding dimensionality $d$ and the margin $\gamma$ on the performance of TFLEX.

**Impacts of Embedding Dimensionality** Our experiments indicate that the selection of embedding dimension has a substantial influence on the effectiveness of TFLEX. We train TFLEX with embedding dimension $d \in \{300, 400, 500, 600, 700, 800, 900, 1000\}$ and plot results based on the validation set, as shown in Figure 9a. With the increase of $d$, the model performance (indicated by MRR) increases rapidly and reaches its top at $d = 800$. Therefore, we assign 800 as the best setting.

---

[†]TFLEX: https://github.com/LinXueyuanStdio/TFLEX

[‡]GQE, Query2box, BetaE: https://github.com/snap-stanford/KGReasoning

[§]ConE: https://github.com/MIRALab-USTC/QE-ConE

Table 13: The mean values and standard variances of TFLEX's MRR results on ICEWS14.

| Pe | Pe2 | Pe3 | e2i | e3i | | | |
|---|---|---|---|---|---|---|---|
| 48.21 | 37.27 | 33.53 | 69.24 | 95.70 | | | |
| ±0.37 | ±0.12 | ±0.42 | ±0.46 | ±0.19 | | | |

| Pt | aPt | bPt | Pe_Pt | Pt_sPe_Pt | Pt_oPe_Pt | t2i | t3i |
|---|---|---|---|---|---|---|---|
| 20.87 | 3.00 | 2.96 | 14.27 | 9.57 | 9.46 | 27.49 | 52.84 |
| ±0.43 | ±0.34 | ±0.50 | ±0.57 | ±0.44 | ±0.66 | ±0.51 | ±0.63 |

| e2i_N | e3i_N | Pe_e2i_N | e2i_PeN | e2i_NPe | | | |
|---|---|---|---|---|---|---|---|
| 45.55 | 99.56 | 34.74 | 35.63 | 38.61 | | | |
| ±0.12 | ±0.50 | ±0.44 | ±0.44 | ±0.20 | | | |

| t2i_N | t3i_N | Pe_t2i_N | t2i_PtN | t2i_NPt | | | |
|---|---|---|---|---|---|---|---|
| 25.38 | 98.91 | 34.05 | 11.42 | 12.07 | | | |
| ±0.60 | ±0.13 | ±0.49 | ±0.57 | ±0.53 | | | |

| e2u | Pe_e2u | | | | | | |
|---|---|---|---|---|---|---|---|
| 29.20 | 42.28 | | | | | | |
| ±0.12 | ±0.43 | | | | | | |

| t2u | Pe_t2u | | | | | | |
|---|---|---|---|---|---|---|---|
| 30.73 | 21.74 | | | | | | |
| ±0.54 | ±0.35 | | | | | | |

| e2i_Pe | Pe_e2i | Pe_aPt | Pe_bPt | Pe_at2i | Pe_bt2i | | |
|---|---|---|---|---|---|---|---|
| 98.86 | 36.77 | 8.66 | 9.74 | 7.90 | 7.78 | | |
| ±0.34 | ±0.42 | ±0.61 | ±0.33 | ±0.39 | ±0.39 | | |

| t2i_Pe | Pe_t2i | Pt_sPe | Pt_oPe | Pt_se2i | Pt_oe2i | between | |
|---|---|---|---|---|---|---|---|
| 96.62 | 64.50 | 4.32 | 10.58 | 8.20 | 7.95 | 2.57 | |
| ±0.13 | ±0.12 | ±0.46 | ±0.20 | ±0.34 | ±0.34 | ±0.14 | |

**Impacts of Parameter** $\gamma$    We train TFLEX with parameter $\gamma \in \{5, 10, 15, 20, 25, 30, 35, 40\}$ and plot MRR results in Figure 9b. Too small and too large $\gamma$ both get bad results, while $\gamma = 15$ in the middle is the best. Therefore, we choose $\gamma = 15$.

**Error Bars of Main Results**    In order to evaluate how stable the performance of TFLEX is, we run five times with random seeds $\{1, 10, 100, 1000, 10000\}$ and report the error bars of these results. Table 13 shows the error bar of TFLEX's MRR results on ICEWS14. Overall, the standard variances are small, which demonstrates that the performance of TFLEX is stable.

### B.7    Detail Metrics of Main Results

We report the MRR results on all query structures for each dataset in Table 14 (ICEWS14), Table 15 (ICEWS05-15) and Table 16 (GDELT-500). Query2box [2] can only answer queries without negation. BetaE [3] and ConE [4] can answer queries with negation, except temporal logic. ConE(temporal) and X(ConE) are the variants of ConE, aiming at exploring the right way to promote the static query embeddings to temporal ones in Section C. FLEX, X(ConE), X-1F, X-logic are the variants of TFLEX, introduced in the main body of the paper.

## C    Exploration of Static Query Embeddings Toward Dynamic

In this section, we present more research questions that aim to explore the right way to promote the static query embeddings to temporal ones. Since ConE [4] is a strong baseline for complex logical reasoning over static knowledge graphs, we conduct exploration experiments by introducing variants of ConE. We present the overall results for each dataset in Table 17, detailed results in Table 14 (ICEWS14), Table 15 (ICEWS05-15) and Table 16 (GDELT-500). Overall, we are interested in the following research questions:

Table 14: MRR results on ICEWS14. The group avg$_x$ wraps to two rows. **AVG** denotes average performance under all query types.

| Model | avg$_e$ | Pe | Pe2 | Pe3 | e2i | e3i | | | |
|---|---|---|---|---|---|---|---|---|---|
| Query2box | 25.06 | 25.81 | 17.29 | 12.97 | 16.00 | 53.25 | | | |
| BetaE | 37.19 | 39.52 | 23.77 | 17.50 | 23.77 | 81.38 | | | |
| ConE | 41.94 | 42.55 | 30.90 | 24.78 | 26.52 | 84.93 | | | |
| ConE(temporal) | 42.23 | 41.21 | 31.83 | 24.99 | 28.98 | 84.17 | | | |
| FLEX | 43.67 | 45.25 | 33.07 | 27.22 | 27.62 | 85.18 | | | |
| TFLEX | 56.79 | 48.21 | 37.27 | 33.53 | 69.24 | 95.70 | | | |
| X(ConE) | 40.93 | 40.58 | 28.84 | 24.31 | 30.93 | 79.98 | | | |
| X-1F | 56.89 | 48.61 | 37.51 | 32.61 | 71.25 | 94.47 | | | |
| X-logic | 56.64 | 48.00 | 37.42 | 31.23 | 73.11 | 93.45 | | | |
| | avg$_t$ | Pt | aPt | bPt | Pe_Pt | Pt_sPe_Pt | Pt_oPe_Pt | t2i | t3i |
| TFLEX | 17.56 | 20.87 | 3.00 | 2.96 | 14.27 | 9.57 | 9.46 | 27.49 | 52.84 |
| X(ConE) | 16.41 | 19.45 | 3.06 | 3.01 | 13.96 | 8.79 | 8.67 | 25.01 | 49.36 |
| X-1F | 18.77 | 21.94 | 3.04 | 2.99 | 16.69 | 10.73 | 10.44 | 29.66 | 54.71 |
| X-logic | 18.03 | 21.33 | 3.10 | 3.05 | 15.76 | 9.85 | 9.94 | 28.57 | 52.63 |
| | avg$_{e,\mathcal{C}_e}$ | e2i_N | e3i_N | Pe_e2i_N | e2i_PeN | e2i_NPe | | | |
| BetaE | 36.69 | 30.44 | 87.93 | 19.94 | 22.80 | 22.36 | | | |
| ConE | 44.88 | 40.98 | 99.12 | 29.22 | 29.84 | 25.22 | | | |
| ConE(temporal) | 44.94 | 41.45 | 98.99 | 29.48 | 30.71 | 24.09 | | | |
| FLEX | 44.41 | 38.30 | 99.22 | 31.18 | 30.49 | 22.84 | | | |
| TFLEX | 50.82 | 45.55 | 99.56 | 34.74 | 35.63 | 38.61 | | | |
| X(ConE) | 42.15 | 39.10 | 96.25 | 26.92 | 27.59 | 20.89 | | | |
| X-1F | 49.78 | 42.31 | 99.31 | 34.89 | 34.77 | 37.61 | | | |
| X-logic | 51.17 | 44.65 | 99.25 | 35.30 | 36.31 | 40.33 | | | |
| | avg$_{t,\mathcal{C}_t}$ | t2i_N | t3i_N | Pe_t2i_N | t2i_PtN | t2i_NPt | | | |
| TFLEX | 36.37 | 25.38 | 98.91 | 34.05 | 11.42 | 12.07 | | | |
| X(ConE) | 35.24 | 26.90 | 98.82 | 30.30 | 9.44 | 10.75 | | | |
| X-1F | 37.73 | 26.40 | 98.82 | 37.85 | 12.38 | 13.23 | | | |
| X-logic | 36.39 | 25.68 | 98.69 | 33.30 | 11.34 | 12.93 | | | |
| | avg$_{\{\mathcal{U}_e\}}$ | e2u | Pe_e2u | | | | | | |
| BetaE | 19.95 | 18.61 | 21.28 | | | | | | |
| ConE | 26.47 | 21.84 | 31.11 | | | | | | |
| ConE(temporal) | 27.63 | 23.01 | 32.26 | | | | | | |
| FLEX | 29.25 | 24.05 | 34.46 | | | | | | |
| TFLEX | 35.74 | 29.20 | 42.28 | | | | | | |
| X(ConE) | 25.46 | 20.26 | 30.67 | | | | | | |
| X-1F | 34.48 | 30.04 | 38.93 | | | | | | |
| X-logic | 34.68 | 29.44 | 39.92 | | | | | | |
| | avg$_{\{\mathcal{U}_t\}}$ | t2u | Pe_t2u | | | | | | |
| TFLEX | 26.24 | 30.73 | 21.74 | | | | | | |
| X(ConE) | 24.07 | 27.63 | 20.51 | | | | | | |
| X-1F | 28.04 | 33.91 | 22.16 | | | | | | |
| X-logic | 26.36 | 31.21 | 21.52 | | | | | | |
| | avg$_x$ | e2i_Pe | Pe_e2i | Pe_aPt | Pe_bPt | Pe_at2i | Pe_bt2i | | |
| TFLEX | 28.03 | 98.86 | 36.77 | 8.66 | 9.74 | 7.90 | 7.78 | | |
| X(ConE) | 26.65 | 87.31 | 28.72 | 11.01 | 10.40 | 11.21 | 11.27 | | |
| X-1F | 29.31 | 98.87 | 36.00 | 10.04 | 10.21 | 8.30 | 8.06 | | |
| X-logic | 28.61 | 97.64 | 36.96 | 10.09 | 10.27 | 8.76 | 8.93 | | |
| | **AVG** | t2i_Pe | Pe_t2i | Pt_sPe | Pt_oPe | Pt_se2i | Pt_oe2i | between | |
| TFLEX | 35.93 | 96.62 | 64.50 | 4.32 | 10.58 | 8.20 | 7.95 | 2.57 | |
| X(ConE) | 30.13 | 95.40 | 63.04 | 3.77 | 8.82 | 6.44 | 5.76 | 3.32 | |
| X-1F | 36.43 | 97.24 | 70.32 | 4.88 | 10.73 | 12.24 | 11.55 | 2.55 | |
| X-logic | 35.98 | 96.35 | 64.31 | 4.97 | 10.35 | 10.59 | 10.35 | 2.34 | |

Table 15: MRR results on ICEWS05-15. The group $\text{avg}_x$ wraps to two rows. **AVG** denotes average performance under all query types.

| Model | $\text{avg}_e$ | Pe | Pe2 | Pe3 | e2i | e3i | | |
|---|---|---|---|---|---|---|---|---|
| Query2box | 24.00 | 25.94 | 16.62 | 14.22 | 17.68 | 45.52 | | |
| BetaE | 31.33 | 35.78 | 21.47 | 18.18 | 18.10 | 63.11 | | |
| ConE | 40.93 | 42.67 | 29.39 | 24.79 | 26.15 | 81.64 | | |
| ConE(temporal) | 40.74 | 42.64 | 29.30 | 24.76 | 25.35 | 81.65 | | |
| FLEX | 38.96 | 41.60 | 29.80 | 24.58 | 24.37 | 74.46 | | |
| TFLEX | 48.99 | 43.04 | 36.28 | 33.89 | 41.17 | 90.60 | | |
| X(ConE) | 36.29 | 39.90 | 26.62 | 22.85 | 22.52 | 69.56 | | |
| X-1F | 49.90 | 43.07 | 36.87 | 34.96 | 40.85 | 93.76 | | |
| X-logic | 44.80 | 40.57 | 31.47 | 30.67 | 35.80 | 85.47 | | |

| | $\text{avg}_t$ | Pt | aPt | bPt | Pe_Pt | Pt_sPe_Pt | Pt_oPe_Pt | t2i | t3i |
|---|---|---|---|---|---|---|---|---|---|
| TFLEX | 4.39 | 10.62 | 0.38 | 0.38 | 7.29 | 2.63 | 1.83 | 3.85 | 8.15 |
| X(ConE) | 4.41 | 10.98 | 0.38 | 0.38 | 6.75 | 2.16 | 1.63 | 4.22 | 8.81 |
| X-1F | 4.43 | 10.71 | 0.38 | 0.38 | 7.40 | 2.73 | 1.86 | 3.80 | 8.16 |
| X-logic | 3.29 | 8.57 | 0.38 | 0.39 | 6.37 | 1.69 | 1.41 | 2.49 | 5.06 |

| | $\text{avg}_{e,\mathcal{C}_e}$ | e2i_N | e3i_N | Pe_e2i_N | e2i_PeN | e2i_NPe | | |
|---|---|---|---|---|---|---|---|---|
| BetaE | 29.70 | 21.68 | 71.08 | 20.31 | 21.11 | 14.30 | | |
| ConE | 43.52 | 43.04 | 96.08 | 28.41 | 28.75 | 21.30 | | |
| ConE(temporal) | 43.34 | 42.85 | 96.08 | 28.33 | 28.62 | 20.83 | | |
| FLEX | 42.10 | 36.22 | 97.47 | 27.93 | 27.23 | 21.64 | | |
| TFLEX | 46.17 | 41.34 | 96.69 | 34.29 | 34.63 | 23.88 | | |
| X(ConE) | 38.12 | 33.70 | 88.83 | 24.80 | 25.60 | 17.64 | | |
| X-1F | 46.11 | 38.06 | 96.82 | 35.76 | 36.20 | 23.73 | | |
| X-logic | 41.92 | 36.57 | 91.16 | 30.37 | 31.42 | 20.07 | | |

| | $\text{avg}_{t,\mathcal{C}_t}$ | t2i_N | t3i_N | Pe_t2i_N | t2i_PtN | t2i_NPt | | |
|---|---|---|---|---|---|---|---|---|
| TFLEX | 30.16 | 16.09 | 98.40 | 31.39 | 3.35 | 1.58 | | |
| X(ConE) | 29.49 | 16.55 | 98.46 | 28.23 | 2.53 | 1.69 | | |
| X-1F | 30.26 | 16.05 | 98.37 | 32.07 | 3.27 | 1.56 | | |
| X-logic | 28.34 | 12.94 | 96.03 | 29.47 | 2.18 | 1.07 | | |

| | $\text{avg}_{\{\mathcal{U}_e\}}$ | e2u | Pe_e2u | | | | | |
|---|---|---|---|---|---|---|---|---|
| BetaE | 21.54 | 20.95 | 22.13 | | | | | |
| ConE | 43.02 | 37.21 | 48.83 | | | | | |
| ConE(temporal) | 43.14 | 37.05 | 49.24 | | | | | |
| FLEX | 44.38 | 35.72 | 53.04 | | | | | |
| TFLEX | 54.37 | 52.99 | 55.75 | | | | | |
| X(ConE) | 36.37 | 29.89 | 42.86 | | | | | |
| X-1F | 54.05 | 52.47 | 55.64 | | | | | |
| X-logic | 45.36 | 43.84 | 46.88 | | | | | |

| | $\text{avg}_{\{\mathcal{U}_t\}}$ | t2u | Pe_t2u | | | | | |
|---|---|---|---|---|---|---|---|---|
| TFLEX | 28.69 | 44.99 | 12.39 | | | | | |
| X(ConE) | 26.40 | 40.35 | 12.46 | | | | | |
| X-1F | 27.70 | 43.38 | 12.02 | | | | | |
| X-logic | 23.39 | 37.07 | 9.71 | | | | | |

| | $\text{avg}_x$ | e2i_Pe | Pe_e2i | Pe_aPt | Pe_bPt | Pe_at2i | Pe_bt2i | |
|---|---|---|---|---|---|---|---|---|
| TFLEX | 24.26 | 94.23 | 57.16 | 5.07 | 4.56 | 4.38 | 3.93 | |
| X(ConE) | 21.69 | 81.36 | 38.20 | 5.63 | 5.03 | 5.80 | 5.36 | |
| X-1F | 24.41 | 94.26 | 55.97 | 5.14 | 4.79 | 4.37 | 3.95 | |
| X-logic | 21.95 | 87.48 | 50.11 | 4.28 | 3.95 | 3.96 | 3.55 | |
| | **AVG** | t2i_Pe | Pe_t2i | Pt_sPe | Pt_oPe | Pt_se2i | Pt_oe2i | between |
| TFLEX | 33.72 | 92.25 | 48.35 | 0.46 | 2.82 | 0.98 | 1.02 | 0.22 |
| X(ConE) | 27.54 | 91.92 | 43.83 | 0.58 | 2.18 | 0.95 | 1.02 | 0.14 |
| X-1F | 33.98 | 92.21 | 50.77 | 0.46 | 2.89 | 1.13 | 1.23 | 0.17 |
| X-logic | 29.86 | 86.17 | 40.78 | 0.62 | 1.98 | 1.10 | 1.21 | 0.16 |

Table 16: MRR results on GDELT-500. The group $\mathrm{avg}_x$ wraps to two rows. **AVG** denotes average performance under all query types.

| Model | $\mathrm{avg}_e$ | Pe | Pe2 | Pe3 | e2i | e3i |
|---|---|---|---|---|---|---|
| Query2box | 9.67 | 10.70 | 4.73 | 3.41 | 12.28 | 17.22 |
| BetaE | 14.75 | 14.02 | 5.76 | 4.71 | 18.61 | 30.67 |
| ConE | 18.44 | 16.65 | 6.18 | 4.70 | 23.11 | 41.55 |
| ConE(temporal) | 18.51 | 17.61 | 5.62 | 4.57 | 24.52 | 40.22 |
| FLEX | 19.07 | 17.72 | 5.78 | 4.67 | 24.30 | 42.85 |
| TFLEX | 19.60 | 18.50 | 5.76 | 4.68 | 25.94 | 43.14 |
| X(ConE) | 17.83 | 17.05 | 5.76 | 4.66 | 24.02 | 37.66 |
| X-1F | 17.92 | 16.18 | 6.35 | 4.84 | 22.41 | 39.80 |
| X-logic | 17.36 | 15.80 | 6.22 | 4.85 | 22.45 | 37.47 |

| | $\mathrm{avg}_t$ | Pt | aPt | bPt | Pe_Pt | Pt_sPe_Pt | Pt_oPe_Pt | t2i | t3i |
|---|---|---|---|---|---|---|---|---|---|
| TFLEX | 5.38 | 6.49 | 3.25 | 3.27 | 6.16 | 3.06 | 2.88 | 6.78 | 11.17 |
| X(ConE) | 3.16 | 2.15 | 3.39 | 3.33 | 6.96 | 2.29 | 2.31 | 2.34 | 2.49 |
| X-1F | 5.49 | 6.62 | 3.30 | 3.30 | 5.84 | 3.07 | 3.05 | 7.08 | 11.66 |
| X-logic | 5.75 | 7.15 | 3.34 | 3.31 | 5.95 | 3.14 | 3.06 | 7.52 | 12.54 |

| | $\mathrm{avg}_{e,\mathcal{C}_e}$ | e2i_N | e3i_N | Pe_e2i_N | e2i_PeN | e2i_NPe |
|---|---|---|---|---|---|---|
| BetaE | 11.15 | 11.22 | 24.17 | 5.53 | 5.46 | 9.39 |
| ConE | 12.67 | 12.69 | 29.09 | 5.47 | 5.42 | 10.66 |
| ConE(temporal) | 12.67 | 13.42 | 27.98 | 5.31 | 5.00 | 11.63 |
| FLEX | 13.35 | 13.57 | 30.25 | 5.47 | 5.47 | 11.99 |
| TFLEX | 13.52 | 14.46 | 30.09 | 5.51 | 5.19 | 12.36 |
| X(ConE) | 12.34 | 13.42 | 26.28 | 5.54 | 5.35 | 11.14 |
| X-1F | 12.13 | 11.77 | 27.63 | 5.70 | 5.40 | 10.14 |
| X-logic | 12.11 | 12.32 | 27.00 | 5.70 | 5.55 | 10.01 |

| | $\mathrm{avg}_{t,\mathcal{C}_t}$ | t2i_N | t3i_N | Pe_t2i_N | t2i_PtN | t2i_NPt |
|---|---|---|---|---|---|---|
| TFLEX | 6.31 | 5.92 | 9.01 | 9.30 | 2.76 | 4.56 |
| X(ConE) | 3.93 | 2.24 | 2.43 | 10.00 | 2.26 | 2.74 |
| X-1F | 6.50 | 5.90 | 9.70 | 9.15 | 2.85 | 4.90 |
| X-logic | 6.86 | 6.51 | 10.94 | 8.78 | 2.94 | 5.14 |

| | $\mathrm{avg}_{\{\mathcal{U}_e\}}$ | e2u | Pe_e2u |
|---|---|---|---|
| BetaE | 6.20 | 4.64 | 7.75 |
| ConE | 6.96 | 4.68 | 9.25 |
| ConE(temporal) | 7.30 | 4.50 | 10.09 |
| FLEX | 7.44 | 4.47 | 10.42 |
| TFLEX | 7.58 | 4.62 | 10.55 |
| X(ConE) | 7.41 | 4.64 | 10.19 |
| X-1F | 6.92 | 4.80 | 9.04 |
| X-logic | 6.91 | 4.74 | 9.09 |

| | $\mathrm{avg}_{\{\mathcal{U}_t\}}$ | t2u | Pe_t2u |
|---|---|---|---|
| TFLEX | 6.71 | 9.03 | 4.38 |
| X(ConE) | 6.35 | 10.39 | 2.31 |
| X-1F | 6.59 | 8.60 | 4.58 |
| X-logic | 6.80 | 8.85 | 4.76 |

| | $\mathrm{avg}_x$ | e2i_Pe | Pe_e2i | Pe_aPt | Pe_bPt | Pe_at2i | Pe_bt2i |
|---|---|---|---|---|---|---|---|
| TFLEX | 6.17 | 17.53 | 6.38 | 4.73 | 4.58 | 4.47 | 4.58 |
| X(ConE) | 6.17 | 17.53 | 6.44 | 6.45 | 6.18 | 6.02 | 5.97 |
| X-1F | 6.47 | 18.13 | 6.82 | 5.48 | 5.35 | 5.19 | 5.31 |
| X-logic | 6.64 | 17.45 | 6.79 | 5.56 | 5.44 | 5.35 | 5.37 |

| | AVG | t2i_Pe | Pe_t2i | Pt_sPe | Pt_oPe | Pt_se2i | Pt_oe2i | between |
|---|---|---|---|---|---|---|---|---|
| TFLEX | 9.32 | 8.36 | 15.83 | 3.11 | 2.81 | 2.93 | 3.07 | 1.85 |
| X(ConE) | 8.17 | 2.45 | 17.97 | 2.70 | 2.24 | 2.20 | 2.29 | 1.76 |
| X-1F | 8.86 | 8.80 | 15.13 | 2.92 | 2.72 | 3.11 | 3.29 | 1.87 |
| X-logic | 8.92 | 10.28 | 15.28 | 3.27 | 3.11 | 3.23 | 3.36 | 1.79 |

**RQ1**: How about incorporating temporal information into the static query embedding, without considering time logic?

**RQ2**: Further, how about enhancing the static query embedding with temporal logic?

To address research question (**RQ1**), we propose a variant of ConE called **ConE(temporal)** that incorporates temporal information into the static query embedding. The only difference between ConE and ConE(temporal) is the projection operator used. In ConE(temporal), the projection operator incorporates temporal information through a technique called TTransE, which uses the temporal embedding of timestamps. In terms of formulation, the projection operator for ConE(temporal) is given by $\mathcal{P}_e(\mathbf{V}_q, \mathbf{r}, \mathbf{V}_t) = g(\mathbf{MLP}(\mathbf{V}_q + \mathbf{r} + \mathbf{V}_t))$, while for ConE it is given by $\mathcal{P}_e(\mathbf{V}_q, \mathbf{r}, \mathbf{V}_t) = g(\mathbf{MLP}(\mathbf{V}_q + \mathbf{r}))$. Note that $\mathbf{V}_q, \mathbf{r}, \mathbf{V}_t$ are cone embeddings.

From Table 17, we observe that there is slightly improvement in terms of MRR when comparing ConE(temporal) with ConE. It shows that the temporal information can help to improve the performance of static query embedding on answering entities over TKGs. However, the improvement is not significant, which indicates that simply aware of the temporal information is not enough to promote the static query embedding to efficient temporal one.

Next, we wonder if it could help when we utilize more queries involving time logic, which provides more temporal information. Therefore, we address research question (**RQ2**) by proposing a variant of ConE and TFLEX, called **X(ConE)**, that enhances the static query embedding with temporal logic. **X(ConE)** replaces the entity part of temporal feature-logic embedding with ConE, as is introduced in Section 5.2. In the other view, **X(ConE)** is based on ConE, using cone embedding concatenated with the time part of temporal feature-logic embedding, which can handle temporal logic.

From Table 17, to our surprise, **X(ConE)** performs worse than ConE and TFLEX. The concatenation of cone embedding and feature-logic embedding does not help to improve the overall performance of temporal complex reasoning. This is because cone embedding is geometric, while temporal feature-logic embedding is fuzzy. We think there is a mismatch or semantic conflict between the two embeddings, which leads to the poor performance of **X(ConE)**.

Be aware that we do not deny the importance of temporal logic. In fact, temporal logic is very important for temporal complex reasoning (See **Ablation on time part** in Section 5.2). However, we think that it is not a good idea to promote the static query embedding to temporal one by simply concatenating the geometric embedding with the fuzzy embedding. The right way to promote remains an open question.

Table 17: Average MRR results for TFLEX and the variants of ConE. $\mathbf{AVG}_e$ denotes average of $\text{avg}_e$, $\text{avg}_{e,\mathcal{C}_e}$ and $\text{avg}_{\mathcal{U}_e}$. **AVG** denotes average of all groups.

| Dataset | Model | $\text{avg}_e$ | $\text{avg}_{e,\mathcal{C}_e}$ | $\text{avg}_{\mathcal{U}_e}$ | $\text{avg}_t$ | $\text{avg}_{t,\mathcal{C}_t}$ | $\text{avg}_{\mathcal{U}_t}$ | $\text{avg}_x$ | $\mathbf{AVG}_e$ | **AVG** |
|---|---|---|---|---|---|---|---|---|---|---|
| ICEWS14 | ConE | 41.94 | 44.88 | 26.47 | | | | | 37.76 | |
| | ConE(temporal) | 42.23 | 44.94 | 27.63 | | | | | 38.27 | |
| | X(ConE) | 40.93 | 42.15 | 25.46 | 16.41 | 35.24 | 24.07 | 26.65 | 36.18 | 30.13 |
| | TFLEX | 56.79 | 50.82 | 35.74 | 17.56 | 36.37 | 26.24 | 28.03 | 47.78 | 35.93 |
| ICEWS05-15 | ConE | 40.93 | 43.52 | 43.02 | | | | | 42.49 | |
| | ConE(temporal) | 40.74 | 43.34 | 43.14 | | | | | 42.41 | |
| | X(ConE) | 36.29 | 38.12 | 36.37 | 4.41 | 29.49 | 26.40 | 21.69 | 36.93 | 27.54 |
| | TFLEX | 48.99 | 46.17 | 54.37 | 4.39 | 30.16 | 28.69 | 24.26 | 49.84 | 33.72 |
| GDELT-500 | ConE | 18.44 | 12.67 | 6.96 | | | | | 12.69 | |
| | ConE(temporal) | 18.51 | 12.67 | 7.30 | | | | | 12.83 | |
| | X(ConE) | 17.83 | 12.34 | 7.41 | 3.16 | 3.93 | 6.35 | 6.17 | 12.53 | 8.17 |
| | TFLEX | 19.60 | 13.52 | 7.58 | 5.38 | 6.31 | 6.71 | 6.17 | 13.57 | 9.32 |

# D   Visualization of Semantic Changes by Neural Temporal Operators

In this section, we present more visualization of the semantic changes of temporal operators in TFLEX. The examples are randomly selected in the test set of ICEWS14. The visualization is shown in Figure 10, Figure 11 and Figure 12. We attach the anchors and the ground-truth answers in the corresponding caption of figures. From the figures, we observe that the model correctly ranks the hard answers at top when answering query **Pt**. It can also be seen from the figure that the semantic changes of temporal operators **aPt** and **bPt** are intuitive and reasonable.

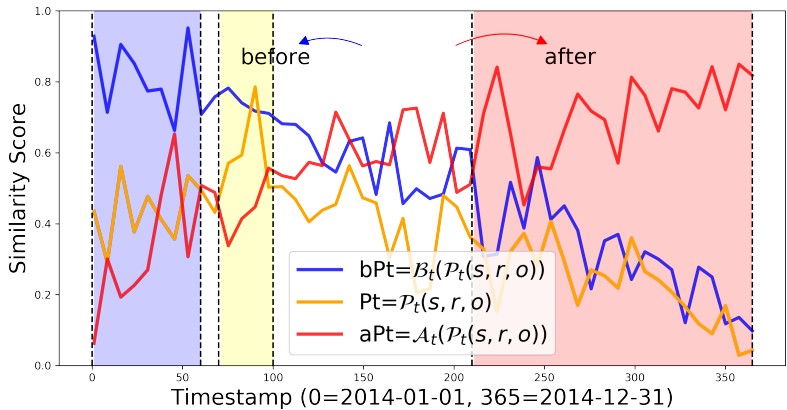

Figure 10: Score distributions of **Pt**, **bPt** and **aPt**, where $s$ is **Citizen (India)**, $r$ is **Criticize or denounce**, $o$ is **Sadhu (India)**, ground-truth answer of **Pt** is **2014-03-27**.

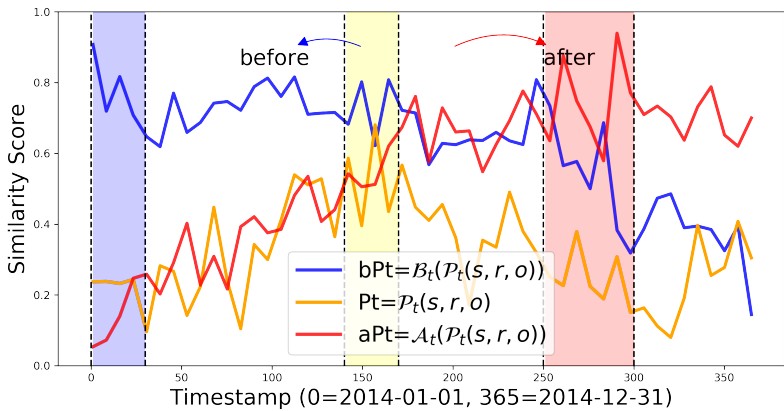

Figure 11: Score distributions of **Pt**, **bPt** and **aPt**, where $s$ is **Citizen (India)**, $r$ is **Use unconventional violence**, $o$ is **Chief Secretary Chandra**, ground-truth answer of **Pt** is **2014-05-11**.

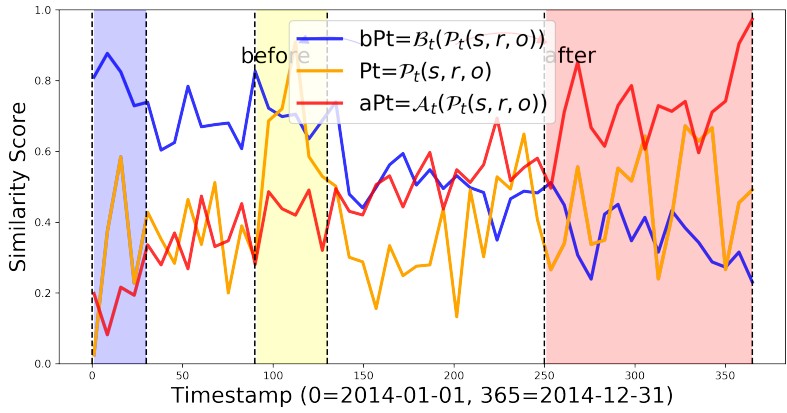

Figure 12: Score distributions of **Pt**, **bPt** and **aPt**, where $s$ is **Mohammad Javad Zarif**, $r$ is **Appeal for economic cooperation**, $o$ is **United Arab Emirates**, ground-truth answer of **Pt** is **2014-04-14**.

# E  Explaining Answers with Temporal Feature-Logic Framework

In this section, we present the explanation of answers given by TFLEX. Below we take the query "**e2i**" as an example. The temporal query "**e2i**" is defined as follows: $q[V_?] = V_?, \exists V_a, V_b, r_1(e_1, V_a, t_1) \wedge r_2(e_2, V_b, t_2)$. Let's consider the specific query: "Who was consulted by Mohammad Javad Zarif on 2014-04-07 and consulted Mohammad Javad Zarif on 2014-04-07?" In this example, $e_1 =$ "Mohammad Javad Zarif", $r_1 =$ "consulted by", $t_1 =$ "2014-04-07", $r_2 =$ "consulted", and $t_2 =$ "2014-04-07". We use TFLEX to execute the query and obtain the answers. The answers are categorized into easy, hard, and wrong ones. Table 18 presents the five most likely answers and their rankings for interpretation. From the table, we observe that TFLEX ranks the answers "Mohammad Javad Zarif", "Sebastian Kurz", "China" and "Catherine Ashton" high, while ranking the wrong answer "Iran" low. TFLEX even infers a hard answer "Catherine Ashton" with higher score than two easy answers "Sebastian Kurz" and "China". This demonstrates that TFLEX successfully identifies the correct answers using complex reasoning and distinguishes them from the wrong ones.

Navigation: Pt(13), Pe(14), Pe2(15), Pe3(16), Pe_Pt(17), e2i(18), e3i(19), e2i_Pe(20), Pe_e2i(21), Pe_t2i(22), t2i(23), t3i(24), t2i_Pe(25), Pt_se2i(26), Pt_oe2i(27), t2i_NPt(28), t2i_PtN(29), Pe_t2i_PtPe_NPt(30), e2i_NPe(31), e2i_PeN(32), Pe_e2i_Pe_NPe(33), e2i_N(34), e3i_N(35), e2u(36), Pe_e2u(37), Pt_sPe(38), Pt_oPe(39), t2i_N(40), t3i_N(41), t2u(42), Pe_t2u(43), Pe_aPt(44), Pe_bPt(45), Pe_at2i(46), Pe_bt2i(47), between(48).

| | Query Sentence | When did North Atlantic Treaty Organization consult Taavi Rõivas? | | |
|---|---|---|---|---|
| **Pt**  $Pt(e_1, r_1, e_2)$ | **Query DNF** | $q[V_?] = V_?, r_1(e_1, e_2, V_?)$ | | |
| | **Rank** | **Query Answers** | **Correctness** | **Answer Type** |
| $e_1$ : North Atlantic Treaty Organization | 1 | 2014-11-20 | ✔ | Hard |
| $r_1$ : Consult | 2 | 2014-12-03 | ✘ | - |
| $e_2$ : Taavi Rõivas | 3 | 2014-06-16 | ✘ | - |
| | 4 | 2014-06-25 | ✘ | - |
| | 5 | 2014-10-13 | ✘ | - |

Figure 13: Intermediate variable assignments and ranks for example Pt query. Correctness indicates whether the answer belongs to the ground-truth set of answers.

| | Query Sentence | Philippines denounced or criticized who on 2014-04-01? | | |
|---|---|---|---|---|
| **Pe**  $Pe(e_1, r_1, t_1)$ | **Query DNF** | $q[V_?] = V_?, r_1(e_1, V_?, t_1)$ | | |
| | **Rank** | **Query Answers** | **Correctness** | **Answer Type** |
| $e_1$ : Philippines | 1 | China | ✔ | Hard |
| $r_1$ : Criticize or denounce | 2 | Japan | ✘ | - |
| $t_1$ : 2014-04-01 | 3 | Malaysia | ✘ | - |
| | 4 | Philippines | ✘ | - |
| | 5 | Iran | ✘ | - |

Figure 14: Intermediate variable assignments and ranks for example Pe query. Correctness indicates whether the answer belongs to the ground-truth set of answers.

| | Query Sentence | On 2014-04-04, who consulted the man who was appealed to or requested by the Head of Government (Latvia) on 2014-08-01? | | |
|---|---|---|---|---|
| **Pe2**  $Pe(Pe(e_1, r_1, t_1), r_2, t_2)$ | **Temporal Query** | $q[V_?] = V_?, \exists V_a, r_1(e_1, V_a, t_1) \wedge r_2(V_a, V_?, t_2)$ | | |
| | **Rank** | **Query Answers** | **Correctness** | **Answer Type** |
| $e_1$ : Head of Government (Latvia) | 1 | François Hollande | ✔ | Easy |
| $r_1$ : Make an appeal or request | 2 | Taavi Rõivas | ✔ | Easy |
| $t_1$ : 2014-08-01 | 3 | Jyrki Katainen | ✔ | Hard |
| $r_2$ : $consult^{-1}$ | 4 | Angela Merkel | ✘ | - |
| $t_2$ : 2014-04-04 | 5 | Head of Government (Latvia) | ✘ | - |

Figure 15: Intermediate variable assignments and ranks for example Pe2 query. Correctness indicates whether the answer belongs to the ground-truth set of answers.

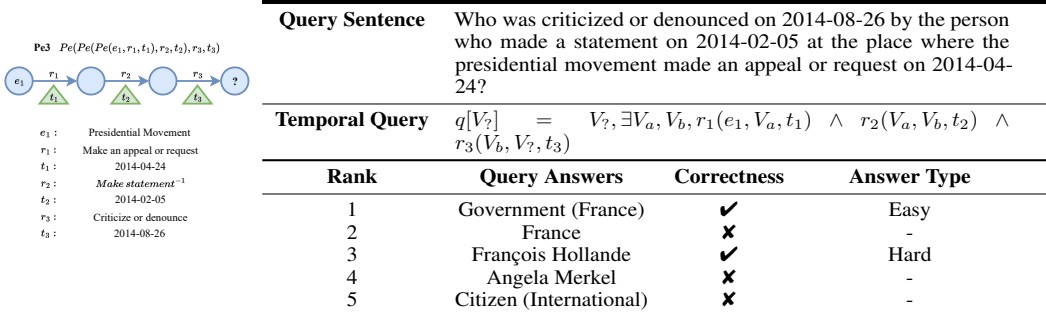

**Pe3**  $Pe(Pe(Pe(e_1,r_1,t_1),r_2,t_2),r_3,t_3)$

$e_1$ :  Presidential Movement
$r_1$ :  Make an appeal or request
$t_1$ :  2014-04-24
$r_2$ :  $Make\ statement^{-1}$
$t_2$ :  2014-02-05
$r_3$ :  Criticize or denounce
$t_3$ :  2014-08-26

| | |
|---|---|
| **Query Sentence** | Who was criticized or denounced on 2014-08-26 by the person who made a statement on 2014-02-05 at the place where the presidential movement made an appeal or request on 2014-04-24? |
| **Temporal Query** | $q[V_?] = V_?, \exists V_a, V_b, r_1(e_1, V_a, t_1) \wedge r_2(V_a, V_b, t_2) \wedge r_3(V_b, V_?, t_3)$ |

| Rank | Query Answers | Correctness | Answer Type |
|---|---|---|---|
| 1 | Government (France) | ✔ | Easy |
| 2 | France | ✘ | - |
| 3 | François Hollande | ✔ | Hard |
| 4 | Angela Merkel | ✘ | - |
| 5 | Citizen (International) | ✘ | - |

Figure 16: Intermediate variable assignments and ranks for example Pe3 query. Correctness indicates whether the answer belongs to the ground-truth set of answers.

**Pe_Pt**  $Pe(e_1, r_1, before(Pt(e_2, r_2, e_3)))$

$e_1$ :  Citizen (Thailand)
$r_1$ :  $Arrest,\ detain,\ or\ charge\ with\ legal\ action^{-1}$
$e_2$ :  Police (Greece)
$r_2$ :  Arrest, detain, or charge with legal action
$e_3$ :  Illegal Immigrant (Greece)

| | |
|---|---|
| **Query Sentence** | Who arrested, detained, or charged the citizen (Thailand) with legal action after Police (Greece) arrested, detained, or charged the illegal immigrant (Greece) with legal action? |
| **Temporal Query** | $q[V_?] = V_?, \exists T_a, r_1(e_1, V_?, T_a) \wedge r_2(e_2, e_3, T_a)$ |

| Rank | Query Answers | Correctness | Answer Type |
|---|---|---|---|
| 1 | Military (Thailand) | ✔ | Easy |
| 2 | Municipal Court (Thailand) | ✔ | Easy |
| 3 | Thailand | ✔ | Hard |
| 4 | National Council for Peace and Order of Thailand | ✘ | - |
| 5 | Police (Cambodia) | ✘ | - |

Figure 17: Intermediate variable assignments and ranks for example Pe_Pt query. Correctness indicates whether the answer belongs to the ground-truth set of answers.

**e2i**  $And(Pe(e_1, r_1, t_1), Pe(e_2, r_2, t_2))$

$e_1$ :  Mohammad Javad Zarif
$r_1$ :  Consult
$t_1$ :  2014-04-07
$e_2$ :  Mohammad Javad Zarif
$r_2$ :  $Consult^{-1}$
$t_2$ :  2014-04-07

| | |
|---|---|
| **Query Sentence** | Who was consulted by Mohammad Javad Zarif on 2014-04-07 and consulted Mohammad Javad Zarif on 2014-04-07? |
| **Temporal Query** | $q[V_?] = V_?, r_1(e_1, V_?, t_1) \wedge r_2(e_2, V_?, t_2))$ |

| Rank | Query Answers | Correctness | Answer Type |
|---|---|---|---|
| 1 | Mohammad Javad Zarif | ✔ | Easy |
| 2 | Catherine Ashton | ✔ | Hard |
| 3 | Sebastian Kurz | ✔ | Easy |
| 4 | China | ✔ | Easy |
| 5 | Iran | ✘ | - |

Figure 18: Intermediate variable assignments and ranks for example e2i query. Correctness indicates whether the answer belongs to the ground-truth set of answers.

**e3i**  $And3(Pe(e_1, r_1, t_1), Pe(e_2, r_2, t_2), Pe(e_3, r_3, t_3))$

$e_1$ :  Joseph Robinette Biden
$r_1$ :  $Discuss\ by\ telephone^{-1}$
$t_1$ :  2014-10-05
$e_2$ :  Joseph Robinette Biden
$r_2$ :  Discuss by telephone
$t_2$ :  2014-10-05
$e_3$ :  Saud bin Faisal bin Abdul-Aziz
$r_3$ :  Make a visit
$t_3$ :  2014-01-20

| | |
|---|---|
| **Query Sentence** | Where was Joseph Robinette Biden discussed by telephone on 2014-10-05 and visited by Saud bin Faisal bin Abdul-Aziz on 2014-01-20? |
| **Temporal Query** | $q[V_?] = V_?, r_1(e_1, V_?, t_1) \wedge r_2(e_2, V_?, t_2) \wedge r_3(e_3, V_?, t_3)$ |

| Rank | Query Answers | Correctness | Answer Type |
|---|---|---|---|
| 1 | UAE Armed Forces | ✔ | Easy |
| 2 | United Arab Emirates | ✔ | Hard |
| 3 | Middle East | ✘ | - |
| 4 | Morocco | ✘ | - |
| 5 | Abdel Fattah Al-Sisi | ✘ | - |

Figure 19: Intermediate variable assignments and ranks for example e3i query. Correctness indicates whether the answer belongs to the ground-truth set of answers.

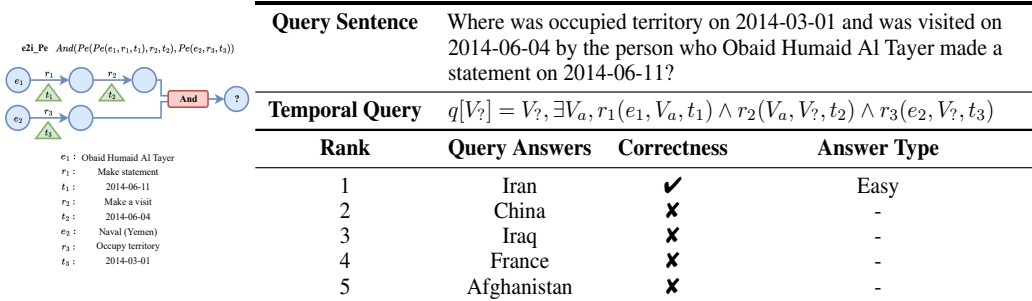

**e2i_Pe** $And(Pe(Pe(e_1, r_1, t_1), r_2, t_2), Pe(e_2, r_3, t_3))$

$e_1$ : Obaid Humaid Al Tayer
$r_1$ : Make statement
$t_1$ : 2014-06-11
$r_2$ : Make a visit
$t_2$ : 2014-06-04
$e_2$ : Naval (Yemen)
$r_3$ : Occupy territory
$t_3$ : 2014-03-01

| Query Sentence | Where was occupied territory on 2014-03-01 and was visited on 2014-06-04 by the person who Obaid Humaid Al Tayer made a statement on 2014-06-11? |
|---|---|
| Temporal Query | $q[V_?] = V_?, \exists V_a, r_1(e_1, V_a, t_1) \wedge r_2(V_a, V_?, t_2) \wedge r_3(e_2, V_?, t_3)$ |

| Rank | Query Answers | Correctness | Answer Type |
|---|---|---|---|
| 1 | Iran | ✔ | Easy |
| 2 | China | ✘ | - |
| 3 | Iraq | ✘ | - |
| 4 | France | ✘ | - |
| 5 | Afghanistan | ✘ | - |

Figure 20: Intermediate variable assignments and ranks for example e2i_Pe query. Correctness indicates whether the answer belongs to the ground-truth set of answers.

**Pe_e2i** $Pe(e2i(e_1, r_1, t_1, e_2, r_2, t_2), r_3, t_3)$

$e_1$ : Labor Union (Croatia)
$r_1$ : Make an appeal or request
$t_1$ : 2014-02-03
$e_2$ : Government (Croatia)
$r_2$ : $Consult^{-1}$
$t_2$ : 2014-03-25
$r_3$ : Consult
$t_3$ : 2014-03-25

| Query Sentence | Where was consulted on 2014-03-25 by the place where Labor Union (Croatia) made an appeal or request on 2014-02-03 and was consulted by Government (Croatia) on 2014-03-25? |
|---|---|
| Temporal Query | $q[V_?] = V_?, \exists V_a, r_1(e_1, V_a, t_1) \wedge r_2(e_2, V_a, t_2) \wedge r_3(V_a, V_?, T_3)$ |

| Rank | Query Answers | Correctness | Answer Type |
|---|---|---|---|
| 1 | Labor Union (Croatia) | ✔ | Hard |
| 2 | Business (Croatia) | ✘ | - |
| 3 | Government (Croatia) | ✔ | Easy |
| 4 | Ministry (Croatia) | ✔ | Easy |
| 5 | Branko Grcic | ✘ | - |

Figure 21: Intermediate variable assignments and ranks for example Pe_e2i query. Correctness indicates whether the answer belongs to the ground-truth set of answers.

**Pe_t2i** $Pe(e_1, r_1, t2i(e_2, r_2, e_3, e_4, r_3, e_5))$

$e_1$ : Government (South Africa)
$r_1$ : $Sign\ formal\ agreement^{-1}$
$e_2$ : Gazprom
$r_2$ : $Sign\ formal\ agreement^{-1}$
$e_3$ : China
$e_4$ : Mzukisi Fatyela
$r_3$ : Make statement
$e_5$ : Police (South Africa)

| Query Sentence | Who signed a formal agreement with Government (South Africa) at the time when Gazprom signed a formal agreement with China and Mzukisi Fatyela made a statement to Police (South Africa)? |
|---|---|
| Temporal Query | $q[V_?] = V_?, \exists T_a, r_1(e_1, V_?, T_a) \wedge r_2(e_2, e_3, T_a) \wedge r_3(e_5, e_5, T_a)$ |

| Rank | Query Answers | Correctness | Answer Type |
|---|---|---|---|
| 1 | South Africa | ✔ | Easy |
| 2 | Tom Motsoahae Thabane | ✘ | - |
| 3 | China | ✘ | - |
| 4 | Bank (China) | ✘ | - |
| 5 | Henry Rotich | ✘ | - |

Figure 22: Intermediate variable assignments and ranks for example Pe_t2i query. Correctness indicates whether the answer belongs to the ground-truth set of answers.

**t2i** $TimeAnd(Pt(e_1, r_1, e_2), Pt(e_3, r_2, e_4))$

$e_1$ : Xi Jinping
$r_1$ : $Engage\ in\ negotiation^{-1}$
$e_2$ : Barack Obama
$e_3$ : Xi Jinping
$r_2$ : $Consult^{-1}$
$e_4$ : Barack Obama

| Query Sentence | At what time did Barack Obama negotiate with and consult Xi Jinping? |
|---|---|
| Temporal Query | $q[T_?] = T_?, r_1(e_1, e_2, T_?) \wedge r_2(e_3, e_4, T_?))$ |

| Rank | Query Answers | Correctness | Answer Type |
|---|---|---|---|
| 1 | 2014-11-15 | ✔ | Easy |
| 2 | 2014-03-24 | ✔ | Easy |
| 3 | 2014-11-12 | ✔ | Easy |
| 4 | 2014-11-11 | ✔ | Easy |
| d 5 | 2014-11-14 | ✔ | Easy |

Figure 23: Intermediate variable assignments and ranks for example t2i query. Correctness indicates whether the answer belongs to the ground-truth set of answers.

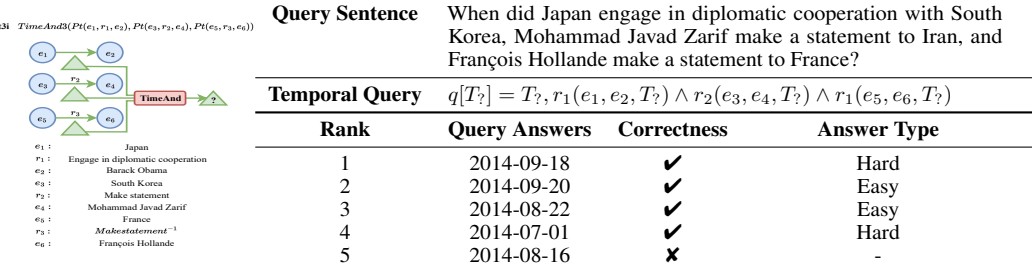

| Query Sentence | When did Japan engage in diplomatic cooperation with South Korea, Mohammad Javad Zarif make a statement to Iran, and François Hollande make a statement to France? | | |
|---|---|---|---|
| **Temporal Query** | $q[T_?] = T_?, r_1(e_1, e_2, T_?) \wedge r_2(e_3, e_4, T_?) \wedge r_1(e_5, e_6, T_?)$ | | |
| **Rank** | **Query Answers** | **Correctness** | **Answer Type** |
| 1 | 2014-09-18 | ✔ | Hard |
| 2 | 2014-09-20 | ✔ | Easy |
| 3 | 2014-08-22 | ✔ | Easy |
| 4 | 2014-07-01 | ✔ | Hard |
| 5 | 2014-08-16 | ✘ | - |

Figure 24: Intermediate variable assignments and ranks for example t3i query. Correctness indicates whether the answer belongs to the ground-truth set of answers.

| Query Sentence | When did Activist (Somalia) fight with small arms and light weapons with Armed Gang (Somalia) while Citizen (Oman) engage in a symbolic act with the person with whom finance / Economy / Commerce / Trade Ministry (Oman) shared intelligence or information on 2014-01-06? | | |
|---|---|---|---|
| **Temporal Query** | $q[T_?] = T_?, \exists V_a, r_1(e_1, V_a, t_1) \wedge r_2(V_a, e_2, T_?) \wedge r_3(e_3, e_4, T_?)$ | | |
| **Rank** | **Query Answers** | **Correctness** | **Answer Type** |
| 1 | 2014-01-30 | ✔ | Easy |
| 2 | 2014-02-05 | ✘ | - |
| 3 | 2014-01-29 | ✘ | - |
| 4 | 2014-02-07 | ✘ | - |
| 5 | 2014-02-04 | ✘ | - |

Figure 25: Intermediate variable assignments and ranks for example t2i_Pe query. Correctness indicates whether the answer belongs to the ground-truth set of answers.

| Query Sentence | When did Iran negotiate with the country that visited Iran on 2014-03-09? | | |
|---|---|---|---|
| **Temporal Query** | $q[T_?] = T_?, \exists V_a, r_1(e_1, V_a, t_1) \wedge r_2(e_2, V_a, t_2) \wedge r_3(V_a, e_3, T_?)$ | | |
| **Rank** | **Query Answers** | **Correctness** | **Answer Type** |
| 1 | 2014-03-10 | ✔ | Easy |
| 2 | 2014-03-08 | ✔ | Easy |
| 3 | 2014-06-16 | ✔ | Easy |
| 4 | 2014-03-13 | ✔ | Easy |
| 5 | 2014-03-09 | ✔ | Easy |

Figure 26: Intermediate variable assignments and ranks for example Pt_se2i query. Correctness indicates whether the answer belongs to the ground-truth set of answers.

| Query Sentence | When did Armenia cooperate militarily with the country that cooperates militarily with Armenia on 2014-12-08 and 2014-07-02? | | |
|---|---|---|---|
| **Temporal Query** | $q[V_?] = V_?, \exists V_a, r_1(e_1, V_a, T_?) \wedge r_2(e_2, V_a, t_1) \wedge r_3(e_3, V_a, t_2)$ | | |
| **Rank** | **Query Answers** | **Correctness** | **Answer Type** |
| 1 | 2014-12-08 | ✔ | Easy |
| 2 | 2014-07-02 | ✔ | Easy |
| 3 | 2014-05-06 | ✔ | Easy |
| 4 | 2014-10-26 | ✔ | Easy |
| 5 | 2014-07-17 | ✘ | - |

Figure 27: Intermediate variable assignments and ranks for example Pt_oe2i query. Correctness indicates whether the answer belongs to the ground-truth set of answers.

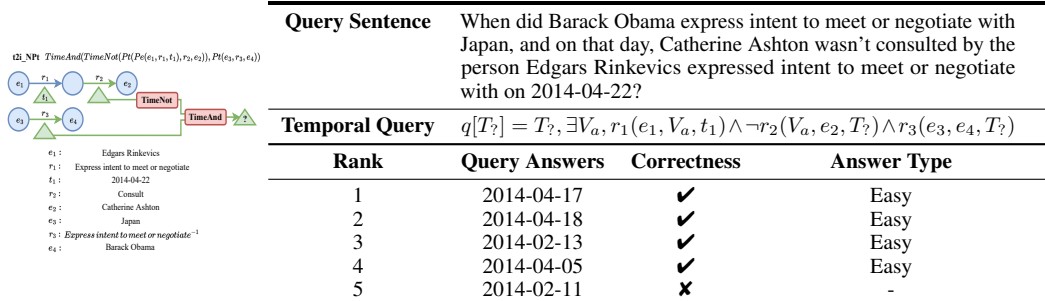

| | | |
|---|---|---|
| **Query Sentence** | When did Barack Obama express intent to meet or negotiate with Japan, and on that day, Catherine Ashton wasn't consulted by the person Edgars Rinkevics expressed intent to meet or negotiate with on 2014-04-22? | |
| **Temporal Query** | $q[T_?] = T_?, \exists V_a, r_1(e_1, V_a, t_1) \wedge \neg r_2(V_a, e_2, T_?) \wedge r_3(e_3, e_4, T_?)$ | |

| Rank | Query Answers | Correctness | Answer Type |
|---|---|---|---|
| 1 | 2014-04-17 | ✔ | Easy |
| 2 | 2014-04-18 | ✔ | Easy |
| 3 | 2014-02-13 | ✔ | Easy |
| 4 | 2014-04-05 | ✔ | Easy |
| 5 | 2014-02-11 | ✘ | - |

Figure 28: Intermediate variable assignments and ranks for example t2i_NPt query. Correctness indicates whether the answer belongs to the ground-truth set of answers.

| | | |
|---|---|---|
| **Query Sentence** | When was the citizen (Greece) made an appeal or request by the country that made an appeal or request to the Party President (Greece), and on that day, Angela Merkel did not express intent to meet or negotiate with Evo Morales? | |
| **Temporal Query** | $q[T_?] = T_?, \exists V_a, r_1(e_1, V_a, t_1) \wedge r_2(V_a, e_2, T_?) \wedge \neg r_3(e_3, e_4, T_?)$ | |

| Rank | Query Answers | Correctness | Answer Type |
|---|---|---|---|
| 1 | 2014-05-16 | ✔ | Easy |
| 2 | 2014-07-24 | ✔ | Easy |
| 3 | 2014-05-25 | ✔ | Easy |
| 4 | 2014-04-29 | ✔ | Easy |
| 5 | 2014-07-29 | ✔ | Easy |

Figure 29: Intermediate variable assignments and ranks for example t2i_PtN query. Correctness indicates whether the answer belongs to the ground-truth set of answers.

| | | |
|---|---|---|
| **Query Sentence** | Who discussed by telephone with Sergey Viktorovich Lavrov on the day when Hun Sen was threatened by the person who was arrested, detained, or charged with legal action by the Buddhist (Cambodia) on 2014-04-30, and on that day, Education (Colombia) made a statement to Eric Garner? | |
| **Temporal Query** | $q[V_?] = V_?, \exists V_a, T_a, r_1(e_1, V_?, T_a) \wedge r_2(e_2, V_a, t_1) \wedge r_3(V_a, e_3, T_a) \wedge \neg r_4(e_4, e_5, T_a)$ | |

| Rank | Query Answers | Correctness | Answer Type |
|---|---|---|---|
| 1 | John Kerry | ✔ | Hard |
| 2 | Cabinet / Council of Ministers / Advisors (United States) | ✔ | Easy |
| 3 | Catherine Ashton | ✘ | - |
| 4 | Sergey Viktorovich Lavrov | ✘ | - |
| 5 | Joseph Robinette Biden | ✘ | - |

Figure 30: Intermediate variable assignments and ranks for example Pe_t2i_N query. Correctness indicates whether the answer belongs to the ground-truth set of answers.

| | | |
|---|---|---|
| **Query Sentence** | Who visited Iran on 2014-04-29 and did not made a statement at the place where Carlos Saul Menem made a statement on 2014-09-20? | |
| **Temporal Query** | $q[V_?] = V_?, \exists V_a, V_b, r_1(e_1, V_a, t_1) \wedge \neg r_2(V_a, V_b, t_2) \wedge r_3(e_2, V_?, t_3)$ | |

| Rank | Query Answers | Correctness | Answer Type |
|---|---|---|---|
| 1 | Santos Edelmar Lopez | ✔ | Easy |
| 2 | Simon Gass | ✔ | Easy |
| 3 | Sebastian Kurz | ✔ | Easy |
| 4 | Catherine Ashton | ✘ | - |
| 5 | Iran | ✘ | - |

Figure 31: Intermediate variable assignments and ranks for example e2i_NPe query. Correctness indicates whether the answer belongs to the ground-truth set of answers.

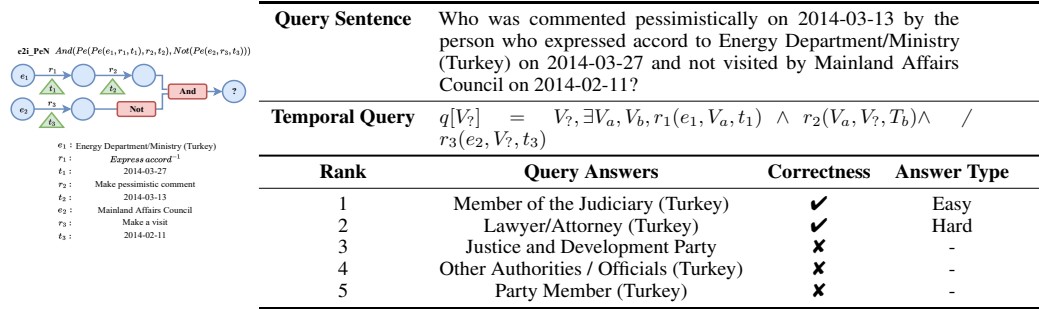

| Query Sentence | Who was commented pessimistically on 2014-03-13 by the person who expressed accord to Energy Department/Ministry (Turkey) on 2014-03-27 and not visited by Mainland Affairs Council on 2014-02-11? |
|---|---|
| Temporal Query | $q[V_?] = V_?, \exists V_a, V_b, r_1(e_1, V_a, t_1) \wedge r_2(V_a, V_?, T_b) \wedge \not{r_3}(e_2, V_?, t_3)$ |

e2i_PeN $And(Pe(Pe(e_1, r_1, t_1), r_2, t_2), Not(Pe(e_2, r_3, t_3)))$

$e_1$ : Energy Department/Ministry (Turkey)
$r_1$ : $Express\ accord^{-1}$
$t_1$ : 2014-03-27
$r_2$ : Make pessimistic comment
$t_2$ : 2014-03-13
$e_2$ : Mainland Affairs Council
$r_3$ : Make a visit
$t_3$ : 2014-02-11

| Rank | Query Answers | Correctness | Answer Type |
|---|---|---|---|
| 1 | Member of the Judiciary (Turkey) | ✔ | Easy |
| 2 | Lawyer/Attorney (Turkey) | ✔ | Hard |
| 3 | Justice and Development Party | ✘ | - |
| 4 | Other Authorities / Officials (Turkey) | ✘ | - |
| 5 | Party Member (Turkey) | ✘ | - |

Figure 32: Intermediate variable assignments and ranks for example e2i_PeN query. Correctness indicates whether the answer belongs to the ground-truth set of answers.

Pe_e2i_Pe_NPe $Pe(And(Pe(e_1, r_1, t_1), Not(Pe(e_2, r_2, t_2))), r_3, t_3)$

$e_1$ : Treasury/Finance Ministry (Eritrea)
$r_1$ : $Consult^{-1}$
$t_1$ : 2014-09-17
$e_2$ : Drug Gang (Beltrán -Leyva Cartel)
$r_2$ : $Arrest,\ detain,\ or\ charge\ with\ legal\ action^{-1}$
$t_2$ : 2014-10-16
$r_3$ : $Engage\ in\ negotiation^{-1}$
$t_3$ : 2014-10-21

| Query Sentence | On 2014-10-21, who negotiated with the person who consulted with the Treasury/Finance Ministry (Eritrea) on 2014-09-17 but did not arrest, detain, or charge Drug Gang (Beltrán -Leyva Cartel) with legal action on 2014-10-16? |
|---|---|
| Temporal Query | $q[V_?] = V_?, \exists V_a, V_b, r_1(e_1, V_a, t_1) \wedge \not{r_2}(e_2, V_a, t_2) \wedge r_3(V_a, V_?, t_3)$ |

| Rank | Query Answers | Correctness | Answer Type |
|---|---|---|---|
| 1 | South Korea | ✔ | Easy |
| 2 | Japan | ✘ | - |
| 3 | China | ✔ | Hard |
| 4 | North Korea | ✘ | - |
| 5 | Thailand | ✘ | - |

Figure 33: Intermediate variable assignments and ranks for example Pe_e2i_N query. Correctness indicates whether the answer belongs to the ground-truth set of answers.

e2i_N $And(Pe(e_1, r_1, t_1), Not(Pe(e_2, r_2, t_2)))$

$e_1$ : Botswana
$r_1$ : $Rally\ opposition\ against^{-1}$
$t_1$ : 2014-03-11
$e_2$ : Citizen (Brunei)
$r_2$ : Make empathetic comment
$t_2$ : 2014-03-26

| Query Sentence | Who rallied opposition against Botswana on 2014-03-11 but was not empathetically commented upon by Citizen (Brunei) on 2014-03-26? |
|---|---|
| Temporal Query | $q[V_?] = V_?, r_1(e_1, V_?, t_1) \wedge \neg r_2(e_2, V_?, t_2)$ |

| Rank | Query Answers | Correctness | Answer Type |
|---|---|---|---|
| 1 | Industry (Botswana) | ✔ | Easy |
| 2 | Citizen (Botswana) | ✔ | Hard |
| 3 | Richard Sezibera | ✘ | - |
| 4 | Labor and Employment Ministry (Botswana) | ✘ | - |
| 5 | Paul Kagame | ✘ | - |

Figure 34: Intermediate variable assignments and ranks for example e2i_N query. Correctness indicates whether the answer belongs to the ground-truth set of answers.

e3i_N $And3(Pe(e_1, r_1, t_1), Pe(e_2, r_2, t_2), Not(Pe(e_3, r_3, t_3)))$

$e_1$ : Envoy (Kazakhstan)
$r_1$ : Praise or endorse
$t_1$ : 2014-05-19
$e_2$ : RIA Novosti
$r_2$ : Sign formal agreement
$t_2$ : 2014-12-12
$e_3$ : Governor (South Korea)
$r_3$ : $Sign\ formal\ agreement^{-1}$
$t_3$ : 2014-05-09

| Query Sentence | Who was praised or endorsed by Envoy (Kazakhstan) on 2014-05-19 and signed a formal agreement with RIA Novosti on 2014-12-12 but not signed a formal agreement with Governor (South Korea) on 2014-05-09? |
|---|---|
| Temporal Query | $q[V_?] = V_?, r_1(e_1, V_?, t_1) \wedge r_2(e_2, V_?, t_2) \wedge \not{r_3}(e_3, V_?, t_3)$ |

| Rank | Query Answers | Correctness | Answer Type |
|---|---|---|---|
| 1 | Iran | ✔ | Easy |
| 2 | Iraq | ✘ | - |
| 3 | China | ✘ | - |
| 4 | Afghanistan | ✘ | - |
| 5 | North Atlantic Treaty Organization | ✘ | - |

Figure 35: Intermediate variable assignments and ranks for example e3i_N query. Correctness indicates whether the answer belongs to the ground-truth set of answers.

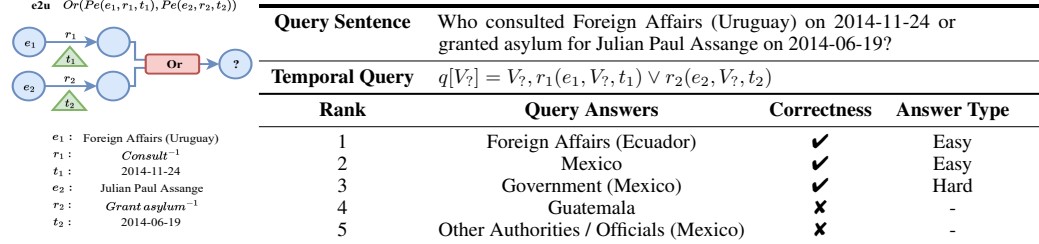

**e2u** $Or(Pe(e_1, r_1, t_1), Pe(e_2, r_2, t_2))$

$e_1$: Foreign Affairs (Uruguay)
$r_1$: $Consult^{-1}$
$t_1$: 2014-11-24
$e_2$: Julian Paul Assange
$r_2$: $Grant\,asylum^{-1}$
$t_2$: 2014-06-19

| | | |
|---|---|---|
| **Query Sentence** | | Who consulted Foreign Affairs (Uruguay) on 2014-11-24 or granted asylum for Julian Paul Assange on 2014-06-19? |
| **Temporal Query** | | $q[V_?] = V_?, r_1(e_1, V_?, t_1) \lor r_2(e_2, V_?, t_2)$ |

| Rank | Query Answers | Correctness | Answer Type |
|---|---|---|---|
| 1 | Foreign Affairs (Ecuador) | ✔ | Easy |
| 2 | Mexico | ✔ | Easy |
| 3 | Government (Mexico) | ✔ | Hard |
| 4 | Guatemala | ✘ | - |
| 5 | Other Authorities / Officials (Mexico) | ✘ | - |

Figure 36: Intermediate variable assignments and ranks for example e2u query. Correctness indicates whether the answer belongs to the ground-truth set of answers.

**Pe_e2u** $Pe(Or(Pe(e_1, r_1, t_1), Pe(e_2, r_2, t_2)), r_3, t_3)$

$e_1$: Lower House of Parliament (France)
$r_1$: Praise or endorse
$t_1$: 2014-11-18
$e_2$: Newspaper (Afghanistan)
$r_2$: Make an appeal or request
$t_2$: 2014-05-05
$r_3$: $Occupy\,territory^{-1}$
$t_3$: 2014-05-24

| | | |
|---|---|---|
| **Query Sentence** | | On 2014-05-24, who occupied the land praised or endorsed by the Lower House of Parliament (France) on 2014-11-18 or requested or appealed by a Newspaper (Afghanistan) on 2014-05-05? |
| **Temporal Query** | | $q[V_?] = V_?, \exists V_a, (r_1(e_1, V_a, t_1) \lor r_2(e_2, V_a, t_2)) \land r_2(V_a, V_?, t_3)$ |

| Rank | Query Answers | Correctness | Answer Type |
|---|---|---|---|
| 1 | Armed Gang (Afghanistan) | ✔ | Easy |
| 2 | Armed Band (Afghanistan) | ✔ | - |
| 3 | Military Personnel - Special (Afghanistan) | ✔ | Hard |
| 4 | Militant (Taliban) | ✘ | - |
| 5 | Afghanistan | ✘ | - |

Figure 37: Intermediate variable assignments and ranks for example Pe_e2u query. Correctness indicates whether the answer belongs to the ground-truth set of answers.

**Pt_lPe** $Pt(Pe(e_1, r_1, t_1), r_2, e_2)$

$e_1$: Middle East
$r_1$: $Make\,a\,visit^{-1}$
$t_1$: 2014-09-15
$r_2$: Make a visit
$e_2$: Iraq

| | | |
|---|---|---|
| **Query Sentence** | | When was Iraq visited by the person who visited The Middle East on 2014-09-15? |
| **Temporal Query** | | $q[T_?] = T_?, \exists V_a, r_1(e_1, V_a, t_1) \land r_2(V_a, e_2, T_?)$ |

| Rank | Query Answers | Correctness | Answer Type |
|---|---|---|---|
| 1 | 2014-09-13 | ✔ | Easy |
| 2 | 2014-06-25 | ✔ | Easy |
| 3 | 2014-06-26 | ✔ | Easy |
| 4 | 2014-06-27 | ✔ | Easy |
| 5 | 2014-06-24 | ✔ | Easy |

Figure 38: Intermediate variable assignments and ranks for example Pt_sPe query. Correctness indicates whether the answer belongs to the ground-truth set of answers.

**Pt_rPe** $Pt(e_1, r_1, Pe(e_2, r_2, t_1))$

$e_1$: France
$r_1$: Meet at a 'third' location
$e_2$: Peter Humphrey
$r_2$: $Make\,statement^{-1}$
$t_1$: 2014-07-31

| | | |
|---|---|---|
| **Query Sentence** | | At what time did France meet the person who made a statement to Peter Humphrey at a 'third' location? |
| **Temporal Query** | | $q[T_?] = T_?, \exists V_a, r_1(e_1, V_a, T_?) \land r_2(e_2, V_a, t_1))$ |

| Rank | Query Answers | Correctness | Answer Type |
|---|---|---|---|
| 1 | 2014-11-21 | ✔ | Hard |
| 2 | 2014-11-25 | ✔ | Easy |
| 3 | 2014-11-20 | ✔ | Easy |
| 4 | 2014-09-23 | ✔ | Easy |
| 5 | 2014-09-21 | ✔ | Easy |

Figure 39: Intermediate variable assignments and ranks for example Pt_oPe query. Correctness indicates whether the answer belongs to the ground-truth set of answers.

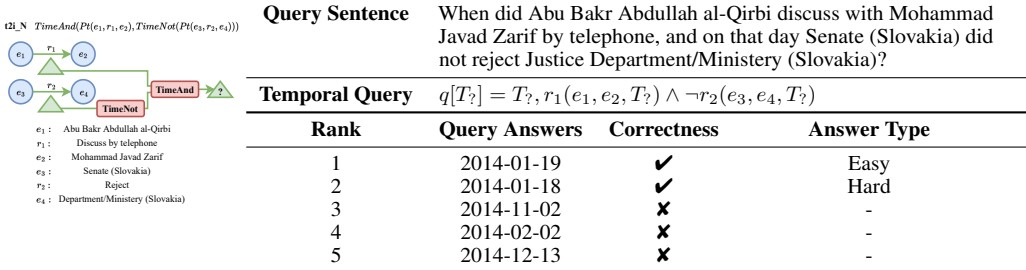

**t2i_N** $TimeAnd(Pt(e_1, r_1, e_2), TimeNot(Pt(e_3, r_2, e_4)))$

$e_1$ : Abu Bakr Abdullah al-Qirbi
$r_1$ : Discuss by telephone
$e_2$ : Mohammad Javad Zarif
$e_3$ : Senate (Slovakia)
$r_2$ : Reject
$e_4$ : Department/Ministery (Slovakia)

| Query Sentence | When did Abu Bakr Abdullah al-Qirbi discuss with Mohammad Javad Zarif by telephone, and on that day Senate (Slovakia) did not reject Justice Department/Ministery (Slovakia)? | | |
|---|---|---|---|
| Temporal Query | $q[T_?] = T_?, r_1(e_1, e_2, T_?) \wedge \neg r_2(e_3, e_4, T_?)$ | | |
| **Rank** | **Query Answers** | **Correctness** | **Answer Type** |
| 1 | 2014-01-19 | ✔ | Easy |
| 2 | 2014-01-18 | ✔ | Hard |
| 3 | 2014-11-02 | ✘ | - |
| 4 | 2014-02-02 | ✘ | - |
| 5 | 2014-12-13 | ✘ | - |

Figure 40: Intermediate variable assignments and ranks for example t2i_N query. Correctness indicates whether the answer belongs to the ground-truth set of answers.

**t3i_N** $TimeAnd3(Pt(e_1, r_1, e_2), Pt(e_3, r_2, e_4), TimeNot(Pt(e_5, r_3, e_6)))$

$e_1$ : Police (Egypt)
$r_1$ : Use tactics of violent repression
$e_2$ : Protester (Egypt)
$e_3$ : Opposition Activist (Ukraine)
$r_2$ : Use tactics of violent repression$^{-1}$
$e_4$ : Police (Ukraine)
$e_5$ : Information / Communication / Transparency Ministry (China)
$r_3$ : Consult
$e_6$ : Warren Truss

| Query Sentence | When did the Police (Egypt) use tactics of violent repression against protesters (Egypt) and the Police (Ukraine) use tactics of violent repression against protesters (Egypt), and on that day Information / Communication / Transparency Ministry (China) did not consult Warren Truss? | | |
|---|---|---|---|
| Temporal Query | $q[T_?] = T_?, r_1(e_1, e_2, T_?) \wedge r_2(e_3, e_4, T_?) \wedge \neg r_3(e_5, e_6, T_?)$ | | |
| **Rank** | **Query Answers** | **Correctness** | **Answer Type** |
| 1 | 2014-01-21 | ✔ | Easy |
| 2 | 2014-01-13 | ✘ | - |
| 3 | 2014-01-22 | ✘ | - |
| 4 | 2014-02-20 | ✘ | - |
| 5 | 2014-01-14 | ✘ | Hard |

Figure 41: Intermediate variable assignments and ranks for example t3i_N query. Correctness indicates whether the answer belongs to the ground-truth set of answers.

**t2u** $TimeOr(Pt(e_1, r_1, e_2), Pt(e_3, r_2, e_4))$

$e_1$ : Nonofo Molefhi
$r_1$ : Obstruct passage, block$^{-1}$
$e_2$ : Citizen (Botswana)
$e_3$ : Zheng Zeguang
$r_2$ : Make statement
$e_4$ : China

| Query Sentence | When did citizens (Botswana) obstruct passage and block Nonofo Molefhi or Zheng Zeguang make statements about China? | | |
|---|---|---|---|
| Temporal Query | $q[T_?] = T_?, r_1(e_1, e_2, T_?) \vee r_2(e_3, e_4, T_?)$ | | |
| **Rank** | **Query Answers** | **Correctness** | **Answer Type** |
| 1 | 2014-04-09 | ✔ | Easy |
| 2 | 2014-05-21 | ✔ | Easy |
| 3 | 2014-05-20 | ✔ | Hard |
| 4 | 2014-05-22 | ✘ | - |
| 5 | 2014-05-22 | ✘ | - |

Figure 42: Intermediate variable assignments and ranks for example t2u query. Correctness indicates whether the answer belongs to the ground-truth set of answers.

**Pe_t2u** $Pe(e_1, r_1, TimeOr(Pt(e_2, r_2, e_3), Pt(e_4, r_3, e_5)))$

$e_1$ : Citizen (Nigeria)
$r_1$ : Make an appeal or request
$e_2$ : University of Cape Town
$r_2$ : Consult
$e_3$ : News Editor (Nigeria)
$e_4$ : Muslim (United Arab Emirates)
$r_3$ : Arrest, detain, or charge with legal action$^{-1}$
$e_5$ : State Security Court (United Arab Emirates)

| Query Sentence | Who was appealed or requested by Citizen (Nigeria) on the day when the University of Cape Town consulted News Editor (Nigeria) or State Security Court (United Arab Emirates) arrested, detained, or charged Muslim (United Arab Emirates) with legal action? | | |
|---|---|---|---|
| Temporal Query | $q[V_?] = V_?, \exists T_a, r_1(e_1, V_?, T_a) \wedge (r_2(e_2, e_3, T_a) \vee r_3(e_4, e_5, T_a))$ | | |
| **Rank** | **Query Answers** | **Correctness** | **Answer Type** |
| 1 | Government (Nigeria) | ✔ | Easy |
| 2 | Assemblies (Nigeria) | ✔ | Hard |
| 3 | Other Authorities / Officials (Nigeria) | ✘ | - |
| 4 | Head of Government (Nigeria) | ✘ | - |
| 5 | Citizen (Nigeria) | ✘ | - |

Figure 43: Intermediate variable assignments and ranks for example Pe_t2u query. Correctness indicates whether the answer belongs to the ground-truth set of answers.

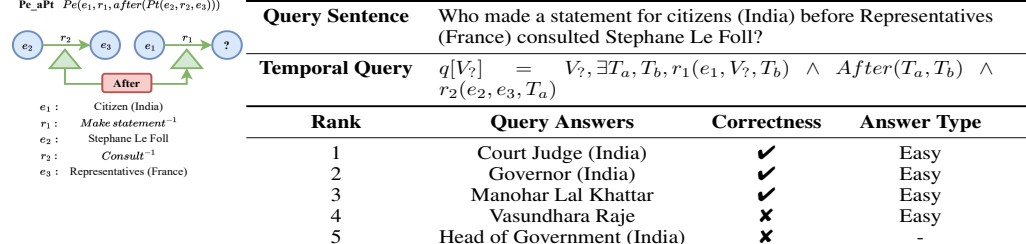

**Pe_aPt** $Pe(e_1, r_1, after(Pt(e_2, r_2, e_3)))$

$e_1$ :  Citizen (India)
$r_1$ :  *Make statement*$^{-1}$
$e_2$ :  Stephane Le Foll
$r_2$ :  *Consult*$^{-1}$
$e_3$ :  Representatives (France)

| Query Sentence | Who made a statement for citizens (India) before Representatives (France) consulted Stephane Le Foll? | | |
|---|---|---|---|
| Temporal Query | $q[V_?] = V_?, \exists T_a, T_b, r_1(e_1, V_?, T_b) \wedge After(T_a, T_b) \wedge r_2(e_2, e_3, T_a)$ | | |
| **Rank** | **Query Answers** | **Correctness** | **Answer Type** |
| 1 | Court Judge (India) | ✔ | Easy |
| 2 | Governor (India) | ✔ | Easy |
| 3 | Manohar Lal Khattar | ✔ | Easy |
| 4 | Vasundhara Raje | ✘ | Easy |
| 5 | Head of Government (India) | ✘ | - |

Figure 44: Intermediate variable assignments and ranks for example Pe_aPt query. Correctness indicates whether the answer belongs to the ground-truth set of answers.

**Pe_bPt** $Pe(e_1, r_1, before(Pt(e_2, r_2, e_3)))$

$e_1$ :  South Korea
$r_1$ :  Criticize or denounce
$e_2$ :  Sadako Ogata
$r_2$ :  *Express intent to meet or negotiate*$^{-1}$
$e_3$ :  Head of Government(Bangladesh)

| Query Sentence | Who expressed intent to meet or negotiate with the Head of Government (Bangladesh) after South Korea Criticized or denounced Sadako Ogata? | | |
|---|---|---|---|
| Temporal Query | $q[V_?] = V_?, \exists T_a, T_b, r_1(e_1, V_?, T_b) \wedge Before(T_a, T_b) \wedge r_2(e_2, e_3, T_a)$ | | |
| **Rank** | **Query Answers** | **Correctness** | **Answer Type** |
| 1 | Japan | ✔ | Easy |
| 2 | North Korea | ✔ | Easy |
| 3 | China | ✔ | Easy |
| 4 | South Korea | ✘ | - |
| 5 | Kim Jong-Un | ✘ | - |

Figure 45: Intermediate variable assignments and ranks for example Pe_bPt query. Correctness indicates whether the answer belongs to the ground-truth set of answers.

**Pe_at2i** $Pe(e_1, r_1, after(t2i(e_2, r_2, e_3, e_4, r_3, e_5)))$

$e_1$ :  North Korea
$r_1$ :  Criticize or denounce
$e_2$ :  Admiral (Russia)
$r_2$ :  Make statement
$e_3$ :  Chuck Hagel
$e_4$ :  Ministry (Madagascar)
$r_3$ :  Make statement
$e_5$ :  Government (Madagascar)

| Query Sentence | Who was criticized or denounced by North Korea on the day before Admiral (Russia) made a statement for Chuck Hagel and Ministry (Madagascar) made a statement for Government (Madagascar)? | | |
|---|---|---|---|
| Temporal Query | $q[V_?] = V_?, \exists T_a, T_b, r_1(e_1, V_?, T_b)) \wedge After(T_a, T_b) \wedge r_2(e_2, e_3, T_a) \wedge r_3(e_4, e_5, T_a)$ | | |
| **Rank** | **Query Answers** | **Correctness** | **Answer Type** |
| 1 | South Korea | ✔ | Easy |
| 2 | Japan | ✔ | Easy |
| 3 | North Korea | ✔ | - |
| 4 | Head of Government (South Korea) | ✔ | Easy |
| 5 | John Kerry | ✔ | Easy |

Figure 46: Intermediate variable assignments and ranks for example Pe_at2i query. Correctness indicates whether the answer belongs to the ground-truth set of answers.

**Pe_bt2i** $Pe(e_1, r_1, before(t2i(e_2, r_2, e_3, e_4, r_3, e_5)))$

$e_1$ :  Citizen (India)
$r_1$ :  Decline comment
$e_2$ :  Presidential Candidate (Mozambique)
$r_2$ :  *Consult*$^{-1}$
$e_3$ :  Employee (Mozambique)
$e_4$ :  Evo Morales
$r_3$ :  Make statement
$e_5$ :  Military (Bolivia)

| Query Sentence | Who was declined commented by Citizen (India) after the day when the employee (Mozambique) consulted the Presidential Candidate (Mozambique), and Evo Morales made a statement to the Military (Bolivia)? | | |
|---|---|---|---|
| Temporal Query | $q[V_?] = V_?, \exists T_a, T_b, r_1(e_1, V_?, T_b) \wedge Before(T_a, T_b) \wedge r_2(e_2, e_3, T_a) \wedge r_3(e_4, e_5, T_a)$ | | |
| **Rank** | **Query Answers** | **Correctness** | **Answer Type** |
| 1 | Media (India) | ✔ | Easy |
| 2 | Member of Parliament (India) | ✘ | - |
| 3 | Court Judge (India) | ✔ | Easy |
| 4 | Head of Government (India) | ✘ | - |
| 5 | Governor (India) | ✘ | - |

Figure 47: Intermediate variable assignments and ranks for example Pe_bt2i query. Correctness indicates whether the answer belongs to the ground-truth set of answers.

between $TimeAnd(after(Pt(e_1, r_1, e_2)), before(Pt(e_3, r_3, e_4)))$

$e_1$ : Other Authorities / Officials (Romania)
$r_1$ : *Appeal for diplomatic cooperation(suchaspolicysupport)$^{-1}$*
$e_2$ : Foreign Affairs (United States)
$e_3$ : Tsai Ing-wen
$r_2$ : Accuse
$e_4$ : Party Member (Taiwan)

| **Query Sentence** | What was the day between the day that Tsai Ing-wen accused the Party Member (Taiwan) and the foreign affairs (United States) appealed for diplomatic cooperation (such as policy support) from the other Authorities/officials (Romania)? |
|---|---|
| **Temporal Query** | $q[T_?] \quad = \quad T_?, \exists T_a, T_b, r_1(e_1, e_2, T_a) \ \wedge \ r_2(e_3, e_4, T_b) \ \wedge$ $After(T_a, T_?) \wedge Before(T_b, T_?)$ |

| **Rank** | **Query Answers** | **Correctness** | **Answer Type** |
|---|---|---|---|
| 1 | 2014-07-19 | ✔ | Easy |
| 2 | 2014-07-29 | ✔ | - |
| 3 | 2014-06-10 | ✔ | Easy |
| 4 | 2014-08-21 | ✔ | - |
| d 5 | 2014-06-25 | ✔ | Easy |

Figure 48: Intermediate variable assignments and ranks for example between query. Correctness indicates whether the answer belongs to the ground-truth set of answers.

