# OpenReview forum: "TFLEX: Temporal Feature-Logic Embedding Framework for Complex Reasoning over Temporal Knowledge Graph"
_NeurIPS.cc/2023/Conference — NeurIPS 2023 poster_

### Official Review · Reviewer_Zrqw · 2023-06-19

**Soundness:** 2 fair
**Presentation:** 2 fair
**Contribution:** 3 good
**Rating:** 5
**Confidence:** 3

**Summary:**

This paper proposes TFLEX, a framework that reasons over temporal knowledge graphs (TKG). It takes a complex query about either an entity or a timestamp and a set of constraints as input. The query is then converted into a directed acyclic graph (DAG), which is a computation graph that projects the query into an embedding space. The paper also defines operations that happens in the graph, including projection, intersection, complement, and other set/temporal logic operations (e.g. After, Before, and Between).

When the query embedding is produced, its distances between all candidate entity/timestamp embeddings are computed. The loss function is designed to minimise the distance between the embedding of the query and that of the correct entity/timestamp and maximise the distance between the query and the incorrect candidates.

The framework was evaluated on three standard datasets with modifications. The assessment process involves predefined complex queries on top of each dataset, including 27 types for training and 40 types for test. Several variants of the proposed framework were compared, as well as a few previous works on embedding query for static knowledge graphs.

**Strengths:**

- This paper discusses complex query embedding in temporal knowledge graphs, which is an important yet under-investigated topic.
- The paper provides a definition to the problem and some attempts to solves the problem.

**Weaknesses:**

Related Works
- From my understanding, TKGC is an important subset of the task (Temporal Complex Query) defined in this paper. There are many TKGC works missing in the related work part. For example, [1] and [2].
- The discussion on complex query embedding works is insufficient. The paper should provide a more substantial explanation as to why existing works on static KG cannot be easily adapted for temporal KGs. "They cannot utilize temporal information" is a result, not a reason.

Method
- In line 146 the paper introduces "the relational embedding" without saying how it is defined. Is it a trainable parameter? How is it initialised?
- The definition to operators appears arbitrary. In equation (3), $V_q + r + V_t$ are adding the embedding for entities, relations, and timestamps together. What is the semantics of addition here?
- Again, in equation (7), something like $q_f^t+\frac{1+q_l^t}{2}$ is in essence adding the entity feature with its a truth value. What is the semantics of addition between these two values?
- Later in the definition of distance function (line 199), $q_l^e + ||v_f^e-q_f^e||_1$ is adding a truth value with a L1-norm. What is the semantics of addition here?

Experiments:
- The method is evaluated on 40 types of generated queries from three datasets for TKGC. The paper did no discuss the reason for choosing these specific 40 types or how complete these types are. Nor has the it discussed whether TFLEX can generalise to queries outside of those 40 types.
- There is no comparison with TKGC works. Since the TKGC task aims at completing a simpler query, it is naturally a subtask of what is proposed in this work. In other words, a (s, r, ?, t+1) completion task can be converted to a query and TFLEX should be able to handle it. I would expect at least a separate table comparing TFLEX and state-of-the-art TKGC on a completion task. Otherwise, it is difficult to assess how well TFLEX captures temporal information.

Miscellaneous:
- There are technical details being moved to appendix due to page limit. Please note that the page limit is nine content pages, not eight. The current main text is not self-contained to provide readers a high-level idea about the benefits of the framework. For example, as I pointed out above, many definitions and designs about the operators lack intuition or rationale, and look like something randomly popping out.
- The source code provided in the supplementary material and the link in the paper reveal a non-anonymous email "l****9@sysu.edu.cn". While I believe this is an honest mistake, I urge the author to remove this information.

---
Reference

[1] Chronor: Rotation based temporal knowledge graph embedding, AAAI 2021.
[2] Temporal Knowledge Graph Reasoning Based on Evolutional Representation Learning, SIGIR 2021.

**Questions:**

Please see my points above.

**Limitations:**

The paper has a good discussion about its limitations in page 11.

---

> ### Author Rebuttal · Authors · 2023-08-09
>
> We thank the reviewer for the positive and detailed feedback. Below please find responses to individual comments/questions:
>
> ## Related Works
>
> > TKGC works in the related work part.
>
> see global rebuttal.
>
> > The discussion on complex query embedding works is insufficient.
>
> To begin with, we have many discussions about the static-to-dynamic difficulty. Apart from the related work, we first mention the difficulty in line 29-30 (Introduction Section), then discuss static QEs’ poor results and analyze the reasons in line 230-232 (Experiment Section), and even further explore the right way to promote the static query embeddings to temporal ones in line 670-707 (a full page for exploration in Appendix C).
>
> We would like to add the below sentences in the revision to make it more comprehensive.
>
> ```
> Firstly, static query embeddings (QEs) are built over (s, r, o) triples instead of (s, r, o, t) quartets, thus ignoring the timestamps for temporal complex reasoning. The second reason is the order property of timestamps, which is on the contrary that entities are unordered, leading to that static QEs are unable to handle Before, After temporal logic. Therefore, it is challenging for static QEs to utilize temporal information in the TKGs.
> ```
>
> ## Method
>
> > the relational embedding.
>
> Yes, it’s trainable, and randomly initialised. Besides, entity embedding and timestamp embedding are also trainable and randomly initialised.
>
> > The semantics of addition in equation 3.
>
> The equation 3 follows the assumption of translation-based methods: $q_o \approx q_s + r + t$. As a comparison, static KGE TransE has $o\approx s+r$, and temporal KGE TTransE has $o \approx s + r + t$. The addition represents a semantical translation starting from the source entity set, following the relation and timestamp conditioning, ending at the target entity set.
>
> > What is the semantics of addition between these two values in equation 7?
>
> The entity feature and its truth value together form a fuzzy interval in the semantic space.
>
> Those entities, whose entity features are covered by the fuzzy interval, are viewed as the answers of the query.
>
> > What is the semantics of addition in the distance function (line 199)?
>
> The distance function aims to optimize two losses. One is to push the answers to the neighbor of query in the embedding space. It corresponds to the term L1 distance between answer and query. The other is to reduce the uncertainty of the query (the probability interpretation of the logic part), to make the answers more accurate. We use element-wise addition to combine the two losses.
>
> ## Experiments:
>
> > The paper did no discuss the reason for choosing these specific 40 types or how complete these types are. Nor has the it discussed whether TFLEX can generalise to queries outside of those 40 types.
>
> In Appendix B.2, we discuss why to choose these types in the dataset generation section in detail. We mention that it is in order to keep a similar experiment setting with previous static query embeddings. We also present the comparison of query types between temporal and static ones in Table 5.
>
> About generalization, please be aware that TFLEX is **trained on 27 types and evaluated on 40 types**. The extra 13 types contain various unseen query structures that do not exist in the training set. The experiment result **Out-of-data reasoning** (line257-258) shows the generalization ability of TFLEX. Therefore, it needs not to introduce more types of queries.
>
> > There is no comparison with TKGC works. I would expect at least a separate table comparing TFLEX and state-of-the-art TKGC on a completion task.
>
> We don’t agree with the opinion that we ignore all the TKGC baselines. Actually, we compare to two translation-based TKGC methods (TTransE and HyTE), present the results in Figure 4, and discuss in paragraph **Necessity of training on temporal complex queries** (line264-269). We choose TTransE and HyTE because our projection operator is also translation-based ($\mathcal{P}_e(\mathbf{V}_q,\mathbf{r},\mathbf{V}_t)  \propto \mathbf{V}_q+\mathbf{r}+\mathbf{V}_t$). Such comparison is fair enough to determine how well complex queries improve the performance of translation-base methods. Introducing other TKGC baselines leads to unfair comparison, and unable to investigate whether training on temporal complex queries is necessary or not.
>
> Anyway, we present the Table 1 in pdf in global rebuttal section, which is more complete, comparing TFLEX and SOTA TKGC methods. The table shows that TFLEX is competitive with translation-based methods, but it doesn't outperform the SOTA TKGC methods like ChronoR and TuckERT.
>
> However, the result doesn't affect the novelty and contribution of this paper. Please note that the projection operator of TFLEX is as simple as TTransE, not further optimized for TKGC tasks only. Upgrading the projection operator to outperform SOTA TKGC methods remains a future work. Besides, previous work GQE[1] did not compare to any KGC methods when the paper proposed complex logical reasoning task against KGC for the first time. And the following works including Query2box[2], BetaE[3], ConE[4], none of them compares to KGC methods.
>
> - [1] Embedding Logical Queries on Knowledge Graphs.
> - [2] Query2box: Reasoning Over Knowledge Graphs In Vector Space Using Box Embeddings.
> - [3] Beta Embeddings for Multi-Hop Logical Reasoning in Knowledge Graphs.
> - [4] ConE: Cone Embeddings for Multi-Hop Reasoning over Knowledge Graphs.
>
>
> ## Miscellaneous:
>
> > The presentation of the content.
>
> Thanks for pointing out. We will move more content to the main body in the revision.
>
> > Non-anonymous email in the supplementary material.
>
> Sorry for the mistake. The email belongs to a third-party package author. We will remove it.

---

> ### Author Response · Authors · 2023-08-16
>
> We sincerely appreciate your valuable and constructive comments. We eagerly anticipate receiving any additional feedback you may have. If you find our response satisfactory, we hope you will consider this a valid reason to consider raising your rating. Should you have any lingering questions about our paper, we are more than willing to address them and enhance the quality of our work.

---

> > ### Comment · Reviewer_Zrqw · 2023-08-17
> > **Thank you for the clarification**
> >
> > I have read the feedback from the author(s) and other reviews.
> > The comments have addressed most of my concerns.
> > Based on the experimental result in Table 1, their clarification on the semantics of equations, and the promised revision to improve readability, I would like to raise my rating of the paper.
> >
> > I would like to reiterate to the author(s) that the paper __needs substantial revision to reach publication-level readability__. My concerns regarding the semantics of the equation largely stem from the paper's lack of background and rationale. The paper should benefit a lot if knowledge shared in their feedback can be merged into either the main text or the appendix of the paper.

---

> > > ### Author Response · Authors · 2023-08-18
> > >
> > > Thank you for your insightful feedback on our paper. We appreciate your dedication to ensuring the quality and clarity of our work.
> > >
> > > We understand your concerns about the paper's overall readability. Your suggestions to incorporate additional background and rationale into either the main text or the appendix are invaluable, and we will certainly take this into account to provide a more comprehensive understanding of the context. Actually, we will also include more about the semantics of addition, the motivation of operator design, and the knowledge behind the equations to the main body of the paper. We hope these revision will make our paper publication-ready.
> > >
> > > Once again, we believe that your insightful suggestions will greatly contribute to our paper refinement. We kindly request that you increase your rating to acceptance in light of the improvements we have made. A higher rating is an encouragement and affirmation of our work in this field. Your guidance has been instrumental in guiding our revisions, and we are grateful for your ongoing support.
> > >
> > > Thank you once again for your time and consideration.

---

> > > ### Comment · Area_Chair_zDMr · 2023-08-20
> > > **Please check the authors' improvement on readability**
> > >
> > > Dear Reviewer Zrqw. Since you're mainly concerned about readability, can you please check the authors' recent response at https://openreview.net/forum?id=oaGdsgB18L&noteId=KOfY3P2mqt, where they listed their plan to improve the readability of the paper, as well as a new appendix. Please let me know if your concern about readability is resolved.

---

> > > > ### Comment · Reviewer_Zrqw · 2023-08-22
> > > >
> > > > Dear AC, my concerns have been addressed by the authors' feedback. Provided that the feedback is well incorporated in the future revisions of the paper, as promised by the authors, I have no objection to its acceptance.

---

### Official Review · Reviewer_NJDL · 2023-07-04

**Soundness:** 2 fair
**Presentation:** 2 fair
**Contribution:** 3 good
**Rating:** 6
**Confidence:** 3

**Summary:**

The authors introduce an embedding-based method for answering complex, i.e., multi-hop, queries on temporal knowledge graphs. The Temporal Feature-Logic Embedding framework (TFLEX) uses fuzzy logic to model first-order logic operations on the entity and timestamp sets. The queries, with answers being either entities or timestamps, are embedded via four components, including feature and logic vectors for both entities and timestamps. The correct answer to the query is determined as the entity or timestamp whose embedding is closest to the query embedding according to a predefined distance function. Based on existing benchmark datasets, three new datasets were created that include a variety of complex queries. The proposed method TFLEX is compared against state-of-the-art query embedding methods, where TFLEX outperforms all baselines. Ablation studies show the effectiveness of the particular design of the method.

**Strengths:**

- The paper seems to be the first one to address the multi-hop logical reasoning problem on temporal knowledge graphs, where queries consist of disjunctions of atomic formulas. The authors introduce this new task, which can find application and are relevant in many domains.
- The proposed method supports FOL operations as well as temporal operations and is able to perform multi-hop reasoning.
- Three new datasets are generated, including 40 kinds of complex queries, which can be used for further benchmark experiments.
- The experiments and ablation studies are extensive and confirm the effectiveness of the proposed method.
- The source code and datasets are available online, which supports reproducibility and further evaluation.

**Weaknesses:**

The main weakness concerns the clarity and preciseness of the technical content, which makes it difficult to follow and to understand the methodology, thereby making it challenging to judge the soundness. Here are some examples:
- Figure 1: The query example "During François Hollande was the president of France, which countries did Xi Jinping visit but Obama did not visit?" is given. The logical formulation states that the answer is an entity (country) so that there exists a timestamp T_1 when Xi visited this country and there exists a timestamp T_2 when Obama did not visit this country. This does not seem to reflect the query correctly. The answer should be a country so that there exists a timestamp T_1 when Xi visited this country and for all T_2, Obama did not visit this country. Also, the picture might be easier to understand if the entities for Hollande (e_1, e_3) and France (e_2, e_4) are represented by one node instead of two.
- Line 87: The fact set should be a subset of the complete graph. The set on the right-hand side of the equation, however, denotes the complete graph (all possible quadruples).
- The definitions for the entity query and the timestamp query are difficult to understand. Maybe it is possible to first introduce atomic formulas, then literals, then conjunctions, and last the disjunctive normal form. Some query examples with corresponding query structures could make it more comprehensible. Why should there be the same number (k) of bound variables V_i and T_i? There is only one relation r in the query definition, but is it not common to have several relations (see Figure 1)? Should “Between” also be one of the possibilities for f?
- The method builds on fuzzy logic, which is important for modeling the operators. A short introduction to fuzzy logic directly in the paper (instead of being in the appendix) would help understanding.
- The MLPs in Equation 3 are different, so indices could be added (as in later cases) to make it clearer.
- Line 160: The Alignment Rule is mentioned and is part of the subsequent equations. Since both formulas (for I_e and I_t) look exactly the same, how are the AND operators calculated differently?
- Line 131: ||q|| should be a query set, so is V_q the embedding of all queries or just one query?
- Existing literature for temporal KG completion does not only include the two groups tensor decomposition and translation. There are also methods based on dynamic (time-dependent) embeddings, logical rules, autoregressive models, Markov processes, … (see [1] for a survey from January 2022).
[1] Temporal Knowledge Graph Completion: A Survey. Borui Cai, Yong Xiang, Longxiang Gao, He Zhang, Yunfeng Li, Jianxin Li. https://arxiv.org/abs/2201.08236

**Questions:**

The definitions section and method section contain the formulation for the two cases entity query and timestamp query and include a lot of notation, which impedes the readability. It is sometimes not clear, which equations are needed for which query type. Since both cases are similar in many ways, one suggestion would be to focus on one case, add more details/examples/descriptions, and include the other case in the appendix or mention the differences in a separate section.

Minor comments:
Figure 1: “Obama” -> “Barack Obama” (to be consistent with the other names)
Line 21: The statement “results are inevitably incorrect” is rather strong. Depending on size and content of the KG, results might be correct.
Table 1: The best results could be marked in bold to make it easier to identify the best method.

**Limitations:**

The authors addressed the limitations and corresponding future work adequately in a separate section. Possible broader impact and application areas are also stated in a designated section.

---

> ### Author Rebuttal · Authors · 2023-08-09
>
>
> We would like to express our sincere gratitude for the thoughtful feedback. Below, we address the comments and concerns in a comprehensive manner.
>
> > Figure 1: The logical formulation and the representation of entities in the query example.
>
> The example logical formulation is designed to correctly handle the negation by applying it to the entity set (country) rather than the timestamp set (when François Hollande was the president of France). The set of timestamp already indicates that for-all timestamps in the set. By doing so, we ensure to find the country that meets the specific time constraint for Xi's visit and Obama’s non-visit.
>
> As for the representation of entities for Hollande and France, such representation allows each node corresponds to the term in the above query formulation. We believe that using separate nodes (e_1, e_3, e_2, e_4) enhances clarity and avoids potential confusion in the interpretation of the query.
>
> > The definitions for temporal complex query.
>
> This is a valuable review. Below, we first revise the definitions, then response to the comments. If some parts are still unclear, please let us know. We appreciate your feedback.
>
> Temporal Knowledge Graph (TKG) $G = \{\mathcal{V}, \mathcal{R}, \mathcal{T}, \mathcal{F}\}$ consists of entity set $\mathcal{V}$, relation set $\mathcal{R}$, timestamp set $\mathcal{T}$ and fact set $\mathcal{F} = \{ (s,r,o,t) \} \subseteq \mathcal{V}\times\mathcal{R}\times\mathcal{V}\times\mathcal{T}$ containing subject-predicate-object-timestamp $(s,r,o,t)$ quartets.
> Without loss of generality, $G$ is a first-order logic knowledge base, where each quartet $(s,r,o,t)$ denotes an atomic formula $r(s, o, t)$, with $r$ a binary predicate and $s, o, t$ its arguments.
>
> We focus on Existential Positive First-Order (EPFO) query [1] over TKG, namely Temporal Complex Query $q$, which is categorized into entity query and timestamp query. Formally, the query $q$ consists of a target variable $A$, a non-variable anchor entity set $V_a \subseteq \mathcal{V}$, a non-variable anchor timestamp set $T_a \subseteq \mathcal{T}$, bound variables $V_1,\cdots,V_k$ and $T_1, \cdots, T_l$, logical operations (existential quantification $\exists$, conjunction $\land$, disjunction $\lor$, identity $1$, negation $\lnot$), and extra temporal operations ($\textbf{After}$, $\textbf{Before}$).
> Inspired by [2,3], the disjunctive normal form (DNF) of query $q$ is defined as:
> $$
> \begin{aligned}
>   q[A] = A
>   &\\;\\;:\exists V_1, \cdots, V_k, T_1, \cdots, T_l : (e_{1}^{1} \land \cdots \land e_{n_1}^{1}) \lor \cdots \lor (e_{1}^{m} \land \cdots \land e_{n_m}^{m})\\\\
>   \text{where}
>   & \\;\\; e = f \circ r(V_s, V_o \text{ or } A, g(T))\text{ or }f \circ r(V_s \text{ or } A, V_o, g(T))\\;
>       \text{ if }q\text{ is entity query,}\\\\
>   & \\;\\; e = f \circ r(V_s, V_o, g(T \text{ or } A)) \\;
>       \text{ if }q\text{ is timestamp query} \\\\
>   \text{with}
>   & \\;\\; V_s, V_o \in V_a \cup \\{V_1, \cdots, V_k\\}, T\in T_a \cup \\{T_1, \cdots, T_l\\}, r\in\mathcal{R}, f\in \\{1, \lnot\\}, g\in\\{\textbf{After}, \textbf{Before}\\}\\\\
> \end{aligned}
> $$
>
> In the equation, the DNF is a disjunction of $m$ conjunctions, where $e_{1}^{j} \land \cdots \land e_{n_j}^{j}$ denotes a conjunction between $n_j$ logical atoms, and each $e_i^j$ denotes a logical atom. We ignore indices in the definition of $e_i^j$ to keep the formula clean. The goal of answering the query $q$ is to find the set of entities (or timestamps) $[q]$ that satisfy the query, such that $A\in [ q]$ iff $q[A]$ holds true, where $[ q]$ is the answer set of the query $q$.
>
> - [1] Efficient query evaluation on probabilistic databases.
> - [2] Query2box: Reasoning Over Knowledge Graphs In Vector Space Using Box Embeddings.
> - [3] Beta Embeddings for Multi-Hop Logical Reasoning in Knowledge Graphs.
>
> > Should “Between” also be one of the possibilities for f?
>
> We do not include Between in the definition of f for two reasons. Firstly, Between receives *two* inputs, different from any *unary* f in {1, time negation, After, Before}. Secondly, Between is not atomic, because `Between(t1, t2) = TimeAnd(After(t1), Before(t2))`.
>
> > A short introduction to fuzzy logic in the main body of the paper.
>
> We will add the following introduction to fuzzy logic and vector logic in the method section.
>
> To cope with logical transformation in the vector space, we introduce vector logic, which is a type of fuzzy logic over vector space.
> Fuzzy logic is a generalization of Boolean logic, such that the truth value of a logical atom is a real number in $[0, 1]$. In comparison, as a generalization of a real number, the truth value in vector logic is a vector $[0, 1]^d$ in the semantic space. We denote the logical operations in vector logic as $\textbf{AND}(\land), \textbf{OR}(\lor), \textbf{NOT}(\lnot)$, and so on, which receive one or multiple vectors and output one vector as answer. For more details about fuzzy logic, please refer to Appendix A.
>
> > The indices for MLPs to make it clearer. Line 160: how are the AND operators calculated differently?
>
> It should be clarified that all operators don't share parameters, which means all MLPs are different. We spend efforts to keep the design of operators simple, thus proposing the same structure with trainable parameters for all dyadic operators. So, $\mathcal{I}_e$ and $\mathcal{I}_t$ is the same on logic parts, but not the same on the feature parts. We would revise the notation as follows to make it clearer:
> - $\mathcal{P}_e(\mathbf{V}_q,\mathbf{r},\mathbf{V}_t)         = g(\textbf{MLP}_0^e(\mathbf{V}_q+\mathbf{r}+\mathbf{V}_t))$
> - $\mathcal{P}_t(\mathbf{V}\_{q_1},\mathbf{r},\mathbf{V}\_{q_2}) = g(\textbf{MLP}_0^t(\mathbf{V}\_{q_1}+\mathbf{r}+\mathbf{V}\_{q_2}))$
> - $\alpha_i = …\textbf{MLP}_1 → \alpha^{e,t}_i = …\textbf{MLP}^{e,t}_1$
> - $\beta_i = …\textbf{MLP}_2 → \beta^{e,t}_i = …\textbf{MLP}^{e,t}_2$
>
> > Existing literature for temporal KG completion
>
> see global rebuttal.

---

> ### Author Response · Authors · 2023-08-16
>
> I hope this message finds you well. As the deadline for the reviewer-author discussion draws near, we would like to kindly request your input on our rebuttal. We understand your time is valuable, and we greatly appreciate your initial engagement with our paper.
>
> Your feedback and insights are pivotal to the improvement of our work. If you could spare a moment to review our responses and consider the adjustments we've made, we would be truly grateful. Your evaluation is crucial to the progression of our research, and your thoughtful assessment will undoubtedly contribute to the overall quality of the paper.
>
> Thank you once again for your time and consideration. Please feel free to reach out if you have any questions or require further clarification. We eagerly await your response.

---

> > ### Comment · Reviewer_NJDL · 2023-08-18
> >
> > Thanks for clarifying some of the questions. As other reviewers suggested, you should consider more reference to related work, including (explainable) temporal reasoning on TKGs.

---

> > > ### Author Response · Authors · 2023-08-18
> > >
> > > Thank you for your response! We are delighted to have received your feedback, and your insights are invaluable for enhancing the quality of this paper. Concerning the citations of relevant works, we have added up to 19 additional references on the topic of TKGC in the revised manuscript. These references can be found in the global rebuttal available at (https://openreview.net/forum?id=oaGdsgB18L&noteId=KOfY3P2mqt), showcasing the distinctions and connections between our paper and existing works on the subject. Furthermore, in response to your recent mention of temporal reasoning on TKGs, we have incorporated an additional paper, few-shot temporal reasoning over TKGs [20]. With these amendments, the section on TKG-related works now reads as follows:
> > >
> > > ```markdown
> > > TKGC task aims at inferencing new facts in the TKGs. Existing TKGC methods could be categorized to (1) tensor decomposition [1,2,3], (2) timestamp-based transformation [4,5,6,7,8], (3) dynamic embedding [9,10,11,12], (4) Markov process models [13,14], (5) autoregressive models [15,16,17], (6) others [18,19,20] and so on. Among these works, most of them only confined to the one-hop link prediction task, also known as one-hop reasoning. Some works [12,15,16,17,19] can perform multi-hop reasoning via a path consisting of connected quartets. But none of them could answer logical queries that involve multiple logical operations (conjunction, negation and disjunction). In this paper, we focus on the temporal complex query answering task, which is more challenging than TKGC task.
> > > ```
> > >
> > > This revision includes representative works up until 2022. Despite this, none of the referenced articles delve into reasoning involving complex symbolic logic, as they remain restricted to single-hop and multi-hop reasoning. Therefore, we firmly believe that this paper marks the first multi-hop logical reasoning framework on TKGs with a temporal component that encompasses logic. We are confident that our paper has made indispensable contributions to the field of knowledge graphs, encompassing the introduction of the initial formal definition, the creation of a novel reasoning dataset, the establishment of a TKGR framework, and the identification of future directions for this domain. Your reviews have also consistently acknowledged these contributions. We hope these can serve as justifiable reasons for you to consider a higher rating. It is really important to support our future work.
> > >
> > > Based on your brief response, it appears that aside from the section on related works, you are satisfied with the rebuttal provided. Should you have any further inquiries or if you believe there are other aspects we have overlooked, please do not hesitate to bring them to our attention. We are eager to engage in further communication and discussion with you.
> > >
> > > References:
> > > - [1] Tensor Decomposition-Based Temporal Knowledge Graph Embedding.
> > > - [2] Tensor decompositions for temporal knowledge base completion.
> > > - [3] Tucker decomposition-based Temporal Knowledge Graph Completion.
> > > - [4] ChronoR: Rotation Based Temporal Knowledge Graph Embedding
> > > - [5] Deriving validity time in knowledge graph.
> > > - [6] Leveraging Static Models for Link Prediction in Temporal Knowledge Graphs.
> > > - [7] TeRo: A Time-aware Knowledge Graph Embedding via Temporal Rotation.
> > > - [8] HyTE: Hyperplane-based Temporally aware Knowledge Graph Embedding.
> > > - [9] Temporal Knowledge Graph Completion Based on Time Series Gaussian Embedding.
> > > - [10] DyERNIE: Dynamic Evolution of Riemannian Manifold Embeddings for Temporal Knowledge Graph Completion.
> > > - [11] Know-Evolve: Deep Temporal Reasoning for Dynamic Knowledge Graphs.
> > > - [12] TeMP: Temporal Message Passing for Temporal Knowledge Graph Completion.
> > > - [13] RTFE: A Recursive Temporal Fact Embedding Framework for Temporal Knowledge Graph Completion.
> > > - [14] Learning Dynamic Embeddings for Temporal Knowledge Graphs.
> > > - [15] Recurrent Event Network: Autoregressive Structure Inference over Temporal Knowledge Graphs.
> > > - [16] Temporal Knowledge Graph Reasoning Based on Evolutional Representation Learning.
> > > - [17] Learning Neural Ordinary Equations for Forecasting Future Links on Temporal Knowledge Graphs.
> > > - [18] Explainable Subgraph Reasoning for Forecasting on Temporal Knowledge Graphs.
> > > - [19] Learning to Walk across Time for Interpretable Temporal Knowledge Graph Completion.
> > > - [20] Learning to Sample and Aggregate: Few-shot Reasoning over Temporal Knowledge Graphs

---

> > > ### Comment · Area_Chair_zDMr · 2023-08-20
> > > **Please clarify if added references are sufficient to change your score**
> > >
> > > Dear Reviewer NJDL. I really appreciate that you are insisting on a high standard here. It seems that your questions are all addressed except "more references." Can you comment if you're willing to change your score?
> > >
> > > Second, regarding your comment on "more references", can you clarify if you're asking the authors to
> > >
> > > 1. add more references to the paper, or
> > > 2. substantially change the paper to discuss and compare with more related work, including adding experiments

---

### Official Review · Reviewer_SQqW · 2023-07-06

**Soundness:** 3 good
**Presentation:** 2 fair
**Contribution:** 3 good
**Rating:** 6
**Confidence:** 3

**Summary:**

This paper proposes a method to learn on temporal knowledge graphs, using a combination of fuzzy logic with a temporal extension and node embeddings. To test the method, three new datasets were generated.

The choices made for the model seem logical and are over all well motivated in the paper (e.g. figure 2). The final model was tested on multiple data sets and compared to several non-trivial baselines. Furthermore, ablation studies were performed to analyze the effect of several components of the TFLEX model.

While this paper has several strong points when it comes to presentation (e.g. figure 2 and the colors in line 164), there are some lacking areas. Firstly, having to much mathematical notation spread throughout the text itself (e.g. lines 173-180) compromises readability. Secondly, given the novelty of the model, I would have liked to see a detailed diagram representing the model.

This paper provides a novel approach for learning a temporal graphs and provides three data sets for this task. Research on temporal graphs is somewhat limited (as seen by the limitations of the baselines), making this a welcome addtion to the sub-area of machine learning on graphs.




**Strengths:**

This paper performs a analysis by motivating choices and thoroughly evaluating the model. The figures that are present are very clear.

**Weaknesses:**

Some of the math could have been beter separated from the text, making the paper more readable.  There is not figure to visually convey the model itself.

**Questions:**

To me, it is not clear why we would want up to n compositions of the identity, negation, before and after operations (see line 105).
  In lines 124-125 you seem to assume a query only has one answer. Is this only for training purposes, or is this a limitation of the model?

**Limitations:**

The model seems to be limit the answer set of queries to just one answer (the embedding closest to the query embedding).

---

> ### Author Rebuttal · Authors · 2023-08-09
>
> We appreciate the reviewer's thoughtful assessment of our paper. We would like to address the points raised in the review and provide clarification and additional information to enhance the overall understanding and appreciation of our work.
>
> > To me, it is not clear why we would want up to n compositions of the identity, negation, before and after operations (see line 105).
>
> Well, we also find that it is not necessary. We will remove it in the revision. For now, you can refer to our response to reviewer NJDL, where we present a revised version of the definition.
>
> > In lines 124-125 you seem to assume a query only has one answer. Is this only for training purposes, or is this a limitation of the model?
>
> To clarify, a query has an answer set instead of one answer only. Each answer in the answer set is named Temporal Query Answer. For each answer, its embedding should be close to the query embedding.
>
> Once again, we appreciate the opportunity to receive feedback and engage in this scholarly discourse, and we are dedicated to delivering an enhanced version of our paper that aligns with the reviewer's and the broader academic community's expectations.

---

> > ### Comment · Area_Chair_zDMr · 2023-08-20
> > **Reviewer please acknowledge having read the rebuttal**
> >
> > Review SQqW please let me know if your questions are addressed and want to revise your score if applicable.

---

### Official Review · Reviewer_BpBt · 2023-07-07

**Soundness:** 4 excellent
**Presentation:** 3 good
**Contribution:** 4 excellent
**Rating:** 7
**Confidence:** 4

**Summary:**

This paper studied the multi-hop logical reasoning problem on temporal knowledge graphs and proposed the first temporal complex query embedding framework named Temporal Feature-Logic Embedding framework(TFLEX).Firstly, they defined the task of multi-hop logical reasoning over TKGs. Secondly, they designed the first multi-hop logical reasoning framework which utilizes fuzzy logic to compute the logic part and extend fuzzy logic on timestamp set for supporting all FOL operation and extra temporal operations(After, Before and Between). At last, they generated three new TKG datasets for the task of multi-hop logical reasoning. Experiments on benchmark datasets demonstrate the efficacy of the proposed framework in deal with different operations in complex queries.

**Strengths:**

1.	They creatively conducted research multi-hop logical reasoning problem on the TKGs and proposed the first TFLEX framework to answer the temporal complex queries.
2.	they relatively proposed using fuzzy logic to handle temporal feature-logical embedding and expanded extra temporal operations.
3.	They provided three new TKG datasets and compared them with multiple benchmarks for experimental verification in order to better validate the reliability of their framework.
4.	They had clear explanations and explanations of the definitions and methods used, with reasonable classification and appropriate illustrations.
5.	There are more detailed compensations and explanations in the appendix to better explain the details of the method and experiment.
6.	There are also corresponding open source code and datasets.


**Weaknesses:**

1.	In the section on related work, there is a lack of appropriate explanations for the problems and shortcomings of the methods cited in the relevant articles, as well as a comparison with current work.
2.	The appendix section provides some overly detailed supplements to the overall concept, such as a more concise description of fuzzy logic.
3.	It is not recommended to directly apply source code for explanation and display in the appendix section.


**Questions:**

1.	It is recommended that the author compare the methods of relevant articles with the improvements of the author's article to highlight the efficiency of their methods
2.	It is suggested that the author can further optimize the content of the paper to streamline the appendix content, and simplify the subsection: Explaining answers with the framework and Experimental analysis into the main body of the paper to better demonstrate its experiments and principles.
3.	It is suggested that Pseudocode should be used for interpretation, which is helpful for reading and analysis and is more standardized.


**Limitations:**

The author thoroughly analyzed its limitations in the article, including insufficient time operators, improved time embedding, long query generation time, MRR and Hits@k Weak. Also proposed its plan for future improvement work

---

> ### Author Rebuttal · Authors · 2023-08-09
>
>
> We appreciate the reviewer's time and effort in evaluating our manuscript. We have carefully considered the provided feedback and would like to address each point of concern:
>
> > Lack of comparison with related work:
>
> (see pdf in the global rebuttal section)
>
> We acknowledge the reviewer's comment regarding the need for more explicit discussions about the problems and shortcomings of the methods cited in the related articles. In the pdf in global rebuttal section, we include a comprehensive comparison between the methods outlined in the related work and our proposed approach. This will not only highlight the limitations of existing methods but also underscore the advancements offered by our approach.
>
> > Overly detailed supplements in the appendix:
>
> We understand the reviewer's concern about the detailed supplements in the appendix. While we believe that these supplemental explanations contribute to a deeper understanding of the concepts, we will revisit the appendix section to ensure that the content remains concise and directly relevant to the main concepts presented in the paper.
>
> > Using pseudocode for interpretation:
>
> We appreciate the suggestion to use pseudocode for better interpretation. In the revised manuscript, we will include pseudocode to present the key algorithms and operations in our TFLEX framework. This will enhance the readability and facilitate a standardized understanding of our proposed methodology.
>
> > Streamlining appendix content and simplifying subsections:
>
> We take the reviewer's point about streamlining the appendix content and simplifying certain subsections. To enhance the readability and coherence of the paper, we will integrate the content from the subsections "Explaining answers with the framework" and "Experimental analysis" into the main body of the paper. This adjustment will help our readers better understand the experimental setup and principles underlying our work.
>
> ----
>
> In conclusion, we are genuinely thankful for the reviewer's insightful feedback, which has guided us towards refining our manuscript. We are committed to addressing each concern and ensuring that the revised paper comprehensively presents our work while effectively addressing the points raised.
> Thank you for your time and consideration.

---

### Official Review · Reviewer_AEhV · 2023-07-07

**Soundness:** 3 good
**Presentation:** 3 good
**Contribution:** 3 good
**Rating:** 7
**Confidence:** 3

**Summary:**

Authors present new embedding framework called TFLEX to embed complex temporal queries over temporal knowledge graphs to perform multi-hop reasoning with time constraints on TKGs. Authors present the overall embedding framework using Fuzzy logic to model complex logical queries and extending fuzzy logic to include three temporal operators After, Before and Between. It presents new benchmark datasets to evaluate embedding on TKGs and show experimental results to show the benefits of proposed approach.

**Strengths:**

paper presents a new embedding framework and dataset for complex temporal queries, which can benefit the community in extending research in this direction.

**Weaknesses:**

Not sure if this framework can handle complex queries without temporal constraints on par with other complex query embedding methods?
suppose if we ignore the temporal aspect of current framework and run the standard benchmarks on complex query handling does it work as well as other methods in the literature.

**Questions:**

Can this solve complex query answering equally well on other bench mark tasks and how does it fare against the prior art?

**Limitations:**

Yes

---

> ### Author Rebuttal · Authors · 2023-08-10
>
>
> We appreciate the reviewer's constructive feedback on our paper. We are pleased that the reviewer recognizes the merits of our work and acknowledges the contributions we have made to the field. We would like to address the concerns and questions below.
>
> > Not sure if this framework can handle complex queries without temporal constraints on par with other complex query embedding methods? suppose if we ignore the temporal aspect of current framework and run the standard benchmarks on complex query handling does it work as well as other methods in the literature. Can this solve complex query answering equally well on other bench mark tasks and how does it fare against the prior art?
>
> It's an interesting question. And of course, the answer is yes. Ignoring the time parts of TFLEX degenerates to the variant FLEX in our experiments.
> We report the results of FLEX on the standard datasets FB237, FB15k and NELL, comparing to famous QE baselines (GQE, Query2box, BetaE, ConE) in the tables below.
> From the tables, we observe that FLEX achieves competitive performance compared to the four baselines. This is similar to the situation on temporal complex queries over TKGs.
> We attribute the improvement to the fuzzy logic operators, which can handle complex queries better than the geometric operators in QE baselines.
>
> Table 1. MRR results for answering queries without negation ($\exists$, $\land$, $\lor$) on FB15k, FB237 and NELL. The best results are in bold. **AVG** denotes average performance.
>
> | **Dataset** | **Model** | **1p**   | **2p**   | **3p**   | **2i**   | **3i**   | **pi**   | **ip**   | **2u**   | **up**   |  **AVG** |
> | :---------- | :-------- | :------- | :------- | :------- | :------- | :------- | :------- | :------- | :------- | :------- | -------: |
> | FB15k       | GQE       | 53.9     | 15.5     | 11.1     | 40.2     | 52.4     | 27.5     | 19.4     | 22.3     | 11.7     |     28.2 |
> |             | Q2B       | 70.5     | 23.0     | 15.1     | 61.2     | 71.8     | 41.8     | 28.7     | 37.7     | 19.0     |     40.1 |
> |             | BetaE     | 65.1     | 25.7     | 24.7     | 55.8     | 66.5     | 43.9     | 28.1     | 40.1     | 25.2     |     41.6 |
> |             | ConE      | 73.3     | 33.8     | 29.2     | 64.4     | 73.7     | 50.9     | 35.7     | **55.7** | 31.4     |     49.8 |
> |             | FLEX      | **77.1** | **37.4** | **31.6** | **66.4** | **75.2** | **54.2** | **42.4** | 52.9     | **34.3** | **52.4** |
> |             |
> | FB237       | GQE       | 35.2     | 7.4      | 5.5      | 23.6     | 35.7     | 16.7     | 10.9     | 8.4      | 5.8      |     16.6 |
> |             | Q2B       | 41.3     | 9.9      | 7.2      | 31.1     | 45.4     | 21.9     | 13.3     | 11.9     | 8.1      |     21.1 |
> |             | BetaE     | 39.0     | 10.9     | 10.0     | 28.8     | 42.5     | 22.4     | 12.6     | 12.4     | 9.7      |     20.9 |
> |             | ConE      | 41.8     | 12.8     | 11.0     | 32.6     | 47.3     | 25.5     | 14.0     | 14.5     | 10.8     |     23.4 |
> |             | FLEX      | **43.6** | **13.1** | **11.1** | **34.9** | **48.4** | **27.4** | **16.1** | **15.4** | **11.1** | **24.6** |
> |             |
> | NELL        | GQE       | 33.1     | 12.1     | 9.9      | 27.3     | 35.1     | 18.5     | 14.5     | 8.5      | 9.0      |     18.7 |
> |             | Q2B       | 42.7     | 14.5     | 11.7     | 34.7     | 45.8     | 23.2     | 17.4     | 12.0     | 10.7     |     23.6 |
> |             | BetaE     | 53.0     | 13.0     | 11.4     | 37.6     | 47.5     | 24.1     | 14.3     | 12.2     | 8.5      |     24.6 |
> |             | ConE      | 53.1     | 16.1     | 13.9     | 40.0     | **50.8** | 26.3     | 17.5     | 15.3     | 11.3     |     27.2 |
> |             | FLEX      | **57.8** | **16.8** | **14.7** | **40.5** | **50.8** | **27.3** | **19.4** | **15.6** | **11.6** | **28.2** |
>
> Table 2. MRR results for answering queries with negation on FB15k, FB237, and NELL. The best results are in bold. **AVG** denotes average performance.
>
> | **Dataset** | **Model** | **2in**  | **3in**  | **inp**  | **pin**  | **pni**  |  **AVG** |
> | :---------- | :-------- | :------- | :------- | :------- | :------- | :------- | -------: |
> | FB15k       | BetaE     | 14.3     | 14.7     | 11.5     | 6.5      | 12.4     |     11.8 |
> |             | ConE      | 17.9     | 18.7     | 12.5     | 9.8      | 15.1     |     14.8 |
> |             | FLEX      | **18.0** | **19.3** | **14.2** | **10.1** | **15.2** | **15.4** |
> |             |
> | FB237       | BetaE     | 5.1      | 7.9      | 7.4      | 3.6      | 3.4      |      5.4 |
> |             | ConE      | 5.4      | 8.6      | 7.8      | **4.0**  | **3.6**  |      5.9 |
> |             | FLEX      | **5.6**  | **10.7** | **8.2**  | **4.0**  | **3.6**  |  **6.5** |
> |             |
> | NELL        | BetaE     | 5.1      | 7.8      | 10.0     | 3.1      | 3.5      |      5.9 |
> |             | ConE      | 5.7      | 8.1      | 10.8     | 3.5      | 3.9      |      6.4 |
> |             | FLEX      | **5.8**  | **9.1**  | **10.9** | **3.6**  | **4.1**  |  **6.7** |

---

### Author Rebuttal · Authors · 2023-08-10


Dear Reviewers,

We would like to express our sincere gratitude to the reviewers for their thoughtful and constructive feedback on our submission. We greatly appreciate the time and effort dedicated to evaluating our work, and we are excited to engage in this rebuttal process to address the raised concerns and comments.

Below, we provide the response for the points that mostly concerned.

> [@BpBt, @NJDL, @Zrqw] Existing literature for temporal KG completion and comparison with related TKGC works.

We select some related TKGC works that we have already read before or cited by the survey (Temporal Knowledge Graph Completion: A Survey.) mentioned by reviewer NJDL. We would like to add the following sentences to the related work section.

```md
TKGC task aims at inferencing new facts in the TKGs.
Existing TKGC methods could be categorized to (1) tensor decomposition [1,2,3], (2) timestamp-based transformation [4,5,6,7,8], (3) dynamic embedding [9,10,11,12], (4) Markov process models [13,14], (5) autoregressive models [15,16,17], (6) others [18,19] and so on. Among these works, most of them only confined to the one-hop link prediction task. Some works [12,15,16,17,19] can perform multi-hop reasoning via a path consisting of connected quartets. But they cannot answer logical queries that involve multiple logical operations involving conjunction, negation and disjunction. In this paper, we focus on the temporal complex query answering task, which is more challenging than TKGC task.
```

> [@Zrqw] A separate table comparing TFLEX and state-of-the-art TKGC on a completion task.

we attach the table in the pdf.

> [@BpBt, @NJDL, @Zrqw] Improvement of readability

In order to improve the readability of the paper, we plan to revise as follows:


1. More readable notations in the formulas. More explanations about the symbols. [#our response to NJDL and Zrqw].
2. A leading sentence in the method section.
   ```md
   In this section, we replace the variables in the query formulation with temporal feature-logic embeddings, and perform logical operations via neural networks. We first introduce the temporal feature-logic embedding for entities, timestamps, and queries in Section 4.1. Afterwards, we introduce logical operators in Section 4.2 and how to train the model in Section 4.3.
   ```
3. A short introduction to fuzzy logic to be presented in method section. Please refer to our response to NJDL for detail. [#our response to NJDL]
4. A brief introduction to query types in the dataset setting in experiment section [#our response to Zrqw].
5. Streamlining appendix content and simplifying subsections. [#our response to BpBt]

------

In conclusion, we once again thank the reviewers for their valuable insights and feedback. We have carefully considered all the comments and suggestions and have made corresponding revisions to improve the quality and clarity of our work. We believe that our paper contributes significantly to the field by presenting a novel approach for complex logical reasoning over temporal knowledge graphs. We are confident that the revisions we have made address the reviewers' concerns. We look forward to the opportunity to present our findings and engage in insightful discussions with the community.
Thank all for the time and consideration.

------

References:

- [1] Tensor Decomposition-Based Temporal Knowledge Graph Embedding.
- [2] Tensor decompositions for temporal knowledge base completion.
- [3] Tucker decomposition-based Temporal Knowledge Graph Completion.
- [4] ChronoR: Rotation Based Temporal Knowledge Graph Embedding
- [5] Deriving validity time in knowledge graph.
- [6] Leveraging Static Models for Link Prediction in Temporal Knowledge Graphs.
- [7] TeRo: A Time-aware Knowledge Graph Embedding via Temporal Rotation.
- [8] HyTE: Hyperplane-based Temporally aware Knowledge Graph Embedding.
- [9] Temporal Knowledge Graph Completion Based on Time Series Gaussian Embedding.
- [10] DyERNIE: Dynamic Evolution of Riemannian Manifold Embeddings for Temporal Knowledge Graph Completion.
- [11] Know-Evolve: Deep Temporal Reasoning for Dynamic Knowledge Graphs.
- [12] TeMP: Temporal Message Passing for Temporal Knowledge Graph Completion.
- [13] RTFE: A Recursive Temporal Fact Embedding Framework for Temporal Knowledge Graph Completion.
- [14] Learning Dynamic Embeddings for Temporal Knowledge Graphs.
- [15] Recurrent Event Network: Autoregressive Structure Inference over Temporal Knowledge Graphs.
- [16] Temporal Knowledge Graph Reasoning Based on Evolutional Representation Learning.
- [17] Learning Neural Ordinary Equations for Forecasting Future Links on Temporal Knowledge Graphs.
- [18] Explainable Subgraph Reasoning for Forecasting on Temporal Knowledge Graphs.
- [19] Learning to Walk across Time for Interpretable Temporal Knowledge Graph Completion.

---

### Decision · Program_Chairs · 2023-09-21

**Decision:**

Accept (poster)

**Comment:**

The paper introduces TFLEX, a novel embedding framework to embed complex temporal queries over temporal knowledge graphs to perform multi-hop reasoning with time constraints on TKGs. TFLEX uses fuzzy logic to model complex logical queries and extends to include three temporal operators (After, Before and Between). They provided three new TKG datasets and compared them with multiple benchmarks for experimental verification in order to better validate the reliability of their framework. As a result, TFLEX is considered by reviewers a valuable addition to the field of learning temporal graphs.